# Omni-Mol: Multitask Molecular Model for Any-to-any Modalities

**Chengxin Hu**[1,*]  **Hao Li**[2,*]  **Yihe Yuan**[1,*]  **Zezheng Song**[3]
**Chenyang Zhao**[4]  **Haixin Wang**[4,†]
[1] National University of Singapore    [2] Independent Researcher
[3] University of Maryland, College Park    [4] University of California, Los Angeles
{e1324268@u.nus.edu, whx@ucla.edu}

 **GitHub:** Omni-Mol-Code        **HuggingFace:** Omni-Mol-Data&Weight

## Abstract

In the molecular domain, numerous studies have explored the use of multimodal large language models (LLMs) to construct a general-purpose, multi-task molecular model. However, these efforts are still far from achieving a truly universal molecular model. We identify three key challenges in this endeavor: (1) Existing molecular task datasets are typically small in scale and lack comprehensive domain coverage. (2) Tasks from different molecular subfields are difficult to effectively learn jointly through LLMs due to significant distributional shifts and competition among tasks, which introduces instability in the learning process. (3) Both inter-task and intra-task molecular representations demand different intrinsic dimensions in the language space, making it challenging to balance between redundancy and insufficiency in language model representations. To address these challenges, we innovatively categorize existing small-molecule tasks into four types: Mol2Mol, Mol2Text, Mol2Num, and Text2Mol. We then collect a dataset encompassing over 16 tasks with more than 1.4 million samples, making it the largest molecular instruction-tuning dataset to date. Leveraging the extensive pretraining of LLMs on existing chemical literature, we propose a novel multimodal LLM framework, named **Omni-Mol**, which unifies all small-molecule tasks and supports both molecular generation and understanding. The core of Omni-Mol is our proposed MoGE, which dynamically adapts to the intrinsic rank of different tasks. This mixture-of-experts architecture enhances the model's ability to handle diverse tasks and modalities effectively. Our model achieves unified instruction tuning across 16 tasks and attains state-of-the-art performance on 13 of them. Extensive experiments further demonstrate the scalability and versatility of Omni-Mol.

## 1 Introduction

Large language models (LLMs), especially multimodal LLMs, have achieved significant breakthroughs in various scientific tasks due to their powerful representational capabilities and general reasoning abilities, spanning domains such as medicine [29, 82], chemistry [6], and biology [77]. This cutting-edge technology has also sparked an increasing number of studies exploring how to align molecular representation spaces with textual representation spaces [10, 13, 21, 11, 25]. These works hold great promise to build powerful AI chemists for advancing molecule captioning, property/structure prediction, and text-conditioned de novo drug design.

---

[*]Equal contribution, listings are alphabetic.
[†]Corresponding author.

39th Conference on Neural Information Processing Systems (NeurIPS 2025).

| Benchmark | 1. No LLM Pre-training | 2. One model fits all | 3. 3D adaptive ability | 4. Data scaling | 5. Parameter scaling | 6. Task type |
|---|---|---|---|---|---|---|
| InstructMol [10] | ✓ | ✗ | ✗ | ✗ | ✓ | 3 |
| HIGHT [13] | ✓ | ✗ | ✗ | ✗ | ✗ | 3 |
| PRESTO [11] | ✗ | ✓ | ✗ | ✓ | ✗ | 2 |
| 3D-MoLM [41] | ✓ | ✗ | ✓ | ✗ | ✗ | 2 |
| ReactXT [48] | ✗ | ✗ | ✗ | ✗ | ✗ | 2 |
| Omni-Mol | ✓ | ✓ | ✓ | ✓ | ✓ | **4** |

Table 1: A comprehensive comparison between Omni-Mol and other molecular LLMs.

The first step in creating an AI chemist is to develop a generalist model with universal capabilities, enabling it to understand diverse molecular structures and their interactions under multiple chemical domains. Pioneering works, such as Text+Chem T5 [14], introduce the first multi-domain, multi-task language model capable of unifying molecular and textual representations. Following this, the recent state-of-the-art PRESTO [11] further enhances performance by progressively improving multimodal LLMs through cross-modal alignment and multi-graph understanding.

From this trend, it can be seen that the community is pursuing a "one-model-fits-all" [83, 79] paradigm, rather than using models like InstructMol [10] that rely on different Low-Rank Adapters(LoRA) [26] to learn different tasks.

However, existing approaches remain far from achieving a truly general-purpose molecular model. For instance, while PRESTO can generate molecules given molecular inputs or predict properties based on molecular structures, it does not support tasks such as describing molecules or designing molecules according to specified textual requirements. To date, it is rare to find a model that supports a sufficiently broad range of task types under a unified "one-model-fits-all" framework. We identify three main challenges in constructing such a universal molecular model. First, existing molecular task datasets are generally small in scale and lack coverage across diverse domains. Second, molecular tasks from different domains exhibit significant distributional discrepancies, making it difficult for LLMs to learn effectively and stably across tasks. Finally, both intra-task molecular instances and inter-task representations differ in their intrinsic dimensionality within language space, making it challenging for the model to balance redundancy and insufficiency. These issues impede the development of a general-purpose AI expert for molecular tasks.

In this paper, we seek the answer to the following question:

> *Is it possible to develop a generalist molecular LLM capable of effectively learning across diverse task domains?*

This question drives us to develop **Omni-Mol**, a scalable and general-purpose Multimodal LLM-based framework for unified molecular understanding and generation. Omni-Mol provides four key innovations. **(1)** we conduct a comprehensive investigation of small molecule tasks and innovatively categorize these tasks according to their input-output modalities into four types: `Mol2Mol`, `Mol2Text`, `Mol2Num`, and `Text2Mol`. Subsequently, we construct the Omni-Mol dataset, which comprises over 1.4 million samples and represents the most extensive instruction-tuning dataset for small molecule tasks to date. **(2)** Leveraging this dataset, we propose a unified instruction tuning paradigm and build the most comprehensive general-purpose multimodal molecular LLM based on LLaMA 3 [18] for the first time. **(3)** To address the challenge of varying intrinsic dimensions across different domains and tasks, we propose Gradient Adaptive LoRA (GAL), a novel adaptive mechanism that extends existing LoRAs [26, 17, 68, 74, 73, 67] to better handle multi-task learning scenarios. GAL mitigates conflicts that arise when standard LoRA struggles to accommodate dynamically shifting intrinsic dimensions during training. **(4)** To further manage inter-task and cross-modal interference, we adopt a Mixture-of-Experts (MoE) framework to develop a Mixture-of-GAL-Experts (MoGE) fine-tuning strategy. By integrating shared experts and routed experts, our model is capable of both robustly capturing general knowledge and differentiating across diverse tasks.

Comprehensive experiments on our datasets show that Omni-Mol achieves significant improvements across 13 tasks simultaneously, setting new state-of-the-art results among both finetuned open-source LLMs and in-context learned closed-source LLMs. Additionally, we observe that Omni-Mol scales effectively with increases in data volume and model size, indicating the model's tremendous potential under larger computational budgets. Furthermore, by analyzing the representations of models trained on progressively more tasks, we discover that the representations become increasingly similar as the number of tasks grows. This provides robust evidence that the model is learning general

| Task | #Samples |  | Task | #Samples |
|---|---|---|---|---|
| Mol2Mol | 688,725 |  | Mol2Num | 412,231 |
| Forward | 125,384 |  | HO.LU. | 263,100 |
| Reagent | 125,384 |  | Weight | 13,979 |
| Retrosyn | 129,684 |  | TPSA | 13,979 |
| Solvent | 74,892 |  | LogP | 12,458 |
| Catalyst | 11,094 |  | Yield | 9,715 |
| MolEdit | 222,287 |  |  |  |

| Task | #Samples |  | Task | #Samples |
|---|---|---|---|---|
| Mol2Text | 247,652 |  | Text2Mol | 72,559 |
| Experiment | 147,788 |  | I2S | 57,575 |
| DQA | 66,885 |  | MolDesign | 14,984 |
| Molcap | 32,979 |  |  |  |

Table 2: Task statistics, HO.LU.: HOMO-LUMO prediction, I2S: IUPAC2SELFIES. The total molecule data volume reaches 1.42 million.

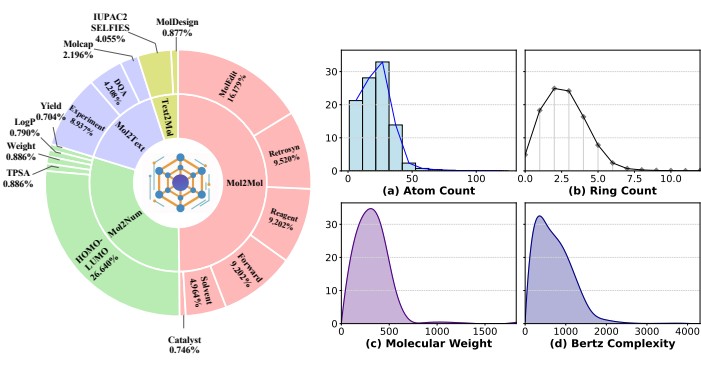

(a) Domain and tasks     (b) Molecule statistics

Figure 1: The sunburst chart of Omni-Mol dataset tasks and the statistics of the molecules. Our dataset exhibits substantial diversity across several molecular attributes, including atom count, ring count, molecular weight, and Bertz complexity.

representations effectively. We hope our dataset and model can pave the way for the community to build more powerful generalist AI chemists.

## 2 Related Works

### 2.1 Molecular Foundation Models

Researchers are trying to leverage the world knowledge embedded in LLMs to build higher-quality molecular representations by fine-tuning on task-specific instructions. Mol-Instruction [21] pioneers the instruction fine-tuning dataset, demonstrating the potential of LLMs in molecular modeling. Subsequently, InstructMol [10] introduces 2D graph features of molecules based on SMILES [69], showing that LLMs can also enhance performance by aligning and fine-tuning their understanding of graph-based features. Soon after, 3D-MoLM [41] explores the advantages of 3D molecular representations in multimodal LLMs, while HIGHT [13] investigates the impact of multi-level 2D graph features on molecular understanding. More recently, PRESTO [11] enhances LLMs' comprehension of molecular-related knowledge through extensive domain-specific pretraining across eight tasks.

### 2.2 Unified Generative Modeling

The GPT models [7, 1] have achieved unification across all text-based tasks through large-scale pretraining and instruction tuning. Subsequently, the community has successfully constructed models that can understand data from multiple modalities and simultaneously perform tasks related to different modalities by converting features from each modality into tokens [4, 40, 39, 16, 44]. More recently, the community has also been exploring unified understanding and generation, allowing models not only to understand multimodal data but also to generate multimodal data [83, 79, 34]. This development is driving models towards convergence into a truly general-purpose model capable of solving all tasks. [27] suggests that as models grow more powerful and general, their representations tend to converge, approaching a universal space that reflects the fundamental laws of the world. This insight inspires us to explore whether a multi-task generalist also exists in the molecular domain.

## 3 Overview

Omni-Mol is a multimodal LLM framework to handle $K$ understanding and generative molecular tasks simultaneously. It comprises a language model, a graph encoder $f_{\mathcal{G}}$, and a projector $f_p$. The inputs include a text instruction $\mathbf{X}_I$, a SELFIES string $\mathbf{X}_S$, and the graph data $\mathbf{X}_G$ corresponding to the input molecules, where $\mathbf{X}_G$ is converted from $\mathbf{X}_S$ using RDKit. We model the response $\mathbf{Y}$ as the

probability of the next token as:

$$P(\mathbf{Y}|\mathbf{X}_I, \mathbf{X}_S, \mathbf{H}_G) = \prod_i P_\theta(\mathbf{Y}_i|\mathbf{X}_I, \mathbf{X}_S, \mathbf{H}_G, \mathbf{Y}_{<i}) \tag{1}$$

where $\mathbf{H}_G = f_p(f_\mathcal{G}(\mathbf{X}_G))$, and $\theta$ is the parameter of the LLM. The graph encoder encodes the molecule graph into its representation $\mathbf{h_g} \in \mathbb{R}^{n \times d_1}$, where $n$ is the length of the representation, the projector then projects its dimension to the LLM's hidden size and obtain $\mathbf{H}_G \in \mathbb{R}^{n \times d_2}$. The overview is shown in Figure 2. The complete multimodal architecture details are in Appendix C.

## 4 Omni-Mol Data Curation

The first step in constructing Omni-Mol is the collection of comprehensive and diverse data. In our review of existing work, we identify a wide range of chemical tasks, whose inputs and outputs can generally be categorized into the following modalities: Text, 1D molecular sequences, and Tabular numerical value. For instance, a typical chemical reaction task involves mapping input molecules to another molecule, which falls under the category of molecule-to-molecule tasks. Some studies [10, 11, 13] incorporate graph neural networks (GNNs) to encode molecular graph information as input; however, graph features are typically not used as outputs.

After a systematic review, we categorize the tasks into the following four major types: (1) `Mol2Mol`, (2) `Mol2Num`, (3) `Mol2Text`, and (4) `Text2Mol`. We observe that existing works are not yet capable of achieving learning across arbitrary modality pairs. For example, PRESTO covers `Mol2Num` and `Mol2Mol` tasks but lacks support for `Mol2Text` and `Text2Mol`. Similarly, InstructMol and 3D-MoLM include `Mol2Num`, `Mol2Mol`, and `Mol2Text` tasks, but do not support `Text2Mol`. Omni-Mol will be trained across all four types of tasks.

**Data format**. To this end, we construct a unified instruction-tuning dataset, standardizing the data format as follows: (1) A unique, clear, and concise instruction; (2) A 1D representation of the molecule (no molecule input for Text2Mol tasks); (3) The corresponding task output.

**SELFIES v.s. SMILES**. Both SELFIES [37] and SMILES [69] are 1D modalities for representing molecules as text. In the Omni-Mol Dataset, we opted for the SELFIES representation. We observed that SMILES strings may fail to be recognized by RDKit [38]; this issue is particularly pronounced in SMILES generated by LLMs. In contrast, a syntactically correct SELFIES string can be robustly decoded into a valid molecule [37]. This reliable decoding is necessary to guarantee the conversion to a 2D graph, which in turn ensures the generation of a 3D graph. These 2D and 3D graph representations are critical for downstream tasks such as molecular docking and conformer generation.

**Preprocessing**. After collecting the data, we perform a series of preprocessing steps and establish comprehensive metrics for molecular LLMs on the selected task. We also look into tasks that are similar to each other and remove samples that are potentially data leakage. To further investigate the understanding of 3D molecules by LLMs, following 3D-MOIT [41], we preprocess the Omni-Mol data using RDKit to obtain the **3D** representation graphs of the molecules. Detailed information can be found in Appendix B.

**Unified Encoding**. We encode data from different modalities uniformly into tokens. For molecular 1D representations, textual data, and tabular numerical data, we convert them into character sequences and tokenize them using the tokenizer and word embeddings of the LLM. For molecular graph representations, we utilize the node embeddings generated by a GNN as input tokens. Since the lengths of molecular sequences and graph embeddings vary, we apply padding to standardize their shapes into a uniform tensor format, thereby enabling parallel training.

**Dataset Statistic.** We conducted a comprehensive statistical analysis of the entire dataset, including metrics such as atom count and ring count. The task types along with the names of individual tasks in Figure 1. The final Omni-Mol dataset contains over 1.4 million samples, providing ample training fuel for the development of Omni-Mol. The detailed information on each task and the chat template can be found in Appendix B. Moreover, we introduce MolEdit as a novel task within our collected data. As the original data was not formatted for instruction-following, we designed and authored specific instructions for molecular editing.

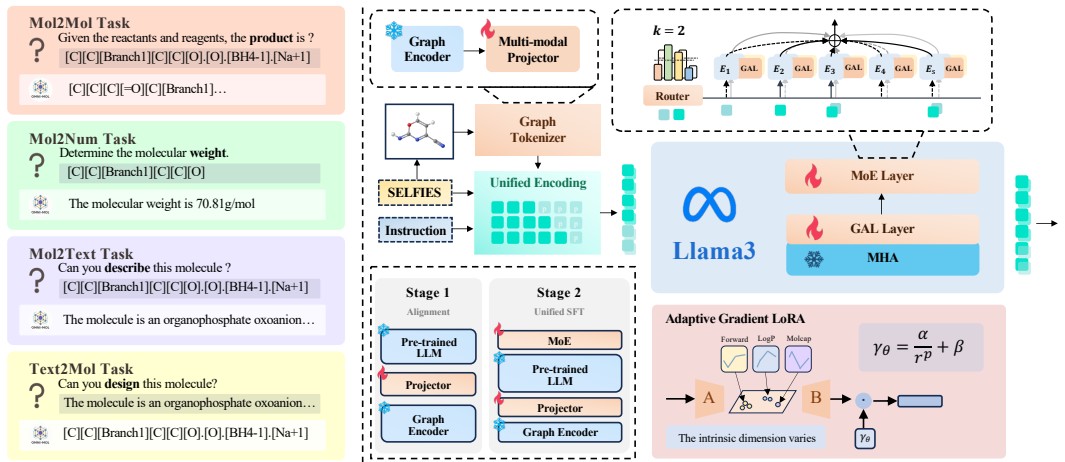

Figure 2: Overview of our proposed Omni-Mol, a scalable and general-purpose molecular LLM-based framework for both understanding and generation.

# 5  Method

## 5.1  Gradient Adaptive LoRA

In a standard LoRA framework, the update $\Delta\mathcal{W}$ of the model's weight $\mathcal{W}_0$ is defined as: $\mathcal{W}' = \mathcal{W}_0 + \Delta\mathcal{W} = \mathcal{W}_0 + \gamma\mathcal{B}\mathcal{A}$, where $\mathcal{A}$ and $\mathcal{B}$ are low rank matrices. As defined in the original paper of LoRA [26], the scaling factor $\gamma$ depends on the rank $r$ of the LoRA, *i.e.*, $\gamma = \gamma(r) = \alpha/r$, where $\alpha$ is a hyper-parameter that controls the overall update magnitude of the low-rank adaptation and balance adaptation capacity and training stability.

We conduct experiments on tasks across different domains in an attempt to identify an appropriate optimal rank. As shown in Figure 3, we observe that the performance of each domain-specific task varies under different rank settings, and the optimal rank also differs across domains. For instance, the optimal rank for forward prediction is 128, whereas for molcap it is 32. We attribute this to the differences in the intrinsic dimensionality of attention weights across tasks [2]. Therefore, employing a static adapter to simultaneously learn multiple molecular tasks may be suboptimal.

During multi-task training, certain tasks provide highly informative signals, whereas others contribute predominantly redundant information. Consequently, it is desirable to introduce a dynamic coefficient that adaptively amplifies the gradients associated with each task throughout the training process.

We propose an adaptive adapter, **G**radient **A**daptive **L**oRA (GAL), which introduces a dynamic scaling factor to modulate the fusion of the updated weights, $\gamma_\theta = \alpha/r^p + \beta$, where $\theta = \{\alpha, p, \beta\}$ are learnable parameters. Here, the $p$ exponent lets us model rank effects and $\beta$ can provide a direct adjustment to the scaling factor. This simple yet effective modification enables the adapter to dynamically adjust its scaling factor during training, allowing it to better adapt to the intrinsic dimension of the data.

Through this approach, we can dynamically adjust the amplitude of the gradient. The backward propagation of a LoRA on the downstream fine-tuning data $\mathcal{D}$ is modified as:

$$\nabla_{\Delta\mathcal{W}} = \frac{\partial\mathcal{L}\left(\mathcal{D}; \mathcal{W}_0 + \Delta\mathcal{W}\right)}{\partial\Delta\mathcal{W}} \rightarrow \nabla_{\Delta\mathcal{W}} = \frac{\partial\mathcal{L}\left(\mathcal{D}; \mathcal{W}_0 + \boxed{\gamma_\theta} \cdot \Delta\mathcal{W}\right)}{\partial\Delta\mathcal{W}} \tag{2}$$

Implementation details are provided in Appendix C.

## 5.2  Mixture-of-GAL-Experts (MoGE) Expansion

Omni-Mol needs to learn a wide range of different tasks and handle multiple modalities, including graph features, text, and SELFIES. While SELFIES is treated as regular text input to the LLM, it inherently differs significantly from natural language semantics, requiring the model to separately learn how to understand and generate SELFIES expressions.

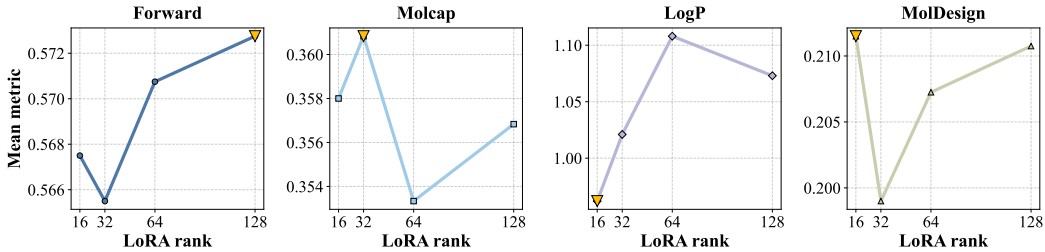

Figure 3: Evidence of varying intrinsic dimensions across task-specific representations. We observe that the optimal LoRA rank (indicated by the yellow triangle) differs across tasks.

We aim for the model to simultaneously learn general knowledge while also differentiating different modalities and tasks. Hence, we propose **Mixture-of-GAL-Experts (MoGE)**. We borrow the idea of MoE [15] and perform upcycling [36, 42] with the aforementioned GAL. We first construct $\mathcal{N}$ routed experts, each targeting specialized knowledge areas, and dynamically balance conflicting signals among these experts to effectively mitigate task-level conflicts. Besides, we introduce an additional shared expert to learn the common knowledge that underpins fundamental understanding across tasks, by consistently capturing and aligning shared features to maintain a stable global representation.

Specifically, we modify the Multi-Head Attention(MHA) and Feed Forward Network(FFN) layer as:

$$h'_l = h_{l-1} + \text{MHA}_{\phi'}(\text{LN}(h_{l-1}))$$

$$
\begin{aligned}
h'_l &= h_{l-1} + \text{MHA}_\phi(\text{LN}(h_{l-1})) \\
h_l &= h'_l + \text{FFN}_\rho(\text{LN}(h'_l))
\end{aligned}
\quad \rightarrow \quad
h_l =
\begin{cases}
h'_l + \text{FFN}_{\rho'}(\text{LN}(h'_l)), & l = 1 \dots l_{\text{MoGE}} \\
h'_l + \text{MoGE}_{\rho'_i, \psi}(\text{LN}(h'_l)), & l = l_{\text{MoGE}} \dots L
\end{cases}
\tag{3}
$$

where $h_l$ is the hidden states of the l'th layer, $\phi$, $\rho$ are the pre-trained parameters of the LLM, LN represents the norm layer of the LLM. We wrap the parameter $\rho' = \rho + \text{GAL}_\rho$, $\phi' = \phi + \text{GAL}_\phi$. For MoGE layer, we initialize $\mathcal{N} + 1$ experts with the weight of the pre-trained FFN $\rho$. Here, it concludes $\mathcal{N}$ routed experts to learn specialized knowledge and 1 shared experts to learn the common knowledge. Let $\rho_i$ be the parameter of the $i$-th expert, and at the beginning of the training, these experts have identical weights, *i.e.*, $\rho_1 = \rho_2 = \cdots = \rho$. Router $R_\psi$ is randomly initialized with Kaiming uniform [24], where $\psi$ is the learnable parameter of the router.

### 5.3 Training

The training strategy of Omni-Mol consists of two stages.

**Stage 1:** We perform multimodal alignment on PubChem [33], learning to describe molecules through graph modality features. The input consists of instructions and graph data, excluding SELFIES. Only the multimodal projector $f_p$ is trainable.

**Stage 2:** We fine-tune Omni-Mol by freezing the pre-trained parameters (wrapped by GAL), while the adapters, expert router, and the multimodal projector stay active throughout fine-tuning.

Training loss of both stages for language modeling is:

$$\mathcal{L}_{\text{LM}} = -\sum_i \log P_\theta(\mathbf{Y}_i | \mathbf{X}_I, \mathbf{X}_S, \mathbf{H}_G, \mathbf{Y}_{<i}) \tag{4}$$

For stage 2, we incorporate an additional auxiliary load balancing loss for the MoGE layers, assume an input tensor $x \in \mathbb{R}^{B \times T \times d}$, and $\mathcal{E}$ experts out of $\mathcal{N}$ is selected, the load balancing loss is: $\mathcal{L}_{\text{aux}} = \frac{1}{B} \sum_{i=1}^B \sum_{j=1}^{\mathcal{N}} C_{ij} \cdot \bar{s}_{ij}$, where $C_{ij} = \frac{\mathcal{N}}{T\mathcal{E}} \sum_{t=1}^{T\mathcal{E}} \mathbf{1}\{\text{t'th token selects expert j}\}$, $\bar{s}_{ij} = \frac{1}{T} \sum_{t=1}^T s_{i,j,t}$ and $\mathbf{1}\{\cdot\}$ is an indicator function. Here, $s_{i,j,t}$ is the router logit of the t'th token for j'th expert in batch i. This load balancing loss used in [43] additionally considers the sequence-level information.

The total loss is a combination of $\mathcal{L}_{\text{LM}}$ and $\mathcal{L}_{\text{aux}}$ with a coefficient $\lambda$: $\mathcal{L} = \mathcal{L}_{\text{LM}} + \lambda \mathcal{L}_{\text{aux}}$.

| Model | | #Par | Exa | BLEU | Lev | RDK | MAC | Mor | Val |
|---|---|---|---|---|---|---|---|---|---|
| **Forward Reaction Prediction Task** | | | | | | | | | |
| DeepSeekV3 [80] | ICL | 685B | 0.35 | 0.939 | 12.76 | 0.719 | 0.823 | 0.68 | 1.00 |
| Llama2 [65] | SL | 6.7B | 0.01 | 0.804 | 29.95 | 0.499 | 0.649 | 0.41 | 1.00 |
| Mol-Ins [21] | SL | 6.7B | 0.05 | 0.654 | 27.26 | 0.313 | 0.509 | 0.26 | 1.00 |
| HIGHT [13] | SL | 6.7B | 0.29 | 0.935 | 16.69 | 0.774 | 0.618 | 0.57 | 1.00 |
| InstructMol [10] | SL | 6.7B | 0.54 | 0.967 | 10.85 | 0.776 | 0.878 | 0.74 | 1.00 |
| PRESTO* [11] | GL | 3.2B | 0.69 | 0.976 | 6.53 | 0.871 | 0.931 | 0.84 | 1.00 |
| **Omni-Mol** | GL | 2.2B | **0.73** | **0.980** | **5.55** | **0.895** | **0.947** | **0.87** | 1.00 |

| Model | | #Par | Exa | BLEU | Lev | RDK | MAC | Mor | Val |
|---|---|---|---|---|---|---|---|---|---|
| **Retrosynthesis Task** | | | | | | | | | |
| DeepSeekV3 [80] | ICL | 685B | 0.29 | 0.93 | 14.32 | 0.725 | 0.827 | 0.68 | 1.00 |
| Llama2 [65] | SL | 6.7B | 0.00 | 0.283 | 53.51 | 0.136 | 0.294 | 0.11 | 1.00 |
| Mol-Ins [21] | SL | 6.7B | 0.01 | 0.705 | 31.23 | 0.283 | 0.487 | 0.23 | 1.00 |
| HIGHT [13] | SL | 6.7B | 0.20 | 0.914 | 20.20 | 0.772 | 0.623 | 0.58 | 0.99 |
| InstructMol [10] | SL | 6.7B | 0.41 | 0.941 | 13.97 | 0.753 | 0.852 | 0.71 | 1.00 |
| PRESTO* [11] | GL | 3.2B | 0.53 | 0.958 | 10.30 | 0.823 | 0.887 | 0.79 | 1.00 |
| **Omni-Mol** | GL | 2.2B | **0.57** | **0.960** | **8.97** | **0.864** | **0.909** | **0.83** | 1.00 |

| Model | | #Par | Exa | BLEU | Lev | RDK | MAC | Mor | Val |
|---|---|---|---|---|---|---|---|---|---|
| **Reagent Prediction Task** | | | | | | | | | |
| DeepSeekV3 [80] | ICL | 685B | 0.26 | 0.684 | 15.20 | 0.501 | 0.581 | 0.47 | 0.99 |
| Llama2 [65] | SL | 6.7B | 0.00 | 0.283 | 53.51 | 0.136 | 0.294 | 0.11 | 1.00 |
| Mol-Ins [21] | SL | 6.7B | 0.04 | 0.224 | 23.17 | 0.237 | 0.364 | 0.21 | 1.00 |
| HIGHT [13] | SL | 6.7B | 0.07 | 0.482 | 27.17 | 0.462 | 0.346 | 0.30 | 1.00 |
| InstructMol [10] | SL | 6.7B | 0.13 | 0.610 | 19.66 | 0.444 | 0.539 | 0.40 | 1.00 |
| PRESTO* [11] | GL | 3.2B | 0.21 | 0.712 | 16.31 | 0.544 | 0.607 | 0.48 | 1.00 |
| **Omni-Mol** | GL | 2.2B | **0.23** | **0.726** | **14.59** | **0.557** | **0.627** | **0.52** | 1.00 |

| Model | | #Par | Exa | BLEU | Lev | RDK | MAC | Mor | Val |
|---|---|---|---|---|---|---|---|---|---|
| **Catalyst Prediction** | | | | | | | | | |
| DeepSeekV3 [65] | ICL | 685B | 0.28 | 0.189 | 7.83 | 0.510 | 0.481 | 0.29 | 0.99 |
| Vicuna v1.5 [80] | SL | 6.7B | 0.69 | 0.703 | 2.45 | 0.883 | 0.869 | 0.69 | 1.00 |
| nach0-base | - | - | 0.00 | 0.07 | 36.44 | 0.129 | 0.055 | 0.01 | 0.85 |
| Mol-Ins [21] | SL | 6.7B | 0.00 | 0.110 | 28.42 | 0.031 | 0.045 | 0.02 | 0.99 |
| T5Chem | GL | - | 0.02 | 0.346 | 13.41 | 0.146 | 0.268 | 0.20 | 0.99 |
| PRESTO [11] | GL | 6.7B | **0.77** | **0.814** | **1.76** | **0.914** | **0.895** | **0.77** | 1.00 |
| **Omni-Mol** | GL | 2.2B | 0.72 | 0.792 | 1.96 | 0.904 | 0.886 | 0.72 | **1.00** |

| Model | Type | #Param | HOMO | LUMO | GAP | Avg. |
|---|---|---|---|---|---|---|
| **Quantum Mechanics Property Prediction Task** | | | | | | |
| DeepSeekV3 | ICL | 685B | 0.0200 | 0.0599 | 0.0457 | 0.0456 |
| LLaMA2 | ICL | 6.7B | 0.7367 | 0.8641 | 0.5152 | 0.7510 |
| Vicuna | ICL | 13B | 0.7135 | 3.6807 | 1.5407 | 1.9783 |
| Mol-Ins | SL | 6.7B | 0.0210 | 0.0210 | 0.0203 | 0.0210 |
| HIGHT | SL | 6.7B | 0.0056 | 0.0065 | 0.0077 | 0.0066 |
| InstructMol | SL | 6.7B | 0.0048 | 0.0050 | 0.0061 | 0.0050 |
| **Omni-Mol** | GL | 2.2B | **0.0038** | **0.0047** | **0.0049** | **0.0044** |

| Model | Type | #Param | B-2 | B-4 | R-1 | R-2 | R-L | M |
|---|---|---|---|---|---|---|---|---|
| **Molecular Captioning Task** | | | | | | | | |
| DeepSeekV3 | ICL | 685B | 0.181 | 0.095 | 0.319 | 0.133 | 0.249 | 0.231 |
| GPT-4-0314 | RT | - | **0.607** | **0.525** | **0.634** | **0.476** | **0.562** | **0.610** |
| BioMedGPT | GL | 10B | 0.234 | 0.141 | 0.386 | 0.206 | 0.332 | 0.308 |
| Mol-Ins | SL | 6.7B | 0.249 | 0.171 | 0.331 | 0.203 | 0.289 | 0.271 |
| HIGHT | SL | 6.7B | 0.498 | 0.397 | 0.582 | 0.414 | 0.518 | 0.525 |
| InstructMol | SL | 6.7B | 0.475 | 0.371 | 0.566 | 0.394 | 0.502 | 0.509 |
| **Omni-Mol** | GL | 2.2B | 0.529 | 0.440 | 0.604 | 0.447 | 0.541 | 0.571 |

| Model | | #Par | B-2 | B-4 | R-1 | R-2 | R-L | M |
|---|---|---|---|---|---|---|---|---|
| **Description Q&A Task** | | | | | | | | |
| DeepSeekV3 | ICL | 685B | 0.39 | 0.31 | 0.50 | 0.34 | 0.46 | 0.54 |
| Llama2 | SL | 6.7B | 0.28 | 0.23 | 0.35 | 0.22 | 0.30 | 0.47 |
| 3D-MoLM(S) | SL | 6.7B | 0.32 | 0.26 | 0.40 | 0.26 | 0.35 | 0.52 |
| 3D-MoLM(G) | GL | 6.7B | 0.32 | 0.26 | 0.40 | 0.26 | 0.35 | 0.52 |
| **Omni-Mol** | GL | 2.2B | **0.52** | **0.44** | **0.53** | **0.38** | **0.49** | **0.58** |

| Model | | #Par | B-H | S-M |
|---|---|---|---|---|
| **Yield Prediction** | | | | |
| DeepSeekV3 | ICL | 685B | -0.55 | -1.09 |
| Llama2 | SL | 6.7B | -0.48 | 0.12 |
| Vicuna v1.5 | SL | 6.7B | -0.13 | 0.15 |
| PRESTO | GL | 6.7B | 0.94 | 0.65 |
| **Omni-Mol** | GL | 2.2B | **0.94** | **0.68** |

| Model | | #Par | Weight | LogP | TPSA |
|---|---|---|---|---|---|
| **More Mol2Num** | | | | | |
| DeepSeekV3 | ICL | 685B | 98.63(100) | 2.26(100) | 60.32(100) |
| Llama2 | SL | 6.7B | 22.10 (96) | 1.45 (95) | 15.87 (92) |
| 3D-MoLM(S) | SL | 6.7B | 14.79 (95) | 0.66 (97) | 9.71 (93) |
| 3D-MoLM(G) | GL | 6.7B | 16.58 (92) | 0.78 (95) | 10.90 (90) |
| **Omni-Mol** | GL | 2.2B | **11.07(100)** | **0.49(100)** | **5.89(100)** |

| Model | | #Par | Exa | BLEU | Lev | RDK | MAC | Mor | Val |
|---|---|---|---|---|---|---|---|---|---|
| **Solvent Prediction** | | | | | | | | | |
| DeepSeekV3 | ICL | 685B | 0.06 | 0.471 | 5.08 | 0.196 | 0.248 | 0.15 | 0.99 |
| Vicuna v1.5 | SL | 6.7B | 0.32 | 0.436 | 3.81 | 0.459 | 0.486 | 0.43 | 1.00 |
| nach0-base | - | - | 0.00 | 0.072 | 36.44 | 0.129 | 0.055 | 0.01 | 0.85 |
| Mol-Instruction | SL | 6.7B | 0.00 | 0.155 | 25.12 | 0.030 | 0.122 | 0.04 | 1.00 |
| T5Chem | GL | - | 0.08 | 0.311 | 16.22 | 0.458 | 0.424 | 0.40 | 0.99 |
| PRESTO | GL | 6.7B | 0.42 | 0.695 | 2.76 | 0.529 | 0.547 | 0.51 | 0.91 |
| **Omni-Mol** | GL | 2.2B | **0.52** | **0.759** | **2.71** | **0.671** | **0.673** | **0.64** | 1.00 |

| Model | | #Par | B-2 | B-4 | R-1 | R-2 | R-L |
|---|---|---|---|---|---|---|---|
| **Experimental Procedure** | | | | | | | |
| DeepSeekV3 | ICL | 685B | - | - | - | - | - |
| TextChemT5 | GL | 220M | 0.541 | 0.406 | 0.615 | 0.403 | 0.564 |
| MolT5-Large | SL | 780M | 0.545 | 0.410 | 0.625 | 0.409 | 0.572 |
| Galactica | SL | 1.3B | 0.535 | 0.395 | 0.609 | 0.386 | 0.552 |
| MolCA, Galac | SL | 1.3B | 0.549 | 0.415 | 0.625 | 0.404 | 0.570 |
| ReactXT, Galac | SL | 1.3B | **0.574** | 0.440 | **0.644** | **0.427** | **0.589** |
| **Omni-Mol** | GL | 2.2B | _0.572_ | **0.448** | 0.532 | 0.274 | 0.464 |

| Model | | #Param | QED ≥ 0.6 | drd2 ≥ 0.5 | QED ≥ 0.6 drd2 ≥ 0.5 | Δ ≥ 0.4 QED ≥ 0.6 | Δ ≥ 0.4 drd2 ≥ 0.5 | Δ ≥ 0.4 QED ≥ 0.6 drd2 ≥ 0.5 |
|---|---|---|---|---|---|---|---|---|
| **Molecule Editing** | | | | | | | | |
| Llama3.2-1B* | SL | 1.2B | 0.8846 | 0.0507 | 0.0311 | 0.4441 | 0.0360 | 0.0271 |
| DeepSeekV3 | ICL | 685B | 0.8750 | 0.0000 | 0.0023 | 0.7500 | 0.0000 | 0.0017 |
| **Omni-Mol** | GL | 2.2B | **0.9612** | **0.0653** | **0.0412** | **0.7913** | **0.0560** | **0.0341** |

Table 3: Comprehensive comparisons on `Mol2Mol`, `Mol2Text` and `Mol2Num` tasks. Par: Parameters, Lev: Levenshtein, MAC: MACCS, Mor: Morgan, Val: Validity, Avg.: Average. B-2: BLEU-2, B-4: BLEU-4, R-1: ROUGE-1, R-2: ROUGE-2, R-L: ROUGE-L, M: METEOR, ICL: In-Context Learning, SL: Specialist, GL: Generalist, RT: Retrieval. * means our re-implementation. 3D-MoLM(S) is the specialist version and 3D-MoLM(G) is the generalist version.

# 6 Experiments

We aim to address the following concerns: (1) Compared with existing baselines, can Omni-Mol achieve the best performances on the comprehensive omni-molecular datasets with 16 tasks? (2) Is Omni-Mol a scalable framework with the capacity and potential to solve complex molecular tasks? (3) Are all key components of Omni-Mol essential for omni-molecular task learning? We begin by describing the experimental setup and then answer all the questions in the subsequent sections.

## 6.1 Experimental Setup

**Baselines.** To ensure a fair comparison, we first choose representative LLM-based models such as InstructMol and HIGHT, and also report several previous baselines, including Mol-Instruction,

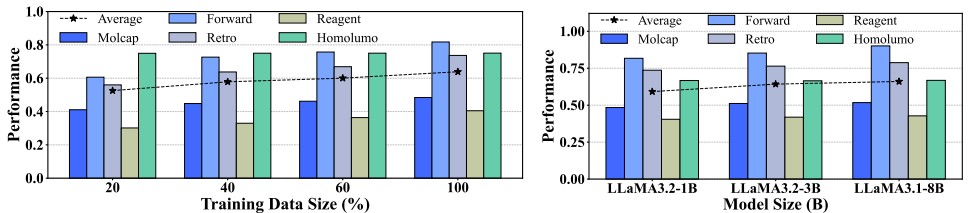

Figure 4: Scaling trend of Omni-Mol. (Left) The scaling trend with respect to dataset proportion, metrics are averaged across tasks. (Right) The scaling trend with respect to model size.

| Model | | #Par | Exa | BLEU | Lev | RDK | MAC | Mor | Val | Model | | #Param | Exa | BLEU | Lev | RDK | MAC | Mor | Val |
|---|---|---|---|---|---|---|---|---|---|---|---|---|---|---|---|---|---|---|---|
| IUPAC2SELFIES | | | | | | | | | | Text Guided Molecule Generation | | | | | | | | | |
| Llama3.2-1B* | SL | 1.2B | 0.31 | 0.947 | 16.63 | 0.639 | 0.818 | 0.60 | 0.995 | Llama3.2-1B* | SL | 1.2B | 0.04 | 0.789 | 28.11 | 0.447 | 0.629 | 0.329 | 0.941 |
| DeepSeek | ICL | 685B | 0.00 | 0.828 | 30.37 | 0.177 | 0.399 | 0.14 | 0.893 | DeepSeek | ICL | 685B | 0.02 | 0.658 | 35.27 | 0.217 | 0.398 | 0.170 | 0.608 |
| **Omni-Mol** | GL | 2.2B | **0.39** | **0.952** | **13.38** | **0.729** | **0.871** | **0.69** | **0.996** | **Omni-Mol** | GL | 2.2B | **0.12** | **0.824** | **23.59** | **0.562** | **0.721** | **0.442** | **0.963** |

Table 4: Results of Text2Mol tasks, * means we train the model with LoRA and the same multimodal configuration as Omni-Mol.

Llama, and Vicuna [80]. We also conduct 5 shot in-context learning test on powerful open-source models like DeepSeekV3. For datasets with fewer models, we re-implement PRESTO as a baseline.

**Backbone.** We utilize LLaMA 3.2-1B [18] as the backbone, a single linear layer as the projector, and MoleculeSTM [54] as the graph encoder. For MoGE expansion, we set $l_{\text{MoGE}} = 1/4L$ and the number of experts to 5, there are 2 routed experts and 1 shared expert in total. More details about model implementation can be found in Appendix C.

**Evaluation Metric.** Following [11], we evaluate `Mol2Num` tasks with MAE and $R^2$. For `Mol2Text` we adopt the standard NLP suite, which are BLEU-2, BLEU-4, ROUGE-2, ROUGE-L, and METEOR outlined in [10, 41]. Text2Mol and reaction-related `Mol2Mol` tasks, following [10, 78] are gauged with Exact Match, Levenshtein score, MACCS similarity, Morgan similarity, and RDK similarity, quantifying how 1D molecular strings encode functional and structural information. For the molecule editing task in `Mol2Mol`, we report both unconstrained and constrained success rates. The constrained variant requires (i) the optimized molecule to satisfy the property thresholds, which are QED (Quantitative Estimate of Drug-likeness) [5] $\geq 0.6$ and DRD2 score (probability of being an active DRD2 ligand) [55] $\geq 0.5$, and (ii) a 2D topological similarity to its precursor of at least $\Delta = 0.4$. QED is computed with RDKit. All of our evaluation metrics take into account both the linguistic quality and their biological relevance. Details are in Appendix B.

**Training Details.** We use PyTorch [56] with DeepSpeed ZeRO-2 [58]. For unified tuning, we train 15 epochs with GAL rank of 64. For separate tuning, model is trained for 10 epochs with the same GAL configuration. The learning rate is set to 8e-5 from the grid search for all experiments. For consistency, the random seed is set to 0. More details can be found in Appendix D.

## 6.2 Main Results

Here, we obtain the answer that *Omni-Mol can achieve the best performance across almost all tasks.* As the results shown in Table 3, we have the following observations. Omni-Mol significantly outperforms almost all specialist baselines while utilizing only 33% of the parameters. Furthermore, Omni-Mol surpasses the corresponding state-of-the-art generalist baseline by an average of approximately 5%, 7%, 9%, 11%, and 40% across forward prediction, retrosynthesis, reagent prediction, molcap, and Description Q&A separately. We further notice that on `Mol2Num` tasks, Omni-Mol improves the $R^2$ score by 21.4%, and lowers the Mean Absolute Error(MAE) by 25.1%, 25.8%, 39.3% on Weight, LogP and TPSA regression respectively. That is to say, Omni-Mol achieves superior performance with greater parameter efficiency, demonstrating its effectiveness in becoming a general AI chemist. To further examine GAL's adaptability to **3D** molecular tasks, we report Omni-Mol's results on ten tasks. Omni-Mol achieves the best performance on all ten tasks, outperforming the above baselines. Additional experimental details are provided in Appendix E.

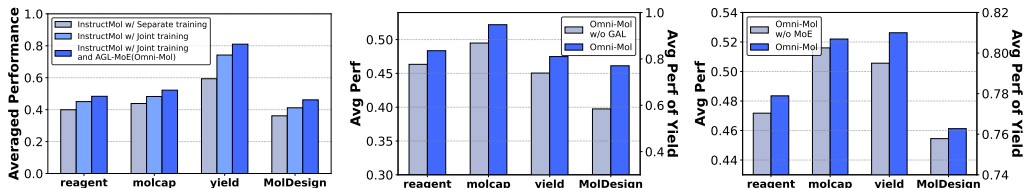

Figure 5: Ablation studies: (Left) Averaged performance comparison, the results demonstrate that InstructMol with joint training outperforms separate training. Further integrating MoGE yields additional performance gains across all tasks. (Middle) Ablation of Gradient Adaptive LoRA. (Right) Ablation of MoGE expansion. For the mid and right figure, the left axis represents the average performance of Reagent, Molcap and MolDesign. The right axis represents the average performance of Yield Prediction.

## 6.3 Is Omni-Mol a Scalable Framework?

One critical property of LLMs is their scaling behavior in relation to both model and data size. In this study, we demonstrate that *Omni-Mol* is a scalable framework by conducting two distinct types of scaling experiments: **(1)** We select three different sizes of LLaMA 3 series, 1B, 3B, and 8B, for language backbone scaling. More backbone results are in Appendix F. **(2)** We evaluate the impact of dataset size by down-sampling the original dataset to 20%, 40%, 60%, and 100% of its full size.

**(1) Data Scaling**. As shown in the left of Figure 4, we observe a clear scaling trend as the dataset proportion increases, indicating that the model's performance improves as the amount of data increases. This suggests that further increasing the dataset size can bring more benefits and build a stronger and more generalized chemical AI.

**(2) Parameter Scaling**. As shown on the right side of Figure 4, the performance of Omni-Mol across all tasks increases as the model size grows. We also observe a clear scaling trend. However, this trend is not as pronounced as the performance gains resulting from increased dataset size, which suggests that there remains potential for further expansion in the amount of data.

## 6.4 Is Unified Instruction Tuning Essential?

We evaluate whether our Omni-Mol dataset can enhance unified instruction tuning. We select one task from each of the following categories: `Mol2Mol`, `Mol2Text`, `Mol2Num`, and `Text2Mol`, and retrain InstructMol on each. We compare the results of individually trained models (one LoRA per task) with those trained in a unified manner (a single LoRA shared across all tasks), as shown in Figure 5. The results indicate that unified learning on the Omni-Mol dataset yields performance improvements. Interestingly, the molcap and yield tasks exhibit noticeable differences from the other tasks, yet still benefit from unified fine-tuning, with yield regression showing the greatest performance gain.

## 6.5 Ablations on MoGE

**How do MoGE help?** Building upon the unified training of InstructMol on our dataset, we incorporate MoGE into the framework. As illustrated in Figure 5, InstructMol with Joint Training and MoGE consistently outperforms both InstructMol with Separate Training and InstructMol with Joint Training, demonstrating the effectiveness of MoGE. By adapting to the intrinsic dimensionality of different tasks and leveraging the specialization among MoGE experts, MoGE enhances Omni-Mol's ability to generalize across a diverse range of tasks and modalities.

**Is GAL essential?** We compare our Omni-Mol with Omni-Mol w/o GAL, which replaces the Gradient Adaptive LoRA with the standard LoRA. As shown in the middle of Figure 5, Omni-Mol w/o GAL consistently exhibits lower performances than Omni-Mol. This consistent decline underscores the effectiveness of our GAL in enhancing performance by adaptively adjusting itself to the intrinsic dimension.

**Is MoGE expansion essential?** We conduct an ablation study by removing the MoGE expansion and training the model with GAL alone (Exclude MoE). The comparison results are shown on the right side of Figure 5. We observe that Omni-Mol consistently outperforms the Omni-Mol w/o

MoE across multiple tasks, including reagent prediction, molcap, yield regression, and text-guided molecule generation. These results demonstrate that Omni-Mol effectively enhances performance by leveraging specialized experts. The most significant improvement is observed in yield regression, where the diversity of experts contributes to better generalization and representation learning.

### 6.6 Convergence Analysis via Mutual Similarity

Omni-Mol is trained across a wide range of tasks, we aim to examine how its learned representations vary with respect to the number of tasks involved in training.

If the model is indeed learning within a progressively smaller solution space, we should observe a convergence in representation similarity. This is because, under a larger number of tasks, the reduction in the solution space is expected to constrain the variation in the learned representations. We compute the representation sequence with the model trained on 1, 2, 4, and 8 tasks with `mutual_knn` [27], the results are shown in Figure 6. Obviously, when the number of tasks increases, the similarity of the representations learned by Omni-Mol also increases. This indicates that the model's representations are gradually converging. This outcome supports the hypothesis that adding more tasks reduces the size of the model's general solution space. Ultimately, these representations converge to a universal form that can effectively solve all tasks. Interestingly, in the mutual similarity analysis of InstructMol, we observe the opposite trend. As the number of tasks increases, the representations learned by InstructMol become progressively less similar to those learned previously. This suggests that with each added task, the changes in the solutions learned by InstructMol become larger, indicating that it is unable to converge to a universal representation space through unified training.

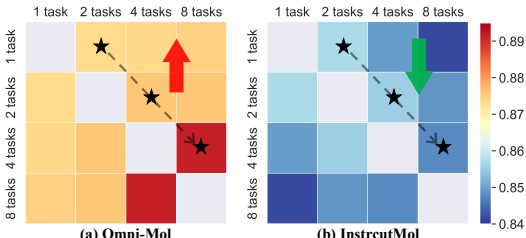

Figure 6: Demonstration of similarity scores heatmap for methods trained on varying numbers of tasks.

## 7 Conclusion

We introduce Omni-Mol, a model that unifies 16 tasks and Omni-Mol dataset, with over 1.4 million samples. Omni-Mol learns generalizable representations and achieves this through unified tuning, MoGE expansion, and Gradient Adaptive LoRA. Omni-Mol achieves SOTA performance across multiple tasks, and we demonstrate its scalability and ability to scale up performance as the number of tasks increases. Finally, we provide experimental evidence showing that Omni-Mol achieves a more general convergent solution space, acquiring the general capability to solve diverse tasks.

## Limitations

We identified two limitations: (1) Due to the limited computational resources, we are unable to further scale up the model with higher computational resources, which prevents us from exploring the limits of model's performance. (2) Omni-Mol's tasks are still primarily focused on small molecules, future work should explore proteins and the interaction between proteins and small molecules.

## Broader Impacts

This paper presents Omni-Mol, which is the first scalable and unified molecular generalist model with outperforming results, enabling tasks such as molecule captioning, property prediction, and drug design. While these advancements provide powerful tools for molecular research, they also raise ethical concerns, such as the risk of misuse in designing harmful molecules. Transparency, responsible use, and interdisciplinary collaboration are essential to ensure these models serve the broader good, paving the way for impactful and responsible scientific innovation.

## Acknowledgment

We thank the University of Maryland, College Park for providing us abundant computational resources. We sincerely thank Fanding Xu for his help in verifying the case study. His professional biochemical knowledge was invaluable to this process.

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

## Appendix

## Table of Content

## A   Experiment Environments

In this section, we provide a summary of our experiment environment.

**Software and Driver Versions**. The experiments are conducted with the following key software

- Python   3.12.1
- Pytorch   2.5.1
- Transformers   4.45.2
- CUDA   12.4

**Accelerators**. Training Omni-Mol costs $576 \times$ NVIDIA A100 80G GPU hours.

## B   Further details on datasets

### B.1   Comprehensive Datasets Construction

In this subsection, we provide a comprehensive list of the datasets used in our study along with their respective sources. While datasets vary across different papers, we observed that many are derived and processed from common sources. To clarify this overlap, we summarize the information in Table and provide a detailed analysis below.

**(1) USPTO** [66]. The USPTO (United States Patent and Trademark Office) dataset is a widely used large-scale chemical reaction dataset extracted and processed from US patent texts. It encompasses a diverse range of organic reaction types, including esterification, amidation, halogenation, Suzuki coupling, Buchwald–Hartwig coupling, addition reactions, condensation reactions, and redox reactions. Following [21], for the **Forward Reaction Prediction** task, we extract data from USPTO, and split the dataset into 124,384 training instances and 1,000 test instances. Partially following [11], for the **Catalyst Prediction** and **Solvent Prediction** tasks, we similarly extract data from USPTO, splitting the training/test sets into 10,079/1,015 and 67,099/7,793, respectively.

USPTO_500_MT [51] is a high-quality multi-task reaction prediction dataset, derived from USPTO through manual processing (including data filtering, deduplication, etc.). This subset retains the 500 most common reaction types. Following [21], for the **Reagent Prediction** task, we split the dataset into 124,384 training instances and 1,000 test instances.

USPTO_500K [51], a subset of organic chemical reaction data extracted from USPTO, is widely used in chemoinformatics for the single-step retrosynthesis task. Following [21], for the **Retrosynthesis** task, the dataset is divided into 128,684 training instances and 1,000 test instances.

USPTO-Applications [50] is another commonly used subset of USPTO, primarily derived from data samples in patent applications. For the **Experiment Procedure Prediction** task, following [48]

(along with the introduction of ORD data), we split the dataset into 80% training, 10% validation and 10% test sets.

**(2) ChEBI-20** [20]. ChEBI-20 is derived from the ChEBI-16 [22] dataset, with further annotations based on PubChem, forming a comprehensive database of chemical entities in the field of biochemistry. Compared to [21], ChEBI-20 provides a more extensive and detailed description of chemical compounds. Therefore, for the **Molecular Captioning** task, following [10], we split the ChEBI-20 dataset (which contains a total of 33,010 instances) into 26,420 training instances, 3295 validation instances and 3,295 test instances.

**(3) QM9** [70]. QM9 is a subset of the GDB-17 [59] database, focusing on quantum chemical property prediction for small organic molecules. It provides comprehensive quantum chemical attributes for molecular compounds, including spatial geometries and electronic properties, such as HOMO/LUMO energy levels obtained via DFT calculations [35]. In this work, we focus on the HOMO/LUMO energy levels of molecules. For the **Quantum Mechanics Property Prediction** task, following [21], we split the dataset into 360,113 training instances and 1,987 test instances.

**(4) PubChem** [32]. PubChem is the world's largest open-access chemical information database, focusing on chemistry, bioinformatics, and drug discovery. It provides comprehensive support for the retrieval and analysis of molecular compound data. Partiallly following [41], for the **Molecular Weight Prediction**, **LogP Prediction**, and **Topological Polar Surface Area Prediction** tasks, we split the dataset into 11,979/2,000, 10,673/1,785, and 11,979/2,000 for training and test sets, respectively. Additionally, for the **Description Q&A** task, also following [41], we split the dataset into 56,885 training instances and 10,000 test instances. For the **IUPAC2SELFIES** task, we split the training and test sets into 54,811 and 2,764 samples, respectively. For the **Text Guided Molecule Generation** task, we split 11,986/2,000 for training and test sets, respectively.

**(5) RNX Yields** [60]. The RNX Yields dataset consists of the Buchwald–Hartwig reaction [3] dataset and the Suzuki–Miyaura reaction [57] dataset, both collected through high-throughput experimentation (HTE). It is designed to predict reaction yields for these two reaction types. Following PRESTO, we split the dataset into 9,515 training instances and 200 test instances for **Yields Regression**.

**(6) ORD** [31]. The ORD (Open Reaction Database) is an open-source database dedicated to the standardization, storage, and sharing of organic chemistry reaction data, providing a unified data schema with structured text for organic reaction datasets. Following [48] (along with the USPTO-Applications), for the **Experimental Procedure Prediction** task, We partition the dataset into 90% for training, 10% for validation, and 10% for testing, based on the total data volume.

**(7) ZINC** [28]. ZINC ("ZINC Is Not Commercial") is an openly accessible repository of purchasable compounds engineered for structure-based virtual screening. It is committed to offering a no-cost, scalable platform for in silico screens. To ensure direct compatibility with leading docking engines, each entry in ZINC undergoes a specialized preprocessing workflow and is furnished with 3D conformers, the number of rotatable bonds, and other routine molecular descriptors. For the **Molecule Editing** task, we split the training and test sets into 218,708 and 3,579 samples, respectively.

Based on the seven datasets presented above, we construct a total of 16 tasks spanning four task types, amounting to 1.4 million data samples. To the best of our knowledge, this represents the most comprehensive dataset to date in the small molecular domain.

### B.2 Preprocessing

We encounter several issues during processing the datasets, we list them below and elaborate our solutions.

**Unable to obtain SELFIES.** We retrieve the SMILES representation of a molecule with its CID using pubchempy [63] API, for CIDs that cannot be found with `pubchempy.Compound.from_cid()`, we discard them. For molecules that cannot be converted to SELFIES, we discard them.

**Overlapped samples.** Datasets from different sources often contain overlapping samples, leading to potential data leakage. For example, solvent and catalyst prediction are subsets of reagent prediction, and molecule description data from PubChem [33] may include samples that overlap with those in ChEBI-20 [20]. Such overlaps create scenarios where a sample from one dataset's training set appears in the test set of another, compromising the reliability of model evaluation. To address this

issue, we conduct a thorough dataset comparison to identify potential overlaps and systematically remove any samples from the training sets that also appear in the test sets of other datasets.

**From SELFIES to 3D molecule.** Compared to the 1D and 2D representations of molecules, the 3D spatial topology reveals richer molecular properties and research value, such as protein interactions and molecular dynamics. Following [41], we preprocess molecular SELFIES using RDKit. First, we convert SELFIES to a 2D representation via the selfies library and RDKit and add hydrogen atoms to facilitate subsequent force field optimization. Finally, after embedding each atom at a random initial coordinate, we optimize the resulting conformations using RDKit's MMFF94 force field (e.g., by computing interatomic interaction potentials). Once we obtain the optimized atomic conformations, we execute 3D molecular tasks using the full 3D information.

## B.3   Details on Evaluation Metrics

**Exact Match.** The Exact Match Score evaluates whether two SMILES strings unequivocally correspond to the same molecular structure. Specifically, a score of 1 is assigned when both SMILES strings are identical following normalization, indicating they represent the same molecule. Meanwhile, a score of 0 is given when the normalized SMILES strings differ, signifying that they correspond to distinct molecules.

**Levenshtein Score.** The Levenshtein Score scores the smallest number of edit operations needed to transform one SMILES string into another. These edit operations typically encompass: (1) Insertion, which involves adding a character at a specific position; (2) Deletion, the removal of a character from a designated location; and (3) Substitution, replacing a character at a particular position with a different one.

**MACCS Similarity.** Within cheminformatics, MACCS Similarity is used to assess and compare the structural likeness of molecules. This approach is grounded in MACCS keys, which are a standardized set of structural descriptors developed by the Molecular ACCess System. These keys capture and represent essential molecular substructures. To determine the similarity between two molecules, the method evaluates the presence or absence of these predefined structural features.

**RDK Similarity.** The RDK Similarity generally involves evaluating and quantifying the similarity between molecules by utilizing fingerprints produced with RDKit.

**Morgan Similarity.** Morgan Similarity is used to evaluate and measure the structural resemblance between molecules by utilizing Morgan fingerprints as its foundational basis.

**Mean Absolute Error (MAE)**. The MAE quantifies the average absolute deviations between predicted results and actual values, which provides a straightforward metric for assessing the accuracy of predictive models by averaging the absolute differences across all instances.

$\mathbf{R}^2$. The $R^2$ metric scores the proportion of variability in the target variable that can be explained by the model's predictors. It can serve as an indicator of the model's explanatory strength, reflecting how well the observed data points are captured by the regression model.

**Unconstrained & Constrained Successful Rate (QED & DRD2)**. Both unconstrained and constrained success rates calculate the proportion of predicted molecules that satisfy a predefined threshold out of all predicted molecules. The unconstrained variant emphasizes the model's exploration of a broader chemical space to produce high-quality molecules that meet the specified properties, such as new scaffolds or substituents with improved characteristics. The constrained variant considers molecular similarity and focuses on preserving key molecular features (e.g., scaffolds, pharmacophores). QED and similarity are readily computed by RDKit. However, the commonly used approach for computing the DRD2 score relies on machine learning methods (e.g., support vector machines) for molecular classification [71, 55]. Although research shows that non–deep learning methods generally outperform deep learning approaches on molecular classification tasks [72], this finding may not fully account for the impact of deeper-level sample distribution factors, such as class imbalance and disparities in feature distributions on dataset size. Moreover, the generalization capability of non–deep learning methods can be limited, which undermines their suitability for DRD2 score computation, since molecules predicted by large language models may be entirely unseen by the DRD2 scoring model. Following [72], we adopt XGBoost [12] as the benchmark representative of machine learning approaches and train GNN-based model GraphMVP [46], on the molecule editing dataset for comparison experimental results and setup details appear in Appendix E. The results

indicate that XGBoost easily overfits to the training data, whereas GraphMVP exhibits significantly superior generalization compared to XGBoost, thereby validating the reliability of our proposed scoring approach.

## C   Further details on model implementation

### C.1   Graph Tokenizer

**Molecule to graph conversion.** GNN is widely used in many scenarios, such as traffic [75], social relationships [76], and also molecules [62, 61]. Following the vanilla setting, we utilize RDKit [38] to transform SELFIES into graph structures in our experiments. For tasks involving a single molecule as input, the molecule is converted directly. For tasks requiring multiple molecules as input, only the first molecule in the input sequence is converted into a graph. Our model does not incorporate multi-graph understanding; instead, it processes both the graph and SELFIES representation of the first molecule, while only the SELFIES representations are provided for the remaining molecules. Meanwhile, since MoleculeSTM [45] incorporates additional molecular graph-text contrastive training compared to GraphMVP [46], which leads to improved multimodal model training efficiency, we adopt MoleculeSTM as the graph encoder.

**Insertion**.  For graph tokens $\mathbf{H}_G = \{H_1, H_2, \ldots, H_n\}$ after projection, we always insert the graph token at the beginning of user instruction $\mathbf{X_I}$. The input instruction will be updated to the concatenation of $\{\mathbf{H}_G, \mathbf{X}_I\}$.

**Multiple molecule inputs.** In some tasks, (*e.g.,* Reagent Prediction), when we need to copy with multiple molecules as the input, we stack them into batch dimensions and feed them to the graph encoder together, the resulting graph features are correspondingly concatenated together.

### C.2   Multimodal Alignment

To balance the molecular graph and text modalities while ensuring training efficiency, we employ a single-layer linear projector in Stage 1. Following [45], we carefully filter PubChem to obtain 310K+ graph-text pairs and convert them into instruction-following data for pretraining. The alignment between the molecular graph and text modalities is enhanced solely by adjusting the parameters of the single-layer linear projector. After that, in the unified instruction tuning stage, we keep the projector active, allowing the projector to adapt to multiple tasks.

### C.3   Gradient Adaptive LoRA (GAL)

The scaling factor is calculated as

$$\gamma_\theta = \frac{\alpha}{r^p} + \beta \tag{5}$$

where $\theta = \{\alpha, p, \beta\}$

Inspired by [30, 53, 9, 23], we initialize these parameters as $\alpha_0 = 16, p_0 = 0.5, \beta_0 = 0$. Additionally, we clip the range of these learnable parameters.

$$\alpha = \mathrm{clip}(\alpha, \alpha_0 - \epsilon, \alpha_0 + \epsilon) \quad p = \mathrm{clip}(p, p_0 - \delta, p_0 + \delta) \quad \beta = \mathrm{clip}(\beta, \beta_0 - \epsilon, \beta_0 + \epsilon) \tag{6}$$

and we set $\epsilon = 0.05$, $\delta = 0.01$. If the rank is set to 64, then $\gamma_\theta \in [1.863, 2.141]$.

### C.4   Mutual Representation Similarity

**Task scaling setup.** We build a sequence of multi-task datasets with detailed composition as follows:

- **1 task**: Reagent Prediction.
- **2 tasks**: Reagent Prediction + Molecular Captioning.
- **4 tasks**: Reagent Prediction + Molecular Captioning + Solvent Prediction + Catalyst Prediction.
- **8 tasks**: Reagent Prediction + Molecular Captioning + Solvent Prediction + Catalyst Prediction + Forward Prediction + Retrosynthesis + Property Prediction + Yield Regression.

**Similarity calculation.** We first extract features $R \in \mathbb{R}^{B \times L \times T \times d}$ from all decoder layers in LLM, where $B, L, T, d$ is batch size, number of decoder layers, sequence length and the hidden dimension

|  | Learning rate | Num Epoch | LR Decay | Stop Epoch | Batch Size | Warmup Ratio |
|---|---|---|---|---|---|---|
| Forward Reaction Prediction | | | | | | |
| Reagent Prediction | | | | | | |
| Retrosynthesis | | | | | | |
| Quantum Mechanics Property Prediction | | | | | | |
| Catalyst Prediction | | | | | | |
| Solvent Prediction | | | | | | |
| Yield Regression | | 15 | | 10 | | |
| Experimental Procedure Prediction | 8e-5 | | cosine | | 128 | 0.0075 |
| Description Q&A | | | | | | |
| Topological Polar Surface Area Prediction | | | | | | |
| Molecular Weight Prediction | | | | | | |
| Text Guided Molecule Generation | | | | | | |
| Molecule Editing | | | | | | |
| IUPAC2SELFIES | | | | | | |
| LogP Prediction | | | | | | |
| Molecular Captioning | | 10 | | 8 | | |
| Omni-Molecular Tasks (2D & 3D) | | 15 | | 15 | | |

Table 5: An overview of the hyper-parameters and training configurations used in all molecular task experiments.

of LLM. The sequence dimension is then averaged.

$$R' = \frac{\left(\sum_{t=1}^{T}(R[:,:,t,:] * m[:,t])\right)}{\sum_{t=1}^{T} m[:,t]} \tag{7}$$

where $R' \in \mathbb{R}^{B \times L \times d}$, and $m \in \mathbb{R}^{B \times T}$ is the mask indicating the padding tokens. We then flatten the first two dimensions and get $R'' \in \mathbb{R}^{(B*L) \times d}$ and calculate the similarity with `mutual_knn` [27].

Let $N = B * L$, and we have two models $A$ and $B$ trained on different multi-task datasets, we first find their $k$ nearest neighbors $\text{knn}^A$ and $\text{knn}^B$.

$$\text{knn}^A = \text{KNN}(R^A, k) \quad \text{knn}^B = \text{KNN}(R^B, k) \tag{8}$$

where $\text{knn}^* \in \mathbb{R}^{N \times k}$, we then create indicator matrices

$$M_{i,j}^A = \begin{cases} 1, & j \in \text{knn}^A[i,:] \\ 0, & \text{otherwise} \end{cases} \qquad M_{i,j}^B = \begin{cases} 1, & j \in \text{knn}^B[i,:] \\ 0, & \text{otherwise} \end{cases} \qquad i,j \in 1,\dots,N \tag{9}$$

The accuracy of a sample is

$$\text{acc}[i] = \frac{1}{k}\left|\text{knn}^A[i,:] \cap \text{knn}^B[i,:]\right| = \frac{1}{k}\sum_{j=1}^{N} M_{i,j}^A \cdot M_{i,j}^B \tag{10}$$

Finally, the alignment score of two models is

$$\text{Score} = \frac{1}{N}\sum_{i=1}^{N} \text{acc}[i] \tag{11}$$

## D   Further details on training

Specialist models are typically fine-tuned on a single task at a time, repeating the process separately for each task, a strategy known as separate tuning. In contrast, generalist models undergo simultaneous fine-tuning across multiple tasks, a process referred to as unified tuning. In this section, we present a detailed training framework for both of them in all experiments.

**Separate instruction tuning.** We follow the training recipe outlined in [10]. However, we observe significant overfitting when training the model on the molcap task for 20–50 epochs, as suggested in [10]. To address this issue, we manually allocate 10% of the training set for validation and re-evaluated all tasks, we find that the recipes for forward prediction, reagent prediction, retrosynthesis,

and Quantum Mechanics Property Prediction from the original paper match our results, however, we identify an updated training strategy tailored to the molcap task. The revised training recipe is summarized in Table 5.

**Unified instruction tuning.** For unified training, we apply a fixed training recipe as shown in Table 5, this recipe is consistent across all Unified Instruction Tuning.

For all experiments, the weight decay is set to 0. The term *Stop Epoch* in Table 5 shows the epoch that the experiment stops. This is because of the early stop mechanism we used to prevent overfitting.

# E    Further details on experimental results

## E.1    Baslines

**In-Context Learning with DeepSeekV3**. To inspect the capability of In-Context Learning with a powerful open-source model like DeepSeekV3, we randomly sample 5 examples from training set and feed the model with instruction, SELFIES representation and the answer. The model is asked to solve the task from test set with these in-context examples. The prompt template is as follows:

> **Instruction**: Given the following examples: {Question: `question` Answer: `answer`}*5. Answer the following question, Question: `question`.

here, `question` and `answer` are the specific samples from the dataset, *5 means we provide 5 question-answer pairs.

**Molecular LLMs**. Mol-Instruction, InstructMol, HIGHT, 3D-MoLM, and PRESTO represent a series of works leveraging large language models (LLMs) to perform molecular tasks. Among them, Mol-Instruction, InstructMol, and HIGHT are specialist models that employ different adapters tailored to specific tasks. 3D-MoLM offers both generalist and specialist variants, and we report the performance of both in Table 3. PRESTO adopts a full fine-tuning strategy using a single model. For datasets and tasks overlapping with ours, we directly report the results from the original paper. For tasks that PRESTO was not designed to handle, we conducted our own re-implementation.

**Older Baselines**. We include some older baselines such as Llama2 [65](result from [21]), nach0 [49](result from [11]), T5Chem [51](result from [48]), TextChemT5 [14](result from [48]), MolT5 [19](result from [48]), Galatica [64](result from [48]), MolCA [47](result from [48]), GPT-4-0314 [1](result from [10]), BioMedGPT [52](result from [10]).

## E.2    3D Adaptive Ability of Omni-Mol

| Model | | #Par | Exa | BLEU | Lev | RDK | MAC | Mor | Val |
|---|---|---|---|---|---|---|---|---|---|
| Forward Reaction Prediction Task | | | | | | | | | |
| Omni-Mol | GL | 2.2B | 0.73 | 0.980 | 5.55 | 0.895 | 0.947 | 0.87 | 1.00 |
| Omni-Mol (3D) | GL | 2.2B | 0.75 | 0.983 | 5.56 | 0.893 | 0.944 | 0.88 | 1.00 |

| Model | | #Par | Exa | BLEU | Lev | RDK | MAC | Mor | Val |
|---|---|---|---|---|---|---|---|---|---|
| Retrosynthesis Task | | | | | | | | | |
| Omni-Mol | GL | 2.2B | 0.57 | 0.960 | 8.97 | 0.864 | 0.909 | 0.83 | 1.00 |
| Omni-Mol (3D) | GL | 2.2B | 0.57 | 0.958 | 9.24 | 0.858 | 0.910 | 0.82 | 1.00 |

| Model | | #Par | Exa | BLEU | Lev | RDK | MAC | Mor | Val |
|---|---|---|---|---|---|---|---|---|---|
| Reagent Prediction Task | | | | | | | | | |
| Omni-Mol | GL | 2.2B | 0.23 | 0.726 | 14.59 | 0.557 | 0.627 | 0.52 | 1.00 |
| Omni-Mol (3D) | GL | 2.2B | 0.25 | 0.706 | 14.35 | 0.603 | 0.663 | 0.56 | 1.00 |

| Model | | #Par | Exa | BLEU | Lev | RDK | MAC | Mor | Val |
|---|---|---|---|---|---|---|---|---|---|
| Catalyst Prediction | | | | | | | | | |
| Omni-Mol | GL | 2.2B | 0.73 | 0.980 | 5.55 | 0.895 | 0.947 | 0.87 | 1.00 |
| Omni-Mol (3D) | GL | 2.2B | 0.77 | 0.836 | 1.55 | 0.913 | 0.898 | 0.78 | 1.00 |

| Model | Type | #Param | HOMO | LUMO | GAP | Avg. |
|---|---|---|---|---|---|---|
| Quantum Mechanics Property Prediction Task | | | | | | |
| Omni-Mol | GL | 2.2B | 0.0038 | 0.0047 | 0.0049 | 0.0044 |
| Omni-Mol (3D) | GL | 2.2B | 0.0032 | 0.0036 | 0.0044 | 0.0037 |

| Model | Type | #Param | B-2 | B-4 | R-1 | R-2 | R-L | M |
|---|---|---|---|---|---|---|---|---|
| Molecular Captioning Task | | | | | | | | |
| Omni-Mol | GL | 2.2B | 0.529 | 0.440 | 0.604 | 0.447 | 0.541 | 0.571 |
| Omni-Mol (3D) | GL | 2.2B | 0.511 | 0.421 | 0.593 | 0.435 | 0.531 | 0.553 |

| Model | | #Par | B-2 | B-4 | R-1 | R-2 | R-L | M |
|---|---|---|---|---|---|---|---|---|
| Description Q&A Task | | | | | | | | |
| Omni-Mol | GL | 2.2B | 0.52 | 0.44 | 0.53 | 0.38 | 0.49 | 0.58 |
| Omni-Mol (3D) | GL | 2.2B | 0.51 | 0.44 | 0.53 | 0.39 | 0.49 | 0.58 |

| Model | | #Par | Weight | LogP |
|---|---|---|---|---|
| More `Mol2Num` | | | | |
| Omni-Mol | GL | 2.2B | 11.07(100) | 0.49(100) |
| Omni-Mol (3D) | GL | 2.2B | 10.98(100) | 0.48(100) |

| Model | | #Param | Exact | BLEU | Levenshtein | RDK | MACCS | Morgan | Validity |
|---|---|---|---|---|---|---|---|---|---|
| Solvent Prediction | | | | | | | | | |
| Omni-Mol | GL | 2.2B | 0.52 | 0.759 | 2.71 | 0.671 | 0.673 | 0.64 | 1.00 |
| Omni-Mol (3D) | GL | 2.2B | 0.56 | 0.77 | 2.44 | 0.70 | 0.70 | 0.68 | 1.00 |

Table 6: Additional results of Omni-Mol with 3D GNN.

| Model | Acc (Training) | Acc (Test) | ROC-AUC (Training) | ROC-AUC (Test) |
|---|---|---|---|---|
| Molecule Editing | | | | |
| XGBoost | 0.9995 | 0.9284 | 0.9976 | 0.7933 |
| GraphMVP | 0.8919 | 0.8748 | 0.9482 | 0.9188 |

Table 7: Comparison between XGBoost and GraphMVP on DRD2 score.

| Model | | HOMO | LUMO | GAP | Weight | LogP | TPSA |
|---|---|---|---|---|---|---|---|
| Mol2Num | | | | | | | |
| GraphMVP | SL | 0.0062 | 0.0082 | 0.0103 | 124.08 | 0.8257 | 42.20 |
| GraphMVP-G | SL | 0.3349 | 0.0076 | 0.0104 | 122.88 | 0.8099 | 42.33 |
| GraphMVP-C | SL | 0.0061 | 0.0076 | 0.0104 | 123.57 | 0.7806 | 41.79 |
| **Omni-Mol** | GL | 0.0038 | 0.0047 | 0.0049 | 11.07 | 0.49 | 5.89 |

Table 8: Comparison between Omni-Mol and GraphMVP

In this section, we answer the following question: *Is it possible to develop a generalist **3D molecular** LLM capable of effectively learning across diverse task domains?* 3D molecules represent the physical form encountered in the real world, revealing richer layers of molecular information. Hence, enabling LLMs to achieve strong performance on 3D tasks is critically important. To more effectively encode 3D molecular graph data, we employ Uni-Mol [81] as the graph encoder, integrating it into our graph tokenizer. To obtain the 3D molecular graph information for the 3D Omni-Mol data, we follow the preprocessing pipeline outlined in Appendix B. As the results shown in Table 6, Omni-Mol achieves state-of-the-art performance on all ten 3D tasks, demonstrating its strong adaptability to tasks that more closely reflect real-world scenarios and its potential significance for practical applications. For 3D unified training, we follow the hyperparameters and training strategies of the 2D Omni-Mol unified training. More details are provided in Appendix D.

### E.3 Comparison on `Mol2Num` Tasks with GNN

In this section, we re-implement GraphMVP as our baseline, we re-train GraphMVP on quantum mechanics property prediction, LogP prediction, molecular weight prediction, TPSA prediction tasks.

As shown in Table 8, Omni-Mol performs significantly better than traditional GNN models like GraphMVP. Omni-Mol provides up to $91\%$ improvement on tasks like molecular weight prediction. GraphMVP-G introduces generative 2D self-supervised learning, i.e., it trains the GNN to reconstruct the attributes of nodes and edges that are randomly masked. GraphMVP-C introduces contrastive 2D self-supervised learning, i.e., it trains the GNN to discriminate between constructed positive and negative molecular-graph sample pairs.

### E.4 Design Computational Tool for DRD2 Score

Since existing methods for computing the DRD2 score do not necessarily generalize to data with different distributions [55, 71], in this session we design a new tool for computing the DRD2 score and demonstrate the superiority of the proposed approach through our experiments. For the Molecule Editing task dataset, we extract unique molecules and the DRD2 score labels provided by [71]. To ensure that the DRD2 score computation method generalizes across different distributions, we enforce minimal similarity among molecules in the training, validation, and test sets, and maintain balanced active and inactive samples with the same threshold of activity in [55]. Following [55], we employ the Butina clustering algorithm [8], yielding 50,552, 5,159, and 6,169 samples in the training, validation, and test sets, respectively. As stated in Appendix B, we choose XGBoost and GraphMVP as representative non–deep-learning and deep-learning models, respectively. Adhering to the training setup of [55], the performance is reported in Table 7. These results demonstrate that GraphMVP attains superior generalization on the Molecule Editing data, which underpins the reliability of DRD2 score evaluation for molecule editing tasks, since the generated molecules are unlikely to have been encountered by the evaluation model during training.

| Model | Exact | BLEU | Levenshtein | RDK | MACCS | Morgan | Validity |
|---|---|---|---|---|---|---|---|
| Forward Reaction Prediction Task | | | | | | | |
| Head tuning | 0.56 | 0.927 | 9.98 | 0.791 | 0.888 | 0.72 | 1.00 |
| Omni-Mol | 0.73 | 0.980 | 5.55 | 0.895 | 0.947 | 0.87 | 1.00 |

| Model | Exact | BLEU | Levenshtein | RDK | MACCS | Morgan | Validity |
|---|---|---|---|---|---|---|---|
| Retrosynthesis Task | | | | | | | |
| Head tuning | 0.45 | 0.942 | 12.82 | 0.79 | 0.865 | 0.75 | 1.00 |
| Omni-Mol | 0.57 | 0.960 | 8.97 | 0.864 | 0.909 | 0.83 | 1.00 |

| Model | Exact | BLEU | Levenshtein | RDK | MACCS | Morgan | Validity |
|---|---|---|---|---|---|---|---|
| Reagent Prediction Task | | | | | | | |
| Head tuning | 0.14 | 0.662 | 17.74 | 0.459 | 0.556 | 0.42 | 1.00 |
| Omni-Mol | 0.23 | 0.726 | 14.59 | 0.557 | 0.627 | 0.52 | 1.00 |

| Model | Exact | BLEU | Levenshtein | RDK | MACCS | Morgan | Validity |
|---|---|---|---|---|---|---|---|
| Catalyst Prediction | | | | | | | |
| Head tuning | 0.71 | 0.727 | 2.56 | 0.888 | 0.872 | 0.72 | 1.00 |
| Omni-Mol | 0.73 | 0.980 | 5.55 | 0.895 | 0.947 | 0.87 | 1.00 |

| Model | BLEU-2 | BLEU-4 | ROUGE-1 | ROUGE-2 | ROUGE-L | METEOR |
|---|---|---|---|---|---|---|
| Molecular Captioning Task | | | | | | |
| Head tuning | 0.450 | 0.338 | 0.546 | 0.359 | 0.477 | 0.482 |
| Omni-Mol | 0.529 | 0.440 | 0.604 | 0.447 | 0.541 | 0.571 |

| Model | Exact | BLEU | Levenshtein | RDK | MACCS | Morgan | Validity |
|---|---|---|---|---|---|---|---|
| Solvent Prediction | | | | | | | |
| Head tuning | 0.39 | 0.68 | 3.34 | 0.530 | 0.547 | 0.50 | 1.00 |
| Omni-Mol | 0.52 | 0.759 | 2.71 | 0.671 | 0.673 | 0.64 | 1.00 |

Table 9: Additional ablation results on 6 tasks. We freeze the LLM decoder and only activate the language model head.

# F  More ablation study results

## F.1  Ablation on Language Backbone

To ensure that the performance gain doesn't come from newer language backbones like Llama3.1 and Llama3.2, we replace the language model in Omni-Mol to **Vicuna 7B** [80] and conduct experience on several tasks. As shown in Table 11, with larger parameter size, Vicuna performs significantly better than Llama3.

## F.2  Ablation on Parameter Efficient Tuning

One alternative tuning method is to tune **only the language model head** while keeping all parameters in the LLM decoder layers frozen. Note that for Llama 3.2 1B, the weights of the language model head are tied with the word embeddings; therefore, tuning the language model head also updates the word embeddings. The results are shown in Table 9, where head tuning performs significantly worse than Omni-Mol.

## F.3  Ablation on Clip in GAL

Clipping is employed to prevent the model from converging to local optima during training. To validate its effectiveness, we remove the clipping mechanism in this experiment. As shown in Table 10, removing clip generally results in performance drop.

| Method | Exact | BLEU | RDK | MACCS | Morgan | Levenshtein | Validity |
|---|---|---|---|---|---|---|---|
| Catalyst Prediction | | | | | | | |
| Omni-Mol | 0.742 | 0.794 | 0.911 | 0.899 | 0.747 | 1.843 | 1.000 |
| w/o clip | 0.731 | 0.775 | 0.898 | 0.878 | 0.737 | 2.144 | 1.000 |
| Forward Prediction | | | | | | | |
| Omni-Mol | 0.738 | 0.985 | 0.884 | 0.941 | 0.865 | 6.06 | 1.000 |
| w/o clip | 0.703 | 0.980 | 0.865 | 0.927 | 0.840 | 7.045 | 1.000 |
| Reagent Prediction | | | | | | | |
| Omni-Mol | 0.266 | 0.749 | 0.586 | 0.651 | 0.542 | 14.026 | 1.000 |
| w/o clip | 0.256 | 0.736 | 0.577 | 0.645 | 0.533 | 14.619 | 1.000 |
| Retrosynthesis | | | | | | | |
| Omni-Mol | 0.594 | 0.962 | 0.861 | 0.910 | 0.828 | 8.386 | 1.000 |
| w/o clip | 0.542 | 0.959 | 0.838 | 0.897 | 0.802 | 9.937 | 1.000 |
| Solvent | | | | | | | |
| Omni-Mol | 0.554 | 0.782 | 0.703 | 0.700 | 0.678 | 2.498 | 1.000 |
| w/o clip | 0.550 | 0.775 | 0.692 | 0.693 | 0.667 | 2.550 | 1.000 |

| Method | HOMO | LUMO | GAP | AVG | Method | B-2 | B-4 | M | R-1 | R-2 | R-L | Method | BH | SM |
|---|---|---|---|---|---|---|---|---|---|---|---|---|---|---|
| Quantum Mechanics | | | | | Molcap | | | | | | | Yield | | |
| Omni-Mol | 0.0038 | 0.0047 | 0.0049 | 0.0044 | Omni-Mol | 0.511 | 0.421 | 0.556 | 0.593 | 0.434 | 0.530 | Omni-Mol | 0.953 | 0.688 |
| w/o clip | 0.0039 | 0.0055 | 0.0047 | 0.0047 | w/o clip | 0.483 | 0.391 | 0.526 | 0.570 | 0.406 | 0.507 | w/o clip | 0.933 | 0.680 |

Table 10: Ablation on clip in GAL

| Model | | #Par | Exa | BLEU | Lev | RDK | MAC | Mor | Val | Model | | #Par | Exa | BLEU | Lev | RDK | MAC | Mor | Val |
|---|---|---|---|---|---|---|---|---|---|---|---|---|---|---|---|---|---|---|---|
| Forward Reaction Prediction Task | | | | | | | | | | Solvent Prediction | | | | | | | | | |
| Vicuna 7B [80] | SL | 6.7B | 0.29 | 0.94 | 16.86 | 0.768 | 0.610 | 0.557 | 1.00 | Vicuna 7B [80] | SL | 6.7B | 0.29 | 0.62 | 4.071 | 0.454 | 0.430 | 0.392 | 1.00 |
| Llama3 1B [18] | SL | 1.3B | 0.28 | 0.92 | 17.03 | 0.759 | 0.603 | 0.555 | 1.00 | Llama3 1B [18] | SL | 1.3B | 0.28 | 0.60 | 4.214 | 0.453 | 0.425 | 0.388 | 1.00 |

| Model | Type | #Param | BLEU-2 | BLEU-4 | ROUGE-1 | ROUGE-2 | ROUGE-L | METEOR | Model | | #Par | LogP |
|---|---|---|---|---|---|---|---|---|---|---|---|---|
| Molecular Captioning Task | | | | | | | | | More Mol2Num | | | |
| Vicuna 7B [80] | SL | 6.7B | 0.483 | 0.390 | 0.559 | 0.386 | 0.493 | 0.522 | Vicuna 7B [80] | SL | 6.7B | 0.5182 |
| Llama3 1B [18] | SL | 1.3B | 0.455 | 0.361 | 0.549 | 0.381 | 0.487 | 0.503 | Llama3 1B [18] | SL | 1.3B | 0.6690 |

Table 11: Comparison with different language backbone. We changed the backbone to Vicuna 7B and trained on several tasks, the results clearly indicate that better performance can be obtained from larger language model.

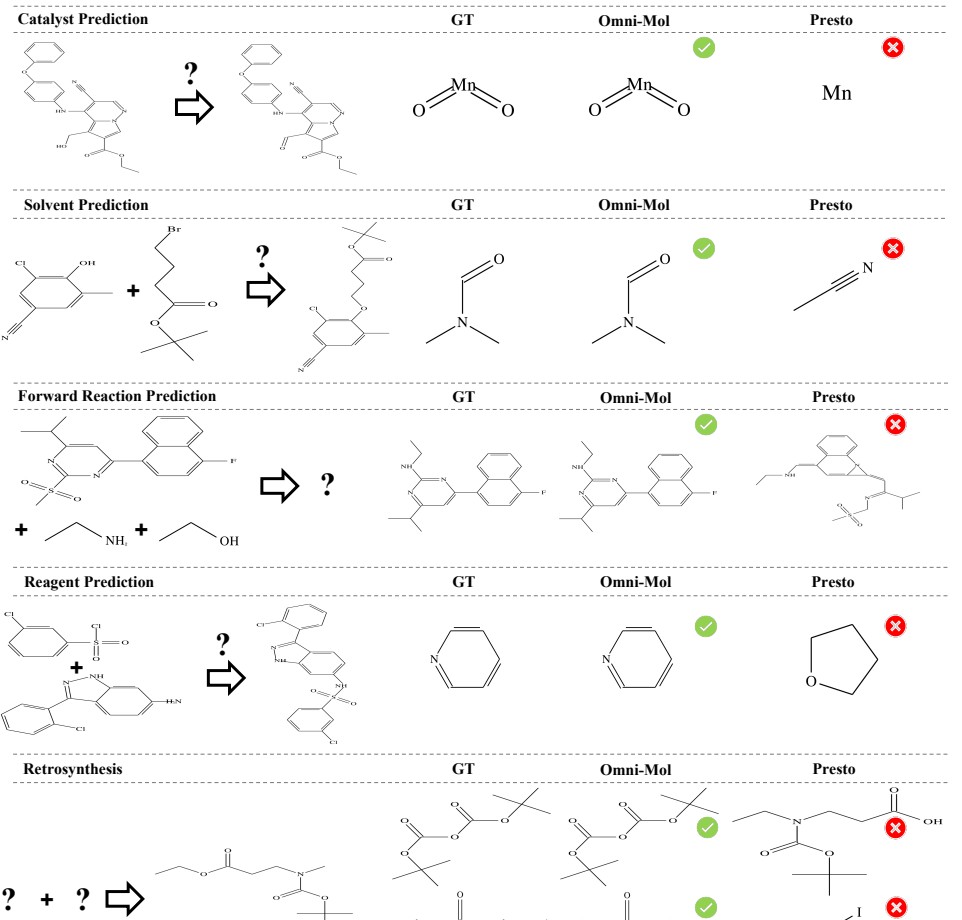

Figure 7: Visualization of the cases generated by Omni-Mol and the baseline on three reaction tasks.

## G   Case Study

### G.1   Reaction Tasks

In this subsection, we visualize specific reactions in three reaction tasks. The results in Figure 7 and 8 demonstrate that our method exhibits more accurate generation capabilities compared to the baseline. For example, in the solvent prediction task, we are given the reactants: [C][C][Branch1][C][C][Branch1][C][C][O][C][=Branch1][C][=O][C][C][C][Br] and [C][C][=C][C][Branch1][Ring1][C][#N][=C][C][Branch1][C][Cl][=C][Ring1][=Branch2][O], as well as the product [C][C][=C][C][Branch1][Ring1][C][#N][=C][C][Branch1][C][Cl][=C][Ring1][=-Branch2][O][C][C][C][C][=Branch1][C][=O][O][C][Branch1][C][C][Branch1][C][C][C]. Omni-Mol

| Case 1 Molecule | GT | Separate Tuning | Unified Tuning |
|---|---|---|---|
| | The molecule is a member of the class of pyrazoles that is 1H-pyrazole that is substituted at positions 1, 3, 4, and 5 by 2,6-dichloro-4-(trifluoromethyl)phenyl, cyano, (trifluoromethyl)sulfanyl, and amino groups, respectively. It is a metabolite of the agrochemical fipronil. It has a role as a marine xenobiotic metabolite. It is a member of pyrazoles, a dichlorobenzene, a member of (trifluoromethyl)benzenes, an organic sulfide and a nitrile. | The molecule is a member of the class of pyrazoles that is 1H-pyrazole which is substituted at positions 1, 3, and 5 by 2,4-dichloro-5-chlorophenyl, cyano, and trifluoromethylsulfanyl groups, respectively. It is a member of pyrazoles, a nitrile, a dichlorobenzene, a member of (trifluoromethyl)benzenes and a member of monochlorobenzenes. | The molecule is a member of the class of pyrazoles that is 1H-pyrazole that is substituted at positions 1, 3, 4, and 5 by 2,6-dichloro-4-(trifluoromethyl)phenyl, cyano, trifluoromethyl, and amino groups, respectively. It is a nitrile, a dichlorobenzene, a secondary amino compound, an aromatic primary alcohol, a member of pyrazoles and a member of (trifluoromethyl)benzenes. |

| Case 2 Molecule | GT | Separate Tuning | Unified Tuning |
|---|---|---|---|
| | The molecule is a steroid acid anion, that is the conjugate base of pyridinestrone-3-carboxylic acid. An abiotic metabolite in the 4,5-seco pathway of aerobic estrogen degradation by the bacterium Sphingomonas sp. strain KC8. It is a conjugate base of a pyridinestrone-3-carboxylic acid. | The molecule is a monocarboxylic acid anion that is the conjugate base of 17-oxoestradiol, obtained by deprotonation of the carboxy group; major species at pH 7.3. It is a conjugate base of a 17-oxoestradiol. | The molecule is a steroid acid anion that is the conjugate base of 9-oxo-15alpha-17-epysterone, obtained by deprotonation of the carboxy group; major species at pH 7.3. It is a conjugate base of a 9-oxo-15alpha-17-epysterone. |

Figure 8: Visualization of the cases generated by unified tuning and separate tuning on the molecular captioning task.

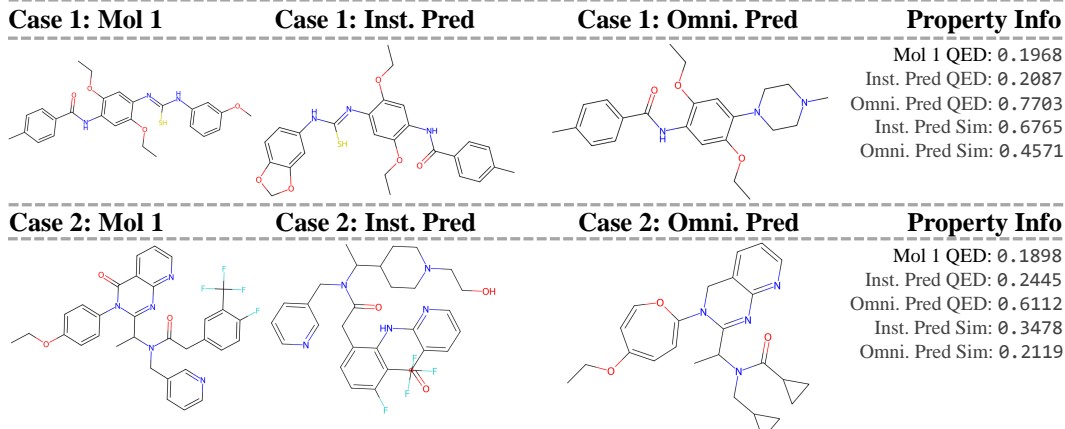

| Case 1: Mol 1 | Case 1: Inst. Pred | Case 1: Omni. Pred | Property Info |
|---|---|---|---|
| | | | Mol 1 QED: 0.1968 |
| | | | Inst. Pred QED: 0.2087 |
| | | | Omni. Pred QED: 0.7703 |
| | | | Inst. Pred Sim: 0.6765 |
| | | | Omni. Pred Sim: 0.4571 |

| Case 2: Mol 1 | Case 2: Inst. Pred | Case 2: Omni. Pred | Property Info |
|---|---|---|---|
| | | | Mol 1 QED: 0.1898 |
| | | | Inst. Pred QED: 0.2445 |
| | | | Omni. Pred QED: 0.6112 |
| | | | Inst. Pred Sim: 0.3478 |
| | | | Omni. Pred Sim: 0.2119 |

Figure 9: Visualization of the cases generated by InstructMol and Omni-Mol on the molecule editing task.

correctly predicts the solvent as [C][N][Branch1][C][C][C][=O], whereas PRESTO predicts an incorrect solvent: [C][C][#N].

## G.2   Molecular Captioning

In the case study of the molecular captioning task, as shown in Figure 8, the model's description of the same molecule becomes more accurate before and after mixed training. It is able to correctly classify and localize functional groups. Does this suggest that the model can learn to identify functional groups from the reaction task? Additionally, constraints from other tasks in the shared representation space also enhance the model's ability to describe molecules. For example, for Case 1 Molecule, Separate Tuning outputs incorrect information regarding the locations of functional groups, whereas Unified Tuning predicts them correctly.

## G.3   Molecule Editing

The case study of the molecule editing task is presented in Figure 9. In this section, we analyze Omni-Mol's "modus operandi" in molecule editing from a biological perspective, examining scaffold architecture, substituent patterns, and other insights to elucidate its potential real-world applicability.

We select optimized molecules meeting both unconstrained and constrained conditions and exceeding the QED threshold (0.6) for comparison, thereby revealing the optimization focus under each setting. And QED is modeled as the desirability profile of eight molecular properties, including molecular

| Model | MW | ALOGP | HBA | HBD | PSA | ROTB | AROM | ALERTS |
|---|---|---|---|---|---|---|---|---|
| QED Properties for Case 1 | | | | | | | | |
| Original | 479.60 | 6.08 | 5 | 3 | 81.18 | 9 | 3 | 3 |
| InstructMol | 493.59 | 5.80 | 6 | 3 | 90.41 | 8 | 3 | 3 |
| Omni-Mol | 397.52 | 3.80 | 5 | 1 | 54.04 | 7 | 2 | 0 |
| QED Properties for Case 2 | | | | | | | | |
| Original | 605.59 | 6.06 | 6 | 0 | 90.21 | 9 | 5 | 1 |
| InstructMol | 587.62 | 4.85 | 7 | 2 | 98.66 | 11 | 3 | 2 |
| Omni-Mol | 434.54 | 4.27 | 5 | 0 | 67.26 | 8 | 1 | 0 |

Table 12: Comparison of QED Properties between InstructMol, and Omni-Mol for two cases

weight (MW: 250–400), lipophilicity (AlogP: 1–4), hydrogen bond donors (HBD: 0–2), hydrogen bond acceptors (HBA: 3–6), polar surface area (PSA: 40–90), number of rotatable bonds (ROTB < 10), aromatic rings (AROM: 0–2), and structural alerts (ALERTS: 0) [5].

For **case 1**, both InstructMol and Omni-Mol retain the benzenediazonium core scaffold while targeting the N′-(3-methoxyphenyl)carbamimidothioic acid substituent, which features two hydrogen-bond donors, two acceptors, two rotatable bonds, and three structural alerts—most notably the carbamimidothioic acid (thiourea-like) moiety associated with acute toxicity. InstructMol merely replaces the terminal anisole with a 1,3-benzodioxole, failing to eliminate the toxicophore and producing negligible changes in overall properties (only one rotatable bond is removed), thus yielding a minimal QED improvement. By contrast, Omni-Mol replaces the entire N′-(3-methoxyphenyl)carbamimidothioic acid substituent with 1-methylpiperazine, eliminating all alerts, removing an aromatic ring, substantially reducing molecular weight (from 479.6 to 397.5 Da) and rotatable bonds, and lowering H-bond donors to one. Benefiting from the hydrophilicity of the 1-methylpiperazine substituent, ALOGP is markedly decreased into the developable range.

For **case 2**, under unconstrained conditions (no hard similarity requirement), the optimization is more flexible, better revealing Omni-Mol's design logic versus InstructMol. InstructMol's molecule still contains multiple aromatic rings, whereas Omni-Mol removes excess aromatics, preserves only the core scaffold, and introduces small saturated rings (e.g., cyclopropyl) to enhance three-dimensionality and rigidity, mitigating risks of over-aromatization. InstructMol's substituents include heavy-atom hydrophobic groups such as trifluoromethyl and fluoro on the aromatic ring, and a reactive aldehyde ($-CHO$) at the terminus. Omni-Mol, however, introduces an ethoxy group in place of the $-CF_3$ unit and installs an amide–cyclopropane fragment, improving physicochemical balance and stability. For side-chain optimization, instead of InstructMol's long, highly polar chain, Omni-Mol cleverly replaces the bulky alkyl-alcohol side chain with a small, conformationally constrained cyclopropyl substituent, also boosting lipophilicity. These combined effects raise the QED score.

In summary, Omni-Mol exhibits a flexible yet systematic optimization strategy—scaffold tuning, side-chain simplification, and substituent "play to strengths and avoid weaknesses", achieving moderate molecular size and balanced hydrophobicity. In contrast, InstructMol tends toward rigid expansion and structural complexity, lacking an adaptable, cohesive design rationale. Thus, Omni-Mol's QED gains derive from rational, streamlined principles that balance flexibility and systemacity, aligning with medicinal chemistry guidelines and reflecting a design ethos likely to resonate with real-world biochemical researchers, offering substantial practical value and significance. More detailed information about the QED properties can be found in Table 12.

# H    task definition and prompt templates

## H.1    Base Chat Template

For LLaMA 3.2 and LLaMA 3.1 instruction-tuned LLMs, we use the base chat template suggested by the official documents, the multi-modal graph tokens are inserted at the beginning of user instructions.

We use `<|finetune_right_pad_id|>` as pad token for SFT.

## H.2 Forward Reaction Prediction

The forward reaction prediction task focuses on determining the chemical product of a reaction given its reactants and reagents. The forward reaction prediction task involves predicting the chemical product of a reaction given the reactants and reagents as input. The input format is structured as the SELFIES representation of reactants, concatenated with a period (".") and the reagent information (*e.g.*, "reactant1.reactant2.reagent"). The task requires the model to process this input and output the corresponding reaction product. The objective is to accurately map the input reaction components to their chemical outcome, leveraging the model's understanding of reaction patterns and transformations. A key challenge in forward reaction prediction is capturing the underlying chemical rules that govern reactivity. The model must infer how functional groups interact, recognize the role of reagents, and apply appropriate transformations to generate the correct product. This process requires a deep understanding of reaction mechanisms, beyond simple pattern recognition. The prompt template is as follows.

---

**Template**

**User Input**: `<user_identifier>` ⊕ `<graph_token>`\n ⊕ Instruction. ⊕ `<SELFIES_reactants>`.`<SELFIES_reagents>` ⊕ `<|eot_id|>` ⊕ `<assistant_identifier>`
**Assistant Output**: `<SELFIES_product>`.`<|eot_id|>`
`<user_identifier>`: `<|start_header_id|>`user`<|end_header_id|>`\n\n
`<assistant_identifier>`: `<|start_header_id|>`assistant`<|end_header_id|>`\n\n

---

**Example**

**Instruction**: Given the reactants and reagents provided, what is a possible product that can be formed?
`<SELFIES_reagents>`: [Br][C][C][Br].[O][C][=C][C][Branch1][C][Br][=C][C][=C][Ring1][#Branch1][Br]
`<SELFIES_reactants>`: [Na+1].[OH-1]

`<SELFIES_product>`: [Br][C][C][O][C][=C][C][Branch1][C][Br][=C][C][=C][Ring1][#Branch1][Br]

---

## H.3 Retrosynthesis

The retrosynthesis task focuses on predicting the reactants required to synthesize a given chemical product, a fundamental challenge in organic chemistry and computational drug discovery. Unlike forward reaction prediction, which maps reactants to products, retrosynthesis operates in reverse, it seeks to determine the most plausible set of precursors that could yield the target compound under appropriate reaction conditions. This task is crucial for designing efficient synthetic routes, enabling chemists to explore viable pathways for molecule construction while minimizing cost and complexity. At the core of this task is a structured input format using SELFIES representations, ensuring a robust and unambiguous encoding of molecular structures. The input consists of the SELFIES representation of the target product, which the model then processes to generate the corresponding reactants. This structured formulation ensures that the model can generalize across diverse chemical transformations, learning the intricate patterns of bond formation and cleavage. A key challenge in retrosynthesis prediction is handling the inherent one-to-many nature of the problem: a single product can often be synthesized through multiple distinct reaction pathways. The model must effectively navigate

this complexity, identifying the most chemically plausible set of reactants based on learned reaction mechanisms. The prompt template is as follows.

---

**Template**

**User Input**: `<user_identifier>` ⊕ `<graph_token>`\n ⊕ Instruction. ⊕ `<SELFIES_product>` ⊕ `<|eot_id|>` ⊕ `<assistant_identifier>`
**Assistant Output**: `<SELFIES_reactants><|eot_id|>`
`<user_identifier>`: `<|start_header_id|>user<|end_header_id|>`\n\n
`<assistant_identifier>`: `<|start_header_id|>assistant<|end_header_id|>`\n\n

---

**Example**

**Instruction**: Which reactants could have been used to generate the given product? The product is:
`<SELFIES_product>`: [C][C][=Branch1][C][=O][C][=C][C][=C][Branch1][C][O][C][Branch1][C][Cl][=C][Ring1][Branch2]

`<SELFIES_reactants>`: [C][C][=Branch1][C][=O][Cl].[O][C][=C][C][=C][C][=C][Ring1][=Branch1][Cl]

---

## H.4 Reagent Prediction

The reagent prediction task focuses on identifying the necessary reagents for a given chemical reaction, a critical step in reaction planning and synthetic chemistry. This task is essential for guiding experimental chemists, as choosing the correct reagents influences reaction efficiency, selectivity, and feasibility. To ensure a structured and standardized input format, we represent the reaction equation using SELFIES, a robust molecular encoding system. The input consists of the SELFIES representations of the reactants, concatenated with a reaction separator "≫", followed by the SELFIES representation of the product. This format (*e.g.*, "reactant1.reactant2≫product") provides a clear, machine-readable structure that allows the model to infer the necessary reagents based on known reaction mechanisms and transformation rules. One of the core challenges in reagent prediction is handling the diversity of chemical transformations. Different reactions require specific reagents that dictate the reaction type, whether it's an oxidation, reduction, coupling, or substitution reaction. The model must learn to recognize reaction context, interpret functional group interactions, and infer the most likely reagents from training data. The prompt template is as follows.

---

**Template**

**User Input**: `<user_identifier>` ⊕ `<graph_token>`\n ⊕ Instruction. ⊕ `<SELFIES_reactants>` ≫ `<SELFIES_product>` ⊕ `<|eot_id|>` ⊕ `<assistant_identifier>`
**Assistant Output**: `<SELFIES_reagents><|eot_id|>`
`<user_identifier>`: `<|start_header_id|>user<|end_header_id|>`\n\n
`<assistant_identifier>`: `<|start_header_id|>assistant<|end_header_id|>`\n\n

---

**Example**

**Instruction**: Can you provide potential reagents for the following chemical reaction? The reaction is
`<SELFIES_reactants>`: [C][C][C][Branch1][#C][C][=C][C][=N][C][Branch1][Ring1][O][C][=C][Ring1][Branch2][C]-[=O][O][C][C][C][O][Ring1][S]
`<SELFIES_product>`: [C][C][C][Branch1][#C][C][=C][C][=N][C][Branch1][Ring1][O][C][=C][Ring1][Branch2][C][O][O]-[C][C][C][O][Ring1][S]

`<SELFIES_reagents>`: [C][C][Branch1][C][C][O].[O].[BH4-1].[Na+1]

---

## H.5 Molecular Captioning

The molecular captioning (Molcap) task focuses on generating descriptive textual information for a given chemical compound based on its molecular structure. This task plays a crucial role in chemical informatics, enabling automated annotation of molecular properties, classification, and functional characteristics. MolCap leverages machine learning models to infer and generate human-readable descriptions that encapsulate key chemical attributes. The input for this task follows a structured format using SELFIES, a robust molecular representation designed for machine learning applications.

The SELFIES encoding of a given compound serves as the input, and the model is responsible for producing a descriptive caption that includes relevant chemical properties. These descriptions can encompass a wide range of molecular characteristics, such as compound classification (*e.g.*, "organic acid," "amine-containing molecule"), pH estimation, presence of functional groups (*e.g.*, "contains a hydroxyl and ketone group"), solubility, toxicity, or other key features. One of the key challenges in molecular captioning is ensuring that the generated text is both chemically accurate and contextually informative. The model must learn to recognize molecular substructures, infer meaningful chemical attributes, and articulate these in a clear and interpretable manner. The prompt template is as follows.

---

**Template**

**User Input**: `<user_identifier>` ⊕ `<graph_token>` \n ⊕ Instruction. ⊕ `<SELFIES_compound>` ⊕ `<|eot_id|>` ⊕ `<assistant_identifier>`
**Assistant Output**: Description.`<|eot_id|>`
`<user_identifier>`: `<|start_header_id|>`user`<|end_header_id|>`\n\n
`<assistant_identifier>`: `<|start_header_id|>`assistant`<|end_header_id|>`\n\n

---

**Example**

**Instruction**: Please give me some details about this molecule. The compound SELFIES sequence is:
`<SELFIES_compound>`:[C][=C][C][=Branch2][Ring1][S][=C][C][=C][Ring1][=Branch1][C][C][Branch2][Ring1][Branch1]-[C][Branch1][P][C] [Branch1][Ring2][O][Ring1][Branch1][C][O][P][=Branch1][C][=O][Branch1][C][O-1][O-1][O][O][O]

**Description**: The molecule is an organophosphate oxoanion obtained by deprotonation of the phosphate OH groups of 4-(5-O-phospho-beta-D-ribofuranosyl)phenol; major species at pH 7.3. It derives from a D-ribofuranose 5-phosphate(2-). It is a conjugate base of a 4-(5-O-phospho-beta-D-ribofuranosyl)phenol.

---

## H.6 Quantum Mechanics Property Prediction

The quantum mechanics property prediction task focuses on determining key quantum-mechanical properties of a given chemical compound, providing critical insights into its electronic behavior, stability, and potential applications. This task is essential in computational chemistry, materials science, and drug discovery, where quantum properties influence molecular interactions, reactivity, and optoelectronic performance. The input follows a structured format using SELFIES, a robust molecular representation optimized for machine learning applications. Given the SELFIES encoding of a molecule, the model is tasked with predicting its quantum properties, such as the highest occupied molecular orbital (HOMO) energy, lowest unoccupied molecular orbital (LUMO) energy, and the HOMO–LUMO gap. These properties are fundamental in determining a molecule's electronic structure, with implications for charge transfer, chemical reactivity, and photophysical behavior. One of the key challenges in quantum property prediction is capturing the underlying quantum-chemical interactions that govern molecular behavior. The prompt template is as follows.

---

**Template**

**User Input**: `<user_identifier>` ⊕ `<graph_token>`\n ⊕ Instruction. ⊕ `<SELFIES_compound>` ⊕ `<|eot_id|>` ⊕ `<assistant_identifier>`
**Assistant Output**: Property.`<|eot_id|>`
`<user_identifier>`: `<|start_header_id|>`user`<|end_header_id|>`\n\n
`<assistant_identifier>`: `<|start_header_id|>`assistant`<|end_header_id|>`\n\n

---

**Example**

**Instruction**: What is the HOMO-LUMO gap of this molecule? The compound SELFIES sequence is:
`<SELFIES_compound>`: [N][=C][O][C][=C][C][=Branch1][Ring2][=N][Ring1][=Branch1][C][#N]

**Property**: 0.1487

---

## H.7 Catalyst Prediction

The catalyst prediction task focuses on identifying the appropriate catalysts required to facilitate a given chemical reaction. Catalysts play a crucial role in modifying reaction pathways, lowering activation energy, and improving reaction efficiency without being consumed in the process. The input follows the SELFIES representation, a robust molecular encoding system designed for computational applications. The reaction is expressed as an equation where the SELFIES representations of the reactants are concatenated and separated from the product using "≫" (*e.g.*, "reactant1.reactant2≫product"). This structured representation allows the model to process the reaction as a whole and infer the most suitable catalyst that enables the transformation. One of the primary challenges in catalyst prediction is understanding the nuanced role that catalysts play in different reaction mechanisms. Unlike reagents, which directly participate in the reaction, catalysts provide alternative pathways to enhance reaction kinetics. The prompt template is as follows.

---

**Template**

**User Input**: `<user_identifier>` ⊕ `<graph_token>`\n ⊕ Instruction. ⊕ `<SELFIES_reactants>` ≫ `<SELFIES_product>` ⊕ `<|eot_id|>` ⊕ `<assistant_identifier>`
**Assistant Output**: `<SELFIES_catalysts><|eot_id|>`
`<user_identifier>`: `<|start_header_id|>user<|end_header_id|>`\n\n
`<assistant_identifier>`: `<|start_header_id|>assistant<|end_header_id|>`\n\n

**Example**

**Instruction**: Given this chemical reaction, what are some catalysts that could have been used? The reaction is
`<SELFIES_reactants>`: [C][C][=C][C][=C][Branch1][Ring1][C][#N][C][=C][Ring1][Branch2][C][Branch1][C][F][Branch1][C][F][F].[O][=C][C][C][C][=Branch1][C][=O][N][Ring1][=Branch1][Br]
`<SELFIES_product>`: [N][#C][C][=C][C][=C][Branch1][Ring1][C][Br][C][Branch1][=Branch2][C][Branch1][C][F][Branch1][C][F][F][=C][Ring1][N]

`<SELFIES_catalysts>`:[O][=C][Branch1][#C][O][O][C][=Branch1][C][=O][C][=C][C][=C][C][=C][Ring1][=Branch1][C][=C][C][=C][C] [=C][Ring1][=Branch1]

---

## H.8 Solvent Prediction

The solvent prediction task focuses on identifying the appropriate solvents required for a given chemical reaction. Solvents play a crucial role in determining reaction efficiency, influencing factors such as solubility, reaction kinetics, selectivity, and stability of intermediates. To ensure a structured and machine-readable representation, the input follows the SELFIES format, a robust molecular encoding system designed for computational applications. The reaction is expressed as an equation where the SELFIES representations of the reactants are concatenated and separated from the product using the reaction separator "≫" (*e.g.*, "reactant1.reactant2≫product"). This structured format allows the model to interpret the reaction context and infer the most suitable solvents required to facilitate the transformation. One of the key challenges in solvent prediction is understanding the diverse roles solvents play in different reaction mechanisms. The prompt template is as follows.

## H.9 Yield Regression

The yield regression task focuses on estimating the proportion of the actual product obtained in a chemical reaction relative to its theoretical maximum. Reaction yield is a critical metric in organic synthesis, pharmaceutical manufacturing, and industrial chemistry, as it directly influences process efficiency, resource utilization, and cost-effectiveness. The input follows the SELFIES format, a robust molecular encoding system tailored for computational chemistry. The reaction is expressed as an equation where the SELFIES representations of the reactants are concatenated and separated from the product using the reaction separator "≫" (*e.g.*, "reactant1.reactant2≫product"). This structured format provides a standardized input for the model, allowing it to interpret the reaction context and estimate the expected yield. One of the key challenges in yield prediction is capturing the complex interplay between reaction conditions, molecular stability, steric effects, and solvent or catalyst influences. The prompt template is as follows.

## H.10 LogP Prediction

The LogP prediction task focuses on determining the octanol–water partition coefficient (LogP) of a given chemical compound, a key physicochemical property that influences molecular behavior

across various environments. LogP quantifies the relative solubility of a compound in octanol versus water, serving as a critical indicator of lipophilicity, hydrophobicity, and membrane permeability. The task employs the SELFIES molecular representation, which encodes chemical structures in a machine-readable form optimized for deep learning models. Given the SELFIES representation of a compound, the model is responsible for predicting its LogP value, a numerical measure that typically ranges from negative values (indicating high water solubility) to positive values (indicating high lipophilicity). This structured approach allows the model to learn patterns between molecular structure and partitioning behavior, enabling accurate and data-driven LogP estimation. One of the key challenges in LogP prediction is capturing the complex molecular interactions that dictate solubility behavior. The prompt template is as follows.

---

**Template**

**User Input**: `<user_identifier>` ⊕ `<graph_token>`\n ⊕ Instruction. ⊕ `<SELFIES_compound>` ⊕ `<|eot_id|>` ⊕ `<assistant_identifier>`
**Assistant Output**: Property.`<|eot_id|>`
`<user_identifier>`: `<|start_header_id|>`user`<|end_header_id|>`\n\n
`<assistant_identifier>`: `<|start_header_id|>`assistant`<|end_header_id|>`\n\n

**Example**

**Instruction**: I am interested in the LogP of this molecule, could you tell me what it is? If uncertain, provide an estimate. Respond with the numerical value only. The molecule SELFIES sequence is:
`<SELFIES_compound>`: [C][O][C][=C][C][=Branch1][=Branch2][=C][C][=Branch1][Ring2][=C][Ring1][=Branch1][O]-[C@H1][C@H1][C@@H1][Branch1][Branch1][C][O][Ring1][Branch1][C@@H1][Branch1][=Branch1][O][C][Ring1][#Branch1][=O][C][=C][C][=Branch1][=C][=C][Branch1][=Branch2][C][=Branch1][Ring2][=C][Ring1][=Branch1][O][C][O][C][O]

**Property**: The LogP for the input molecule is 2.00.

---

## H.11 Molecular Weight Prediction

The molecular weight prediction task focuses on determining the molecular weight of a given chemical compound, a fundamental property that reflects its size and atomic composition. Molecular weight is a crucial parameter in various scientific disciplines, including organic synthesis, drug design, polymer chemistry, and materials science. It influences key aspects such as reaction stoichiometry, diffusion rates, bioavailability, and stability. The input follows the SELFIES format, a robust molecular encoding system designed for computational chemistry applications. The input consists of the SELFIES representation of a molecule, which the model processes to predict its molecular weight in unified atomic mass units (Da). This structured approach allows the model to learn the relationships between molecular structure and atomic composition, enabling precise and efficient molecular weight estimation. The prompt template is as follows.

---

**Template**

**User Input**: `<user_identifier>` ⊕ `<graph_token>`\n ⊕ Instruction. ⊕ `<SELFIES_compound>` ⊕ `<|eot_id|>` ⊕ `<assistant_identifier>`
**Assistant Output**: Property.`<|eot_id|>`
`<user_identifier>`: `<|start_header_id|>`user`<|end_header_id|>`\n\n
`<assistant_identifier>`: `<|start_header_id|>`assistant`<|end_header_id|>`\n\n

**Example**

**Instruction**: Please provide me with the Molecular Weight value of this molecule. Determine the Molecular Weight value of this molecule. If uncertain, provide an estimate. Respond with the numerical value only. The molecule SELFIES sequence is:
`<SELFIES_compound>`: [C][O][C][=C][Branch1][#Branch1][C][=C][C][=N][Ring1][=Branch1][C][=Branch1][C][=O][N][C]-[C][C][C][C][N][Branch1][Branch1][C][C][Ring1][=Branch1][S][=Branch1][C][=O][=Branch1][C][=O][N][C][=Branch1]-[C][=O][N][C][C][C][C][C][C][Ring1][Branch1][C][=C][Ring1][Branch1]

**Property**: The Molecular Weight for the input molecule is 491.60 g/mol.

---

## H.12 Topological Polar Surface Area Prediction

The topological polar surface area (TPSA) prediction task focuses on determining the TPSA value of a given chemical compound, a key descriptor that reflects its molecular polarity and hydrogen bonding capacity. TPSA is widely used in cheminformatics, particularly in drug discovery, where it serves as an important predictor of solubility, permeability, and absorption. A compound's TPSA value influences its bioavailability, blood-brain barrier penetration, and interactions with biological membranes, making accurate prediction essential for pharmaceutical and materials research. The input is the SELFIES representation of the compound, and the model is tasked with predicting the compound's TPSA. The objective is to provide insights into the compound's polarity, solubility, and potential absorption characteristics, which are crucial considerations in areas such as drug discovery and materials research. The prompt template is as follows.

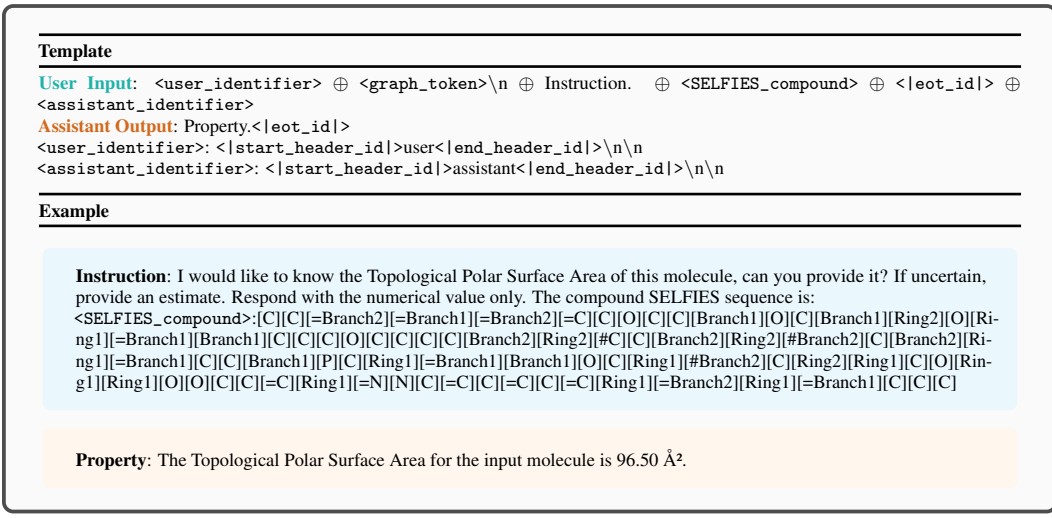

## H.13 Description Q&A

The description question and answer task involves responding to queries regarding a given compound's properties in physical chemistry and related fields. The input is the SELFIES representation of the compound. The model is tasked with providing accurate answers to detailed questions about the compound's physical and chemical attributes, encompassing a broad range of topics—from pharmacological considerations to the specific influence of structural and functional groups on biological activity (*e.g.*, in anticancer agents such as 4-Hydroxycyclophosphamide or Lobaplatin). The objective is to achieve a comprehensive and in-depth understanding of the compound's characteristics. The prompt template is as follows.

**Template**

**User Input**: `<user_identifier>` ⊕ `<graph_token>`\n ⊕ Instruction. ⊕ `<SELFIES_compound>` ⊕ `<|eot_id|>` ⊕ `<assistant_identifier>`
**Assistant Output**: Description.`<|eot_id|>`
`<user_identifier>`: `<|start_header_id|>`user`<|end_header_id|>`\n\n
`<assistant_identifier>`: `<|start_header_id|>`assistant`<|end_header_id|>`\n\n

**Example**

**Instruction**: What is the main component of Lobaplatin that gives it its anticancer properties? The compound SELFIES sequence is:
`<SELFIES_compound>`: [C][C@@H1][Branch1][=Branch1][C][=Branch1][C][=O][O-1][O-1].[C][C][C@H1][Branch1]-[=Branch1][C@@H1][Ring1][Ring2][C][N][C][N].[Pt+2]

**Description**: The main component of Lobaplatin that gives it its anticancer properties is the platinum cation (Pt+2).

## H.14 Experimental Procedure Prediction

The experimental procedure prediction task focuses on generating a detailed, step-by-step protocol for conducting chemical experiments based on a given set of experimental components. This task plays a crucial role in automated synthesis planning, high-throughput experimentation, and robotic chemistry, where structured experimental procedures are essential for reproducibility, efficiency, and accuracy. The input is a mapping between experimental components and their corresponding SELFIES representations (*e.g.*, "Reactants: $index$: SELFIES" ...). The model is tasked with producing a structured sequence of operations that associates each component with the detailed steps of the experiment. The objective is to automate the chemical synthesis process by providing executable, structured experimental procedures. The prompt template is as follows.

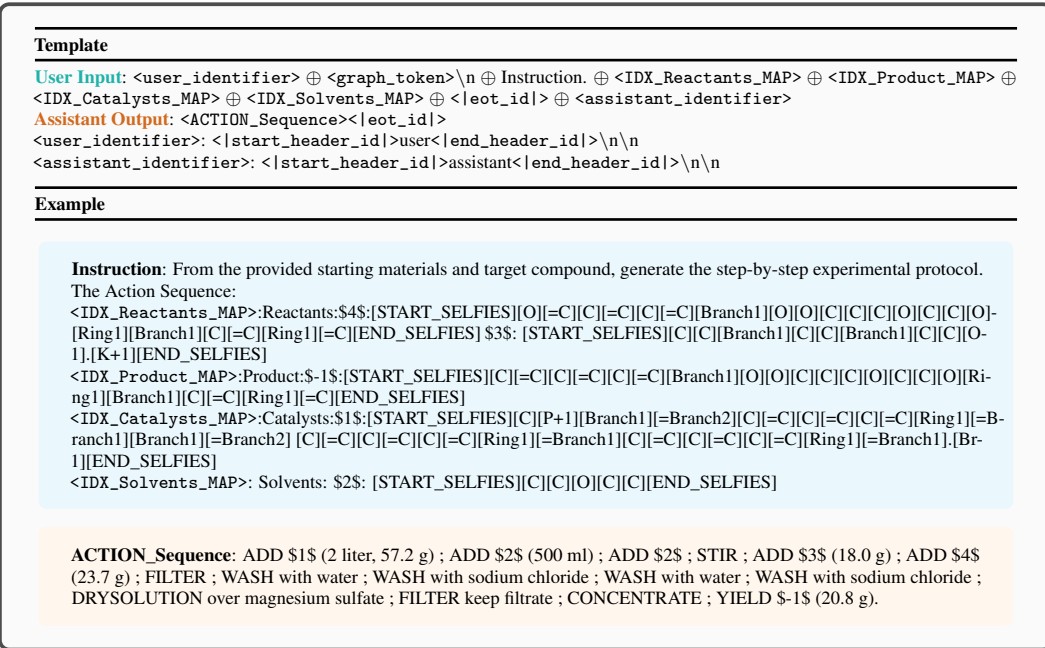

## H.15 IUPAC2SELFIES

In the IUPAC2SELFIES task, the model is asked to generate the SELFIES representation of a molecule based on its IUPAC name. The IUPAC (International Union of Pure and Applied Chemistry) name is a standardized and systematic way to uniquely describe a molecule's structure using rules for parent chains, functional groups, and substituents. To succeed in this task, the model must accurately understand these rules and produce the correct SELFIES representation accordingly.

---

**Template**

**User Input**: `<user_identifier>` ⊕ Instruction. ⊕ `<assistant_identifier>`
**Assistant Output**: `<SELFIES_compound><|eot_id|>`
`<user_identifier>`: `<|start_header_id|>user<|end_header_id|>\n\n`
`<assistant_identifier>`: `<|start_header_id|>assistant<|end_header_id|>\n\n`

**Example**

**Instruction**: What is the molecular structure corresponding to this IUPAC name? The IUPAC name is: N-(1-amino-2-methylpropan-2-yl)-N-propylmorpholine-4-carboxamide

`<SELFIES_compound>`: [C][C][C][N][Branch1][=N][C][=Branch1][C][=O][N][C][C][O][C][C][Ring1][=Branch1][C]-[Branch1][C][C][Branch1][C][C][C][N]

---

### H.16 Text Guided Molecule Generation

In this task, the model is asked to generate a molecule in SELFIES notation based on a textual description. The description includes information such as functional group substitutions, structural features, and desired properties. The model must comprehend this complex information and design a molecule that satisfies the given requirements.

---

**Template**

**User Input**: `<user_identifier>` ⊕ Instruction. ⊕ `<assistant_identifier>`
**Assistant Output**: `<SELFIES_compound><|eot_id|>`
`<user_identifier>`: `<|start_header_id|>user<|end_header_id|>\n\n`
`<assistant_identifier>`: `<|start_header_id|>assistant<|end_header_id|>\n\n`

**Example**

> **Instruction**: Design a molecule according to the description provided. The description is: The molecule is a nitrile that is propionitrile in which one of the methyl hydrogens has been replaced by a phenyl group. It is a nitrile and a member of benzenes. It is functionally related to an acetonitrile. The molecule is a natural product found in Brassica napus, Brassica oleracea, and other organisms with data available.

> `<SELFIES_compound>`: [C][=C][C][=C][Branch1][Branch1][C][=C][Ring1][=Branch1][C][C][C][#N]

---

### H.17 Molecule Editing

The Molecule Editing task is concerned with optimizing one or more target properties of a given molecule, particularly the DRD2 score and QED. Specifically, the objective is to apply structural edits to a molecule with low DRD2 score and low QED in order to generate a variant exhibiting higher DRD2 score and QED. The model is tasked with optimizing the QED or DRD2 score of an input SELFIES molecule while maintaining some extent of structural similarity. It outputs the SELFIES of the optimized molecule. The objective is to optimize specific properties of a given molecule so as to yield novel biological insights. The prompt template is as follows.

---

**Template**

**User Input**: `<user_identifier>` ⊕ `<graph_token>\n` ⊕ Instruction. ⊕ `<SELFIES_compound_Input>` ⊕ `<|eot_id|>` ⊕ `<assistant_identifier>`
**Assistant Output**: `<SELFIES_compound><|eot_id|>`
`<user_identifier>`: `<|start_header_id|>user<|end_header_id|>\n\n`
`<assistant_identifier>`: `<|start_header_id|>assistant<|end_header_id|>\n\n`

**Example**

> **Instruction**: Develop a new molecular structure that enhances the QED score and keeps structural similarity to the initial molecule. The compound SELFIES sequence is:
> `<SELFIES_compound_Input>`: [C][C][=Branch1][C][=O][N][C][C][C][N][Branch2][Ring1][C][C][=C][C][=C][Branch1]-[Branch2][C][=Branch1][C][=O][C][C][#N][C][=C][Ring1][O][C][=Branch1][C][=O][O][Ring1][P]

> `<SELFIES_compound>`: [C][C][=Branch1][C][=O][N][C][C][C][N][Branch2][Ring1][Ring1][C][=C][C][=C][Branch1]-[=Branch2][C][Branch1][C][C][Branch1][C][C][C][C][=C][Ring1][#Branch2][C][=Branch1][C][=O][O][Ring1][S]

---

## I Discussion on Generalist and Specialist

Here, we define the notions of generalist and specialist models. A **generalist model** refers to a single model with shared parameters $\theta$ that is trained across an entire set of tasks $\mathcal{T} = \{T_i\}$. In contrast, a **specialist model** assigns a distinct set of parameters $\theta_i$ to each individual task $T_i$.

Although the authors of InstructMol and HIGHT claim their models to be generalist in their respective papers, they in fact employ distinct LoRA adapters for different tasks, following the formulation $\theta_i = \theta_0 + \theta_{\text{lora}_i}$, where $\theta_0$ denotes the pretrained LLM parameters and $\theta_{\text{lora}_i}$ represents the task-specific LoRA parameters for task $T_i$. As a result, these models do not satisfy the criteria for being

| Configuration | Exact | BLEU | Levenshtein | RDK | MACCS | Morgan | Validity |
|---|---|---|---|---|---|---|---|
| Continual Learning Performance on **Retrosynthesis** | | | | | | | |
| Retrosynthesis only | 0.55 | 0.95 | 9.93 | 0.83 | 0.89 | 0.80 | 1.00 |
| Forward + Retrosynthesis | 0.50 | 0.94 | 11.01 | 0.77 | 0.82 | 0.76 | 1.00 |

Table 13: Continual Learning performance on retrosynthesis task before and after learning on forward prediction.

true generalist models. In contrast, both PRESTO and Omni-Mol utilize a single parameter set $\theta$ shared across all tasks.

## J  Discussion on Continual Learning

Instead of unified instruction tuning, which trains the model to learn all tasks simultaneously in a single procedure, it is also plausible to train the model sequentially on various tasks.

We conduct an experiment in a continual learning fashion: we first train the model on the retrosynthesis task and then use the trained model to learn forward prediction. We record the performance on the retrosynthesis task both before and after training on forward prediction. As shown in Table 13, after training on forward prediction, we observed a significant performance drop. This suggests that the model may forget previously learned knowledge and cannot benefit from earlier tasks. Therefore, in Omni-Mol, we train all tasks jointly.

