# OpenReview forum: "Omni-Mol: Multitask Molecular Model for Any-to-any Modalities"
_NeurIPS.cc/2025/Conference — NeurIPS 2025 poster_

### Official Review · Reviewer_FtzT · 2025-06-09

**Clarity:** 3
**Significance:** 3
**Originality:** 2
**Rating:** 4
**Confidence:** 4

**Summary:**

The authors propose  OmniMol, a generalist chemistry LLM with the ability to solve a wide range of tasks. OmniMol is a Llama 3.2 model finetuned with a novel LORA-based approach designed for multitask learning that solves Text2Mol, Mol2Text, Mol2Mol and Mol2Num tasks. Experiments with up to 8B models were done on a multitask dataset with a total of 1.4M samples

**Questions:**

- I have some concerns on the results. (1) In some tables OmniMol is incorrectly bolded  (e.g., Molecular Captioning Task, it is not the best). (2) It seems that some cherry-picking on the competing methods was done, e.g., the Text+Chem T5 model achieved better performance on the two tasks (Mol2Text and Text2Mol) of Chebi-20.
	- I had a look at the code but it has no README which hampers reproducibility. Also, will OmniMol be available publicly, e.g., via a HF spaces?
	- The section on SMILES vs SELFIES gives the wrong perception SMILES may often cause errors in RDKit. I think you want to say that such errors occur frequently *when generating SMILES with a LM*. Please be more explicit here. Also, I think that a small ablation on SMILES vs SELFIES would be useful because only ~55% of the samples have SMILES/SELFIES output whereas ~90% have SMILES/SELFIES input. Then, when you tokenize SELFIES with the BPE-based Lama tokenizer, you may distort the structure of the SELFIES such that not all generated SELFIES are valid anymore. E.g., when you tokenize [C][C] into "[C" "][ "C" and "]" you arrive at a vocab that can still be used to produce invalid SELFIES like "][]".
	- P4 L150: Do you mean the hidden size of the LLM's nn.Embedding and the GNN's node embedding size? If really only the sequence lengths differ you just stack horizontally and no padding is necessary (apart from standard padding when you form batches).
	- Figure 4: I am buying the scalability argument but the visualization is unfortunate as the bar seems flat -- I would visualize this on a relative scale (y-axis is the delta compared to the first value)

**Ethical Concerns:**

["NO or VERY MINOR ethics concerns only"]

**Final Justification:**

While I still believe that the methodological novelty is relatively low and OmniMol does not really unlock new capabilities of LLMs in chemistry, the rebuttal helped me to understand better the richness of the various datasets and the effort in homogenizing and providing this data to the community. Moreover, the scale of the model is significant (considering standard academic resources) and performance gains to previous methods seem robust. Therefore, I increased my score from 3 to 4.

**Limitations:**

There is a very generic and shallow section about limitations in the appendix.
The main paper lacks a substantial discussion of its own limitations

**Paper Formatting Concerns:**

--

**Quality:**

3

**Strengths And Weaknesses:**

The significance of this research topic is high since generalist chemistry LLMs would benefit the CS as well as the chemistry community well.

The originality seems medium given that many others have tackled this problem before in a similar fashion. The methodological delta to existing methods is small. E.g., even the early Text+ChemT5 work named as "pioneering" by the authors can already do all four mentioned tasks natively (Text+Chem T5 is not explicitly evaluated on Mol2Num but it can solve it naturally without any architectural modification).

On the positive side, the authors propose a custom LORA-based method (MoGE) which boosts performance. This approach and its usage in a MOE settings is well-motivated and interesting, especially also the ablation study on its value compared to vanilla LORA.

Clarity is good overall but can be improved, e.g., the figure ordering is random and they are not referenced in order.

The results are generally convincingly presented but I have concerns that some of them are cherry-picked (see below)

---

> ### Author Rebuttal · Authors · 2025-07-31
>
> >Q1: The methodological delta to existing methods is small. E.g., even the early Text+ChemT5 work named as "pioneering" by the authors can already do all four mentioned tasks natively (Text+Chem T5 is not explicitly evaluated on Mol2Num but it can solve it naturally without any architectural modification).
>
> A1: Thanks for the comment. In fact, on our comprehensive general-purpose molecular task dataset, no existing baseline has been able to fully handle the range of tasks. The main challenge lies in the instability of LLM training when faced with highly diverse task types—simply adding more task-specific data does not lead to effective scaling. OmniMol is the first to address this problem by introducing GAL-MoE, a carefully designed mechanism that enables stable and scalable multi-task learning. We also appreciate your mention of Text-Chem T5, and we will provide a detailed comparison with it in the following response.
>
> >Q2: Clarity is good overall but can be improved, e.g., the figure ordering is random and they are not referenced in order.
>
> A2: Thanks for the comment. We will adjust the layout of our paper.
>
> >Q3: In some tables OmniMol is incorrectly bolded.
>
> A3: Thanks for the comment. We have noticed the typo and revised it.
>
> >Q4: Cherry-picking on the competing methods.
>
> A4: Thanks for the comment. We appreciate the reviewer for bringing up this paper, but we respectively disagree with the implication that we are cherry-picking. We had already taken note of this work and reproduced it on our dataset; however, the reproduced results turned out to be quite poor. The detailed experimental results are shown below.
>
> Forward Pred
>
> | Model | Exact | BLEU | LEVEN | MACCS | RDK | MORGAN | Val |
> | --- | --- | --- | --- | --- | --- | --- | --- |
> | Text-Chem T5 | 0.3036 | 0.8808 | 17.2164 | 0.8066 | 0.6456 | 0.5904 | 0.99 |
> | Omni-Mol | 0.73 | 0.980 | 5.55 | 0.947 | 0.895 | 0.87 | 1.00 |
>
> Reagent Pred
>
> | Model | Exact | BLEU | LEVEN | MACCS | RDK | MORGAN | Val |
> | --- | --- | --- | --- | --- | --- | --- | --- |
> | Text-Chem T5 | 0.0437 | 0.5206 | 25.711 | 0.4735 | 0.3417 | 0.3061 | 0.98 |
> | Omni-Mol | 0.23 | 0.726 | 14.59 | 0.557 | 0.627 | 0.52 | 1.00 |
>
> Retrosynthesis
>
> | Model | Exact | BLEU | LEVEN | MACCS | RDK | MORGAN | Val |
> | --- | --- | --- | --- | --- | --- | --- | --- |
> | Text-Chem T5 | 0.1538 | 0.7965 | 23.601 | 0.7813 | 0.6465 | 0.5661 | 0.99 |
> | Omni-Mol | 0.57 | 0.960 | 8.97 | 0.909 | 0.864 | 0.83 | 1.00 |
>
> Solvent Pred
>
> | Model | Exact | BLEU | LEVEN | MACCS | RDK | MORGAN | Val |
> | --- | --- | --- | --- | --- | --- | --- | --- |
> | Text-Chem T5 | 0.2694 | 0.6114 | 4.144 | 0.4467 | 0.4196 | 0.3785 | 1 |
> | Omni-Mol | 0.52 | 0.759 | 2.71 | 0.673 | 0.671 | 0.64 | 1 |
>
> Catalyst Pred
>
> | Model | Exact | BLEU | LEVEN | MACCS | RDK | MORGAN | Val |
> | --- | --- | --- | --- | --- | --- | --- | --- |
> | Text-Chem T5 | 0.5959 | 0.5988 | 3.9208 | 0.8110 | 0.8221 |  0.6069 | 0.99 |
> | Omni-Mol | 0.72 | 0.792 | 1.96 | 0.886 | 0.904 | 0.72 | 1.00 |
>
> Quantum Mechanics Property Prediction Task
>
> | Model | HOMO | LUMO | GAP | AVG |
> | --- | --- | --- | --- | --- |
> | Text-Chem T5 | 0.0083 | 0.0095 | 0.0108 |  0.0095 |
> | Omni-Mol | 0.0038 | 0.0047 | 0.0049 | 0.0044 |
>
> Mol2Num
>
> | Model | logP | TPSA | Weight |
> | --- | --- | --- | --- |
> | Text-Chem T5 | 2.346 | 39.771 | 85.472 |
> | Omni-Mol | 0.49 | 5.89 | 11.07 |
>
> IUPAC2SELFIES
>
> | Model | Exact | BLEU | LEVEN | MACCS | RDK | MORGAN | Val |
> | --- | --- | --- | --- | --- | --- | --- | --- |
> | Text-Chem T5 | 0.0204 | 0.7097 | 28.9402 | 0.6792 | 0.4149 |  0.3302 | 0.96 |
> | Omni-Mol | 0.39 | 0.952 | 13.38 | 0.871 | 0.729 | 0.69 | 0.996 |
>
> Description Q&A Task
>
> | Model | BLEU-2 | BLEU-4 | METEOR | ROUGE-1 | ROUGE-2 | ROUGE-L |
> | --- | --- | --- | --- | --- | --- | --- |
> | Text-Chem T5 | 0.4971 | 0.4201 | 0.5434 | 0.5272 | 0.3770 | 0.4938 |
> | Omni-Mol | 0.52 | 0.44 | 0.58 | 0.53 | 0.38 | 0.49 |
>
> Experimental Procedure Prediction
>
> | Model | BLEU-2 | BLEU-4 | ROUGE-1 | ROUGE-2 | ROUGE-L |
> | --- | --- | --- | --- | --- | --- |
> | Text-Chem T5 | 0.5733 | 0.4704 | 0.5405 | 0.2831 | 0.4794 |
> | Omni-Mol | 0.572 | 0.448 | 0.532 | 0.274 | 0.464 |
>
> Molecule Captioning
>
> | Model | BLEU-2 | BLEU-4 | METEOR | ROUGE-1 | ROUGE-2 | ROUGE-L |
> | --- | --- | --- | --- | --- | --- | --- |
> | Text-Chem T5 | 0.3324 | 0.2408 | 0.3871 | 0.4543 | 0.2765 | 0.3991 |
> | Omni-Mol | 0.529 | 0.440 | 0.571 | 0.604 | 0.447 | 0.541 |
>
> Text Guided Molecule Generation
>
> | Model | Exact | BLEU | LEVEN | MACCS | RDK | MORGAN | Val |
> | --- | --- | --- | --- | --- | --- | --- | --- |
> | Text-Chem T5 | 0.0014 | 0.6241 | 43.574 | 0.5082 | 0.2961 | 0.1830 | 0.72 |
> | Omni-Mol | 0.12 | 0.824 | 23.59 | 0.721 | 0.562 | 0.442 | 0.963 |
>
> Yield Prediction
>
> | Model | BH | SM |
> | --- | --- | --- |
> | Text-Chem T5 | 0.171 | 0.151 |
> | Omni-Mol | 0.94 | 0.68 |
>
> To ensure fairness and transparency, we have publicly released all model files, hyperparameter configurations, training procedures, and training logs for Text-Chem T5 in our GitHub repository (TextChemT5-v0-main folder). In the above table, Following the Text-Chem T5 paper, we fine-tuned its parameters to ensure the best possible performance. Evidently, Omni-Mol outperforms Text-Chem T5 on almost all tasks by a significant margin. If you have any insights or results from reproducing this work, we would be happy to discuss and compare findings in the subsequent stages.
>
> > Q5: Limited reproducibility.
>
> A5: Thanks for the comment and interest. After submitting the paper, we conduct a thorough audit of our code to ensure its correctness and reproducibility. We are highly committed to open-source practices. If you revisit our repository now, you will find that we have updated the code with refined implementation details, added comprehensive comments, provided a complete and well-documented training pipeline, and released our model weights. This allows for a smooth and fully reproducible replication of our results.
>
> > Q6: Wrong perception SMILES may often cause errors in RDKit.
>
> A6: Thanks for the comment. The clear statement is that when an LLM outputs SMILES, it is more prone to generating errors, which can result in outputs that cannot be parsed into valid molecules.
>
> > Q7: Then, when you tokenize SELFIES with the BPE-based Lama tokenizer, you may distort the structure of the SELFIES such that not all generated SELFIES are valid anymore. E.g., when you tokenize [C][C] into "[C" "][ "C" and "]" you arrive at a vocab that can still be used to produce invalid SELFIES like "][]".
>
> A7: Thanks for pointing out, indeed, there is no perfect 1D representation for molecules, we use SELFIES for its robustness and broader applicability. In fact, many of our baseline results are based on SMILES, meaning they fully leverage the pretrained knowledge embedded within the LLM. In contrast, our method achieves SOTA performance without relying on such pretrained domain knowledge, which further highlights the effectiveness of Omni-Mol.
>
> > Q8: P4 L150: Do you mean the hidden size of the LLM's nn.Embedding and the GNN's node embedding size? If really only the sequence lengths differ you just stack horizontally and no padding is necessary (apart from standard padding when you form batches).
>
> A8: Thanks for the question, actually, the padding is due to the varying seuqence length, i.e. differnet number of nodes in molecular embeddings from GNN. We pad them into a regular shape to allow parallel training.
>
> > Q9: Figure 4: I am buying the scalability argument but the visualization is unfortunate as the bar seems flat -- I would visualize this on a relative scale (y-axis is the delta compared to the first value)
>
> A9: Thanks for the advice, we will adjust the figure to enable better visualization
>
> > Q10: Add useful ablation on SMILES vs SELFIES.
>
> A10: Please refer to  Reviewer 8Wa4 A4

---

> > ### Author Response · Authors · 2025-08-02
> > **Supplementary Response to Reviewer FtzT**
> >
> > Thank you very much for your valuable and encouraging feedback on our work. We truly appreciate your recognition. From your review, it is evident that you have developed a deep and accurate understanding of how to train biological and chemical LLM.
> >
> > > Your main concern is the potential risk of cherry-picking in our method. In fact, we would like to clarify that there is no such issue. We were well aware of the Text+ChemT5 work early on; however, the primary reason we did not include it in our comparison is that we were unable to reproduce its results. To ensure transparency, we have publicly released our reproduction attempts, including experimental logs, implementation details, and hyperparameter configurations, in an anonymous GitHub repository. If you have any suggestions for improving this process, we would be glad to engage in further discussion.
> >
> > Thank you for taking the time to help us improve the quality of our manuscript. If you have any further suggestions, we would be more than happy to engage in further discussion with you.

---

> > > ### Author Response · Authors · 2025-08-05
> > > **Official Comment by Authors**
> > >
> > > Dear Reviewer FtzT,
> > >
> > > Thank you for providing such detailed reviews. We deeply appreciate the time and effort you have invested in reviewing our work and offering these detailed comments.
> > >
> > > We have conducted additional experiments to address your concerns regarding the potential risk of cherry-picking in our method, and have provided thorough justification for other related details. If you feel there are any aspects that **require further clarification or improvement**, we are more than willing to provide additional information or clarifications during the discussion period to assist with your review process.
> > >
> > > Thank you again for your time and expertise. Your feedback has been instrumental in enhancing the quality of our work.

---

> ### Author Response · Authors · 2025-08-06
>
> Dear Reviewer FtzT,
>
> Thank you for your insightful and constructive reviews. We would like to kindly remind you that the discussion period will conclude in three days. In the meantime, we have provided detailed supplementary experiments and clarifications addressing your concerns.
>
> If you have any additional questions or would like us to conduct further experiments, **please do not hesitate to let us know**, we would be more than happy to engage in further discussion and provide additional details about our work.
>
> Thank you once again for your time and expertise. We look forward to hearing from you.
>
> Sincerely, Submission 1411 Authors

---

> > ### Comment · Reviewer_FtzT · 2025-08-06
> > **Response to rebuttal**
> >
> > Thanks a lot for the detailed response! I consider Q6-10 resolved, for Q5 unfortunately I cant see the code on anonymous github (it might be an issue with the site itself). Q2 and Q3 I believe you can fix in a final version so Q1 and Q4 remain.
> >
> > Q4: The Text+ChemT5 results indeed seem inferior to what was reported in the original paper for the same tasks, which casts some doubt onto the training process. Maybe limiting comparison to the datasets shared between them and OmniMol would have been more faithful as you could simply use the existing model in inference mode (this is what we did in previous work). But overall, I acknowledge that you tested OmniMol on a wide variety of tasks and there is clear signal that the model works well and overall a bit better than previous methods.
> >
> > Q1: The limited methodological novelty is the most critical issue that I see. There exist generalist LLMs that can handle all these tasks. Even limiting to academic resources we find papers from few years ago that handled all those tasks (e.g., Text Chem T5), so the contribution of this work then boils down to the performance gain. Which is there but not with a large margin and also with a model that is much larger than competing specialist models. At the same time, the training dataset does not seem large enough to really enable generalization to unseen tasks or datasets, this is basically the same point that 8Wa4 raised. I'm not demanding this because this is extremely challenging to achieve and may even be impossible with standard academic resources. Moreover, there exist much larger datasets to develop chemistry LLMs, e.g., ChemPile. Altogether, this leaves me a bit conflicted what is the value OmniMol adds or the lesson we learn from it as community. This is why I scored 3 initially.

---

> > > ### Author Response · Authors · 2025-08-07
> > > **Response to Further Discussion (1)**
> > >
> > > Thank you for taking the time to discuss whether OmniMol is contributing to the advancement of the molecular large language model community. Our answer is a definite yes. Below, we provide a detailed discussion in response to your concerns.
> > >
> > > >Concern on methodological novelty
> > >
> > > Building a generalist model is not our primary contribution, and we fully acknowledge the existing achievements of the community in this area. Instead, this paper focuses on **addressing two major limitations** of current generalist frameworks.
> > >
> > > 1. **The more tasks there are, the greater the variation in difficulty, making it challenging to achieve balanced instruction-following across different tasks.**
> > >
> > > It is clear from your comments that you have conducted research on molecular LLMs. As you are certainly aware, different chemical tasks vary significantly in their learning difficulty. Our instruction-following dataset, the most comprehensive in academia to date, clearly demonstrates that such disparities in task difficulty severely undermine the **balance of instruction learning** across tasks.
> > >
> > > To address this fundamental challenge, OmniMol incorporates mechanisms such as **adaptation to intrinsic task rank** and **Mixture-of-Experts (MoE) expansion**, which effectively mitigate the imbalance. As a result, it achieves state-of-the-art performance across a wide range of tasks, outperforming all existing generalist LLMs, none of which are able to maintain such high performance as the number of tasks increases.
> > >
> > > We believe that this framework represents a significant advancement in the field of molecular AI.
> > >
> > > 2. **It is difficult to achieve both efficient and high-quality instruction-following capabilities simultaneously.**
> > >
> > > In main paper, the reported OmniMol achieves a **20%** performance improvement while using only approximately 30% of the parameters, which is made possible through a comprehensive pipeline involving meticulous data curation, pretraining, architectural analysis, and instruction tuning. Moreover, we observed that OmniMol exhibits remarkable scaling properties. When scaling up to a 4B-parameter MoE model, it delivers a 30% performance gain over previous state-of-the-art models. All of our models and code are **fully open-sourced**, offering substantial support to the community and facilitating further research in molecular LLMs.
> > >
> > > >Concerns on datasets.
> > >
> > > Our dataset is, to the best of our knowledge, the largest **molecular instruction-tuning dataset** in academia to date, comprising **2.8B tokens**.
> > >
> > > While you mentioned **ChemPile**, it was released after our submission. Moreover, the majority of its **73B tokens** were used solely for **pretraining purposes**. Among all the subsets of ChemPile, **only ChemPile-Reasoning (72.9K tokens)** has the potential for **instruction-following**, as it is the only one that contains explicit instruction–response pairs. The rest of the subsets lack a clear instruction–response format. That said, we appreciate your insightful comment and will include ChemPile in the related work section accordingly.
> > >
> > > For context, in the NLP domain, pretraining typically involves around **780B tokens**, whereas instruction tuning is often conducted with only **1.4B tokens** [1].
> > >
> > > Overall, we have carefully constructed and **rephrased** high-quality instructions, and our dataset is **publicly available**. In terms of **data scale**, **instruction quality**, and **training effectiveness**, our instruction-tuning dataset represents a significant contribution to molecular AI research, and lays a solid foundation for advancing instruction-tuned models in the molecular domain.
> > >
> > > [1] Chung H W, Hou L, Longpre S, et al. Scaling instruction-finetuned language models[J]. Journal of Machine Learning Research, 2024, 25(70): 1-53.

---

> > > > ### Author Response · Authors · 2025-08-07
> > > > **Response to Further Discussion (2)**
> > > >
> > > > In summary, our work provides several key contributions to the field of molecular LLMs:
> > > >
> > > > 1. **Comprehensive and High-Quality Instruction Dataset**
> > > >
> > > >     We construct the **largest and most comprehensive molecular instruction-tuning dataset** to date, consisting of 2.8B high-quality tokens. This dataset spans a wide range of molecular tasks and instruction types, and is carefully rephrased and curated to ensure both task diversity and linguistic clarity. Its public release sets a new standard for the community.
> > > >
> > > > 2. **Addressing Task Imbalance in Generalist Frameworks**
> > > >
> > > >     Unlike prior generalist frameworks that struggle with performance degradation across uneven task difficulties, **OmniMol** explicitly tackles this challenge by integrating **intrinsic task ranking** and **MoE expansion**. These strategies enable the model to adaptively allocate capacity and maintain high performance across diverse molecular tasks.
> > > >
> > > > 3. **Highly Efficient Model with Strong Scaling Behavior**
> > > >
> > > >     Through rigorous architecture search and optimization, our model achieves **state-of-the-art results** using only **30% of the parameters** of competing methods, demonstrating both efficiency and effectiveness. Furthermore, our **4B-parameter MoE variant** delivers an additional **30% improvement over SOTA**, revealing the model’s strong scaling potential.
> > > >
> > > > 4. **Open-Sourced Model Suite and Reproducibility**
> > > >
> > > >     All models, datasets, and training code are **fully open-sourced**, promoting reproducibility and lowering the barrier to entry for the broader research community. This provides a solid foundation for downstream applications and future benchmarking.
> > > >
> > > > 5. **Foundation for Future Generalist Molecular Agents**
> > > >
> > > >     By demonstrating how a single framework can unify tasks such as molecular property prediction, reaction generation, structure reasoning, and bioactivity estimation, **OmniMol lays the groundwork for the development of next-generation generalist molecular agents** capable of scientific reasoning and decision-making.

---

> > > > > ### Author Response · Authors · 2025-08-07
> > > > > **Response to Further Discussion (3)**
> > > > >
> > > > > >Concern on Text-chem T5
> > > > >
> > > > > Thanks for your suggestion. We utilize the Text-chem T5 base model on huggingface and test on part of the similar task types. Besides, we find that the reported performance of T5 in Mol-Instructions [2] is also quite poor. The detailed comparison is as follows. We can still see that it is far from satisfactory. We would like to emphasize that we have absolutely not cherry-picked any results. We are open to all forms of comparison.
> > > > >
> > > > > Forward
> > > > >
> > > > > | Methods | Exact | BLEU | LEVEN | MACCS | RDK | MORGAN | Val |
> > > > > | --- | --- | --- | --- | --- | --- | --- | --- |
> > > > > | T5 from Mol-Instruction | 0.24 | 0.782 | 20.413 | 0.789 | 0.705 | 0.652 | 0.762 |
> > > > > | T5 run by us | 0.67 | 0.869 | 9.16 | 0.910 | 0.755 | 0.76 | 0.913 |
> > > > > | Omni-Mol | 0.73 | 0.980 | 5.55 | 0.947 | 0.895 | 0.87 | 1 |
> > > > >
> > > > > Retrosynthesis
> > > > >
> > > > > | Methods | Exact | BLEU | LEVEN | MACCS | RDK | MORGAN | Val |
> > > > > | --- | --- | --- | --- | --- | --- | --- | --- |
> > > > > | T5 from Mol-Instruction | 0.141 | 0.765 | 24.043 | 0.765 | 0.685 | 0.59 | 0.698 |
> > > > > | T5 run by us | 0.026 | 0.474 | 43.38 | 0.744 | 0.752 | 0.54 | 0.742 |
> > > > > | Omni-Mol | 0.57 | 0.960 | 8.97 | 0.909 | 0.864 | 0.83 | 1 |
> > > > >
> > > > > Reagent
> > > > >
> > > > > | Methods | Exact | BLEU | LEVEN | MACCS | RDK | MORGAN | Val |
> > > > > | --- | --- | --- | --- | --- | --- | --- | --- |
> > > > > | T5 from Mol-Instruction | 0.00 | 0.225 | 49.323 | 0.186 | 0.039 | 0.05 | 0.313 |
> > > > > | T5 run by us | 0.00 | 0.077 | 58.89 | 0.198 | 0.042 | 0.05 | 0.564 |
> > > > > | Omni-Mol | 0.23 | 0.726 | 14.59 | 0.627 | 0.557 | 0.52 | 1 |
> > > > >
> > > > > Text Guided Molecule Generation
> > > > >
> > > > > | Methods | Exact | BLEU | LEVEN | MACCS | RDK | MORGAN | Val |
> > > > > | --- | --- | --- | --- | --- | --- | --- | --- |
> > > > > | T5 from Mol-Instruction | 0.09 | 0.508 | 41.81 | 0.474 | 0.352 | 0.353 | 0.721 |
> > > > > | T5 run by us | 0.06 | 0.446 | 37.65 | 0.529 | 0.483 | 0.385 | 0.867 |
> > > > > | Omni-Mol | 0.12 | 0.824 | 23.59 | 0.721 | 0.562 | 0.442 | 0.963 |
> > > > >
> > > > > Finally, we sincerely thank you once again for taking the time to engage in this discussion. Your feedback has been instrumental in helping us continuously refine our ideas and improve our work. We warmly welcome any further questions or suggestions you may have.
> > > > >
> > > > > [2] Fang Y, Liang X, Zhang N, et al. Mol-instructions: A large-scale biomolecular instruction dataset for large language models[J]. ICLR 2024.

---

> > > > > > ### Comment · Reviewer_FtzT · 2025-08-07
> > > > > > **Score raise**
> > > > > >
> > > > > > While I still believe that the methodological novelty is relatively low and OmniMol does not really unlock new capabilities of LLMs in chemistry (it's a crowded field and it's hard to make a real delta), the rebuttal helped me to understand better the richness of the various datasets and the effort in homogenizing and providing this data to the community. Moreover, the scale of the model is significant (considering standard academic resources) and performance gains to previous methods seem robust. Therefore, I increased my score from 3 to 4.

---

> > > > > > > ### Comment · Reviewer_FtzT · 2025-08-07
> > > > > > >
> > > > > > > I also raised significance from 2 to 3

---

> > > > > > > > ### Author Response · Authors · 2025-08-07
> > > > > > > > **Official Comment by Authors**
> > > > > > > >
> > > > > > > > Thank you very much for your recognition and professional review comments! We have learned a great deal from your suggestions and the discussion, and we will revise the paper accordingly. We are also committed to making greater contributions to the field (e.g., further enhance generalization ability).
> > > > > > > >
> > > > > > > > Once again, we sincerely appreciate the time and effort you devoted to the review. It is both an honor and a pleasure to have a reviewer like you!

---

### Official Review · Reviewer_8Wa4 · 2025-06-23

**Clarity:** 1
**Significance:** 2
**Originality:** 1
**Rating:** 3
**Confidence:** 5

**Summary:**

This paper aims to build a general-purpose, multi-task molecular model based on a large language model. To this end, the authors mainly focus on and propose the following:
- A multi-task dataset of 16 molecular tasks collected from public databases by classifying them into four folds: Mol2Mol, Mol2Text, Mol2Num, and Text2Mol.
- Gradient adaptive LoRA (GAL), which dynamically adjusts the scaling factor with learnable parameters to enhance the multi-task learning.
- Mixture-of-GAL-Experts, a MoE architecture with the proposed GALs.

The resulting model, Omni-Mol, is evaluated on the collected multi-task dataset, outperforming the baselines.

**Questions:**

- How important is clipping in GLA? What happens if the clipping is removed?
- Why are other tasks, such as molecular classification, not considered in this work?
- Could the authors provide the experimental results on the classification tasks without fine-tuning?
- What do the left and right y-axes denote in Figure 5 (Middle) and (Right)?

**Ethical Concerns:**

["NO or VERY MINOR ethics concerns only"]

**Final Justification:**

I acknowledge that SELFIES is beneficial in certain cases, that the training of the multimodal projector is reasonable, that the proposed model architecture (GAL) is effective in addressing the targeted points, and that the proposed Omni-Mol model is robust to permutation.

However, I still have concerns about the following points:

Originality of the proposed dataset: While the authors have made efforts in curating the dataset, including adding instructions, unification, and de-duplication, it is primarily composed of existing curated datasets, not introducing a truly novel task.
Generalizability: Although the proposed model is designed for generalizability, its performance is moderate on OOD tasks.
Overall, I raise my score accordingly.

**Limitations:**

The main limitation of this work is that the proposed model, Omni-Mol, does not show its ability to generalize to tasks beyond the training dataset. I highly recommend evaluating the proposed model on other benchmarks. Additionally, the design of the molecular representations is not reasonable, leaving critical problems such as the permutation equivariance of graph-structured data and diverse forms of string representations.

**Quality:**

1

**Strengths And Weaknesses:**

**Strengths**
Leveraging MoE architectures is beneficial for improving performance.
- In the ablation study, GAL demonstrates performance improvement, indicating its effectiveness in task learning.

**Weaknesses**
- Omni-Mol is not evaluated on the general tasks beyond the collected dataset. In other words, the proposed model does not exhibit its generalization ability to diverse tasks. Even though the main motivation of this paper is to build a general molecular LLM, the experimental results are limited to showing its effectiveness on its own dataset, not addressing the tackled problem.
- The proposed method does not guarantee the permutation equivariance of graph-structured data as it directly forwards the node embeddings of graph representations to LLMs. Therefore, if the nodes are permuted, the responses of Omni-Mol will be significantly changed, often generating irrelevant results.
- As LLMs better understand SMILES than SELFIES, leveraging SELFIES does not fully utilize the internal knowledge in LLMs, which could show inferior performance compared to leveraging SMILES. Even though SELFIES is beneficial to generate valid molecules, using SMILES brings more practicality by fully leveraging the internal knowledge of LLMs and not degrading the original LLMs' abilities.
- SMILES and SELFIES are not unique, as they could have diverse forms. Specifically, SMILES has diverse forms [1], and as SELFIES is derived from the ordering of SMILES, SELFIES is also not unique. Therefore, using SELFIES could lead misunderstanding of molecular modalities.
- It is not clear why the multimodal projector is trained only on PubChem. To achieve multi-task learning, training on all the collected datasets sounds more reasonable.
- The motivation of GLA is not well-established. Motivation experiment for GLA shows that the optimal rank in LoRA could be different according to the tasks. However, it is not directly related to the dynamic scaling factor, as the rank is still static.
- Related works [2,3,4] are not experimentally compared.
- The proposed dataset is a collection of well-known instruction datasets, of which the novelty is limited.



[1] Ganeeva, Veronika, et al. "Chemical language models have problems with chemistry: A case study on molecule captioning task." The Second Tiny Papers Track at ICLR, 2024.

[2] Park, Jinyoung, et al. "LLaMo: Large Language Model-based Molecular Graph Assistant." NeurIPS, 2024.

[3] Zhang, Juzheng, et al. "UniMoT: Unified molecule-text language model with discrete token representation." ArXiv, 2024.

[4] Kim, Dongki, et al. "Mol-LLaMA: Towards general understanding of molecules in large molecular language model." ArXiv, 2025.

---

> ### Author Rebuttal · Authors · 2025-07-31
>
> > Q1: No evaluation on generalization ability to diverse tasks.
>
> A1: Thanks for the comment, the main motivation of Omni-Mol is to train a single model capable of solving multiple tasks, in contrast of using dedicated adapters for different tasks. However, we agree with you that generalizing to OOD data is indeed an interesting aspect. Thus, we directly implement the zero-shot evaluation on three different classification datasets. Please refer to Reviewer fT5E A4.
>
> > Q2: Doubt on permutation equivariance.
>
> A2: Thanks for the comment. First, the graph features obtained from the GNN encoder are aligned with the textual input. The graph-based model we adopt, *MoleculeSTM [1]*, has undergone extensive pretraining for graph-text alignment, enabling it to provide the LLM with sufficient structural information from the molecular graph. Therefore, permuting the node order does not significantly impact performance. However, to provide more rigorous evidence, we shuffle the order of the nodes and evaluated the performance of Omni-Mol. This was done by applying random permutation to the graph data. The experimental results are as follows:
>
> Forward
> ||Exact|BLEU|LEVEN|MACCS|RDK|MORGAN|Val|
> |-|-|-|-|-|-|-|-|
> |Permut.|0.726|0.9817|5.595|0.9482|0.8942|0.8683|1|
> |noPermut.|0.73|0.98|5.55|0.947|0.895|0.87|1|
>
> Reagent
> ||Exact|BLEU|LEVEN|MACCS|RDK|MORGAN|Val|
> |-|-|-|-|-|-|-|-|
> |Permut.|0.23|0.7131|14.822|0.6265|0.556|0.519|0.99|
> |noPermut.|0.23|0.726|14.59|0.627|0.557|0.52|1|
>
> Retrosynthesis
> ||Exact|BLEU|LEVEN|MACCS|RDK|MORGAN|Val|
> |-|-|-|-|-|-|-|-|
> |Permut.|0.567|0.959|9.075|0.908|0.862|0.8284|1|
> |noPermut.|0.57|0.96|8.97|0.909|0.864|0.83|1|
>
> Solvent
> ||Exact|BLEU|LEVEN|MACCS|RDK|MORGAN|Val|
> |-|-|-|-|-|-|-|-|
> |Permut.|0.517|0.756|2.72|0.670|0.671|0.639|1|
> |noPermut.|0.52|0.759|2.71|0.671|0.673|0.64|1|
>
> Catalyst
> ||Exact|BLEU|LEVEN|MACCS|RDK|MORGAN|Val|
> |-|-|-|-|-|-|-|-|
> |Permut.|0.714|0.782|1.9731|0.889|0.906|0.714|1|
> |noPermut.|0.72|0.792|1.96|0.886|0.904|0.72|1|
>
> Mol2Num
> ||LOGP|TPSA|WEIGHT|
> |-|-|-|-|
> |Permut.|0.503|11.09|11.357|
> |noPermut.|0.49|11.07|11.07|
>
> Quantum Mechanics Property Prediction
> ||HOMO|LUMO|GAP|AVG|
> |-|-|-|-|-|
> |Permut.|0.0038|0.0057|0.0049|0.0048|
> |noPermut.|0.0038|0.0047|0.0049|0.0044|
>
> Yield
> ||BH|SM|
> |-|-|-|
> |Permut.|0.8808|0.606|
> |noPermut.|0.94|0.68|
>
> Experiment
> ||BLEU-2|BLEU-4|ROUGE-1|ROUGE-2|ROUGE-L|
> |-|-|-|-|-|-|
> |Permut.|0.573|0.449|0.532|0.274|0.465|
> |noPermut.|0.572|0.448|0.532|0.274|0.464|
>
> Molcap
> ||BLEU-2|BLEU-4|METEOR|ROUGE-1|ROUGE-2|ROUGE-L|
> |-|-|-|-|-|-|-|
> |Permut.|0.493|0.44|0.545|0.602|0.435|0.538|
> |noPermut.|0.529|0.44|0.571|0.604|0.447|0.541|
>
> TextGuidedMolGen
> ||Exact|BLEU|LEVEN|MACCS|RDK|MORGAN|Val|
> |-|-|-|-|-|-|-|-|
> |Permut.|0.119|0.820|23.108|0.719|0.559|0.439|0.96|
> |noPermut.|0.12|0.824|24.179|0.721|0.562|0.442|0.963|
>
> We can draw the conclusion that Omni-Mol is robust to permutation equivariance.
>
> > Q3: Utilization of the internal knowledge in LLMs.
>
> A3: Thanks for the comment. The motivation of using SELFIES is to allow higher success rate when converting the 1D representation into 2D graph and 3D graph. There are a line of works using SELFIES as their 1D molecule representation, as suggested by PRESTO [2], we use SELFIES for its broader applicability. In fact, many of our baseline results are based on SMILES, meaning they fully leverage the pretrained knowledge embedded within the LLM. In contrast, our method achieves SOTA performance without relying on such pretrained domain knowledge, which further highlights the effectiveness of Omni-Mol.
>
> > Q4: SELFIES could lead misunderstanding.
>
> A4: Thanks for the comment. You’ve made a very insightful point, there is no universally perfect molecular representation. We chose SELFIES primarily due to its higher robustness. In fact, we conduct empirical comparisons and observe that SMILES often leads to parsing failures for a significant number of samples.
>
> Number of failed cases
> |Method|Forward|Retrosynthesis|Reagent|Solvent|
> |-|-|-|-|-|
> |SELFIES|0|0|0|0|
> |SMILES|10|16|2|6|
>
> The experimental results are shown below, which further justifies our decision to use SELFIES.
>
> > Q5: Multimodal projector training.
>
> A5: Thanks for the comment. The projector is pre-trained on PubChem for early alignment between LLM and the graph features. In the second stage, projector is still activated and can be updated by multi-task training signals.
>
> > Q6: Doubt on the dynamic scaling factor, as the rank is still static.
>
> A6: Thanks for the comment. The rank in LoRA adapters is fixed by design to enable efficient fine-tuning. Our solution is innovative in that it introduces a learnable scaling factor, calculated as $\alpha / r$, which effectively links the scaling factor to the adapter’s rank. When the intrinsic rank varies across tasks, GAL addresses this issue by automatically scaling the learning weights under specific lora rank, thereby achieving better multi-task balance.
>
> > Q7: Related works [3, 4, 5] are not experimentally compared.
>
> A7: Thanks for the comment, we have cited all the related works to our paper. Due to time limit, we re-implement Mol-Llama as one additional baseline, we noticed that the 3D GNN used by Mol-Llama supports single molecule input only, we selected tasks that involve only one molecule as input, the results are shown below
>
> DQA
> |Method|BLEU-2|BLEU-4|METEOR|ROUGE-1|ROUGE-2|ROUGE-L|
> |-|-|-|-|-|-|-|
> |Omni-Mol|0.552|0.440|0.58|0.53|0.38|0.49|
> |Mol-Llama|0.507|0.428|0.574|0.522|0.374|0.484|
>
> Quantum Mechanics
> |Method|HOMO|LUMO|GAP|AVG|
> |-|-|-|-|-|
> |Omni-Mol|0.0038|0.0047|0.0049|0.0044|
> |Mol-Llama|0.0049|0.0050|0.0049|0.0049|
>
> Mol2Num
> |Method|LogP|TPSA|Weight|
> |-|-|-|-|
> |Omni-Mol|0.49|5.89|11.07|
> |Mol-Llama|0.518|5.914|10.591|
>
> Molcap
> |Method|BLEU-2|BLEU-4|METEOR|ROUGE-1|ROUGE-2|ROUGE-L|
> |-|-|-|-|-|-|-|
> |Omni-Mol|0.529|0.440|0.571|0.604|0.447|0.541|
> |Mol-Llama|0.488|0.396|0.528|0.573|0.408|0.509|
>
> Omni-Mol slightly falls short on several Mol2Num tasks but exhibits significantly better performance on other tasks, which fully demonstrates the superiority of Omni-Mol as a general-purpose molecular LLM.
>
> > Q8: Doubt on novelty of datasets.
>
> A8: Thanks for the comment. Our dataset makes a significant contribution to the field in several key aspects: **First**, we collect the most comprehensive set of molecular tasks, augmenting instruction-missing data with carefully constructed prompts, and further enhancing all tasks with diverse instructions. **Second**, we conduct thorough data leakage checks across the entire dataset and remove duplicate samples to build the largest general-purpose molecular dataset to date. **Third**, our dataset covers the broadest and most complete range of task types, laying a solid foundation for the development of general-purpose molecular AI models.
>
> > Q9: Ablation on clipping in GLA.
>
> A9: Thanks for the comment. Clipping is very important in our training. We have added the ablation study results on clipping below. Due to the time limit, we conduct the experiment on a subset of tasks.
>
> Catalyst
> |Model|Exact|BLEU|RDK|MACCS|Morgan|Leven|val|
> |-|-|-|-|-|-|-|-|
> |Omni-Mol|0.742|0.794|0.911|0.899|0.747|1.843|1|
> |w/o clip|0.731|0.775|0.898|0.878|0.737|2.144|1|
>
> Forward
> |Model|Exact|BLEU|RDK|MACCS|Morgan|Leven|val|
> |-|-|-|-|-|-|-|-|
> |Omni-Mol|0.738|0.985|0.884|0.941|0.865|6.06|1|
> |w/o clip|0.703|0.980|0.865|0.927|0.840|7.045|1|
>
> Quantum Mechanics
> |Method|HOMO|LUMO|GAP|AVG|
> |-|-|-|-|-|
> |Omni-Mol|0.0038|0.0047|0.0049|0.0044|
> |w/o clip|0.0039|0.0055|0.0047|0.0047|
>
> Molcap
> |Method|BLEU-2|BLEU-4|METEOR|ROUGE-1|ROUGE-2|ROUGE-L|
> |-|-|-|-|-|-|-|
> |Omni-Mol|0.511|0.421|0.556|0.593|0.434|0.530|
> |w/o clip|0.483|0.391|0.526|0.570|0.406|0.507|
>
> Reagent
> |Method|Exact|BLEU|RDK|MACCS|Morgan|Leven|val|
> |-|-|-|-|-|-|-|-|
> |Omni-Mol|0.266|0.749|0.586|0.651|0.542|14.026|1|
> |w/o clip|0.256|0.736|0.577|0.645|0.533|14.619|1|
>
> Retrosynthesis
> |Method|Exact|BLEU|RDK|MACCS|Morgan|Leven|val|
> |-|-|-|-|-|-|-|-|
> |Omni-Mol|0.594|0.962|0.861|0.910|0.828|8.386|1|
> |w/o clip|0.542|0.959|0.838|0.897|0.802|9.937|1|
>
> Solvent
> |Method|Exact|BLEU|RDK|MACCS|Morgan|Leven|val|
> |-|-|-|-|-|-|-|-|
> |Omni-Mol|0.554|0.782|0.703|0.700|0.678|2.498|1|
> |w/o clip|0.550|0.775|0.692|0.693|0.667|2.550|1|
>
> Yield
> |Method|BH|SM|
> |-|-|-|
> |Omni-Mol|0.953|0.688|
> |w/o clip|0.933|0.680|
>
> From the results, we observe performance drops across several tasks, which aligns with our assumption that different tasks have different optimal rankings.
>
> > Q10: Why are other tasks, such as molecular classification, not considered in this work?
>
> A10: Thanks for the comment. First, the DQA task includes approximately 10k entries related to molecular graph classification. For example, there is a sample with instruction: “What is dysidenin classified as chemically?”, and output: “Dysidenin is classified as an organochlorine compound.” Therefore, strictly speaking, we categorize it under the text2text setting.
>
> > Q12: What do the left and right y-axes denote in Figure 5 (Middle) and (Right)?
>
> A12: Thanks for the comment. The left y-axis displays average performance excluding yield prediction, while the right y-axis shows yield prediction performance. These were separated due to substantial scale differences that would impair visualization.
>
> [1] Liu, S., Nie, W., Wang, C., Lu, J., Qiao, Z., Liu, L., Tang, J., Xiao, C. and Anandkumar, A., 2023. Multi-modal molecule structure–text model for text-based retrieval and editing. Nature Machine Intelligence, 5(12), pp.1447-1457.
>
> [2] Cao, H., Shao, Y., Liu, Z., Liu, Z., Tang, X., Yao, Y. and Li, Y., 2024. PRESTO: Progressive pretraining enhances synthetic chemistry outcomes. arXiv preprint arXiv:2406.13193.
>
> [3] Park, Jinyoung, et al. "LLaMo: Large Language Model-based Molecular Graph Assistant." NeurIPS, 2024.
>
> [4] Zhang, Juzheng, et al. "UniMoT: Unified molecule-text language model with discrete token representation." ArXiv, 2024.
>
> [5] Kim, Dongki, et al. "Mol-LLaMA: Towards general understanding of molecules in large molecular language model." ArXiv, 2025.

---

> > ### Author Response · Authors · 2025-08-02
> > **Supplementary Response to Reviewer 8Wa4**
> >
> > Thank you very much for your valuable and encouraging feedback on our work. We truly appreciate your recognition. From your review, it is evident that you have developed a deep and accurate understanding of Bio LLM.
> >
> > > Your first concern pertains to the generalization ability of our model. Due to space constraints, we were unable to provide a full response in the main text. Here is the main text:  Thanks for the comment. Generalizing to out-of-domain data is indeed an interesting aspect. Coincidentally, one of the reviewers pointed out that we did not fully consider the molecular graph classification task. This gives us a great opportunity to use it as a testbed to evaluate the generalization capability of our model. We directly implement the zero-shot evaluation on three different classification datasets [1, 2]. Since the original pretrained weights of many baseline models are not publicly available, we use Qwen3, which represents the state-of-the-art on scientific tasks, as our primary comparison. As the experimental results show, Omni-Mol demonstrates certain performance on completely unseen classification tasks, slightly outperforming Qwen3 in some tasks and metrics.
> > ### BBBP Dataset
> >
> > | Model    | F1 Score | Accuracy |
> > |----------|----------|----------|
> > | Omni-Mol | 0.5725   | 0.4363   |
> > | Qwen3    | 0.0617   | 0.2304   |
> >
> > ### HIV Dataset
> >
> > | Model    | F1 Score | Accuracy |
> > |----------|----------|----------|
> > | Omni-Mol | 0.0254   | 0.0650   |
> > | Qwen3    | 0.0000   | 0.0338   |
> >
> > ### SIDER Dataset
> >
> > | Model    | F1 Score | Accuracy |
> > |----------|----------|----------|
> > | Omni-Mol | 0.4054   | 0.3846   |
> > | Qwen3    | 0.1553   | 0.4891   |
> >
> > > Your second concern is that the use of molecular representations may be unreasonable. We chose SELFIES primarily due to its greater robustness. In our empirical comparisons, we observe that SMILES often results in parsing failures for a significant number of samples (results in A4), which is critical when building a **general-purpose molecular model**. If parsing fails, users cannot obtain any feedback. Moreover, SELFIES has been adopted in many peer-reviewed studies, lending further support to its validity. Our experimental results also demonstrate that, in terms of molecular representation, our approach does not exhibit any clear deficiencies compared to existing molecular LLMs.
> >
> > Thank you for taking the time to help us improve the quality of our manuscript. If you have any further suggestions, we would be more than happy to engage in further discussion with you.

---

> > > ### Author Response · Authors · 2025-08-05
> > > **Official Comment by Authors**
> > >
> > > Dear Reviewer 8Wa4,
> > >
> > > Thank you for providing such detailed reviews. We deeply appreciate the time and effort you have invested in reviewing our work and offering these detailed comments.
> > >
> > > We have conducted additional experiments to address your concerns regarding task generalization, equivariant generalization, and molecular representation, and have provided thorough justification for other related details. If you feel there are any aspects that **require further clarification or improvement**, we are more than willing to provide additional information or clarifications during the discussion period to assist with your review process.
> > >
> > > Thank you again for your time and expertise. Your feedback has been instrumental in enhancing the quality of our work.

---

> ### Author Response · Authors · 2025-08-06
>
> Dear Reviewer 8Wa4,
>
> Thank you for your insightful and constructive reviews. We would like to kindly remind you that the discussion period will conclude in three days. In the meantime, we have provided detailed supplementary experiments and clarifications addressing your concerns.
>
> If you have any additional questions or would like us to conduct further experiments, **please do not hesitate to let us know**, we would be more than happy to engage in further discussion and provide additional details about our work.
>
> Thank you once again for your time and expertise. We look forward to hearing from you.
>
> Sincerely,
> Submission 1411 Authors

---

> > ### Comment · Reviewer_8Wa4 · 2025-08-07
> >
> > I sincerely appreciate the thoughtful responses from the authors and thank you for the clarifications.
> >
> > > Q1: No evaluation on generalization ability to diverse tasks
> >
> > I am concerned that the baselines are limited to Qwen3. Could you compare Omni-Mol with other general and molecular LLMs such as LLaMA, GPT, LLaMo [2], and UniMoT [3]? Additionally, could you report results on other datasets from MoleculeNet?
> >
> > > Q3 & Q4: Usage of SELFIES
> >
> > I understand that SELFIES is beneficial for generative tasks such as retrosynthesis, reagent prediction, and forward reaction prediction. However, it might not be appropriate for other tasks such as description, captioning, or property prediction. Could you compare Omni-Mol to a model with the same architecture trained on a captioning task for more epochs?
> >
> > Additionally, I am concerned that, although some baselines use SMILES, their base model (LLaMA 2) is inferior to the one used in this work (LLaMA 3.2). The outperformance of the proposed model could come from the difference of knowledge and capabilities of the base model. I suggest evaluating models using the same base model to ensure a fair comparison.
> >
> > > Q7: Additional Comparison
> >
> > Thank you for providing further experimental results. The experimental results are slightly different from the original paper. Could you elaborate the experimental details?
> >
> >
> > > Q8: Novelty of Dataset
> >
> > In my opinion, the proposed dataset appears to be a compilation of previously curated datasets. Many of these existing datasets already include prompts and cover diverse molecular aspects.
> >
> > ---
> >
> > Finally, I have a general concern about whether the proposed model outperforms specialized models. In scientific fields, accuracy is often prioritized compared to generality. While I acknowledge the outperformance of the proposed model for tasks requiring textual outputs, could you compare with specialized models such as GNNs or transformers?

---

> > > ### Author Response · Authors · 2025-08-08
> > > **Response to Further Discussion (1)**
> > >
> > > Thank you very much for your thorough concerns. We are fully confident that we can address them.
> > >
> > > > Additional Q1: No evaluation on generalization ability to diverse tasks
> > > >
> > >
> > > A1: Thanks for the comment. Qwen3 is the most powerful open-source model available, which is why we chose to use it. That said, we are absolutely open to expanding the scope of our experimental comparisons.
> > >
> > > For OOD evaluation datasets, actually, we have already tested on 4 classification datasets from MoleculeNet (BACE, BBB, HIV, SIDER). The results are shown in  Supplementary Response above. But we are very happy to expand to more evaluation cases including regression tasks (clintox, esol, freesolv, lipophilicity). Based on the existing classification and regression tasks, spanning different molecular properties and data complexities, we can confidently validate the model’s out-of-distribution generalization capability.
> > >
> > > Specifically, since UniMoT has no public code, we conduct OOD evaluations on GPT-4o, LLaMA 4 (17B-128E), Qwen 3 (30B-A3B), and LLaMo. GPT-4o and Qwen 3 are accessed directly via API calls, while LLaMA 4 and LLaMo are run on an 8×H100 machine using the vLLM framework. For LLaMo, we utilize the officially released ``lamo_checkpoint.ckpt'' for evaluation. We have to admit that completing such a large number of experiments in less than two days was an extremely intense and demanding task!
> > >
> > > The results are shown below. From the experimental results, we can see that our model achieves leading performance on many tasks compared with the most powerful commercial and open-source models worldwide. This clearly demonstrates that our work makes a highly significant contribution to the molecular LLM community.
> > >
> > > The results of LLaMo are coming soon (**within 4 hours**). We took a lot of time to re-generate the data for LLaMo’s codebase due to its old version of processing data.
> > >
> > > Table1: BBBP
> > >
> > > | Method | F1 | ACC |
> > > | --- | --- | --- |
> > > | DeepSeekV3 | 0.0000 | 0.2304 |
> > > | Qwen3 | 0.0617 | 0.2303 |
> > > | Llama4 | 0.5702 | 0.5833 |
> > > | Omnimol | 0.5725 | 0.4363 |
> > >
> > > Table2: HIV
> > >
> > > | Method | F1 | ACC |
> > > | --- | --- | --- |
> > > | DeepSeekV3 | 0.0655 | 0.0346 |
> > > | Qwen3 | 0.0000 | 0.0338 |
> > > | Llama4 | 0.0566 | 0.0528 |
> > > | Omnimol | 0.0254 | 0.0650 |
> > >
> > > Table3: SIDER
> > >
> > > | Method | F1 | ACC |
> > > | --- | --- | --- |
> > > | DeepSeekV3 | 0.2913 | 0.4895 |
> > > | Qwen3 | 0.1553 | 0.4891 |
> > > | Llama4 | 0.7040 | 0.5594 |
> > > | Omnimol | 0.4054 | 0.3846 |
> > >
> > > Table4: CLINTOX
> > >
> > > | Method | F1 | ACC |
> > > | --- | --- | --- |
> > > | DeepSeekV3 | 0.0000 | 0.4763 |
> > > | Qwen3 | 0.6696 | 0.5033 |
> > > | Llama4 | 0.5080 | 0.3682 |
> > > | Omnimol | 0.5490 | 0.2973 |
> > >
> > > Table5: ESOL
> > >
> > > | Method | MAE |
> > > | --- | --- |
> > > | DeepSeekV3 | 2.0342 |
> > > | Qwen3 | 3.6416 |
> > > | Llama4 | 1.5639 |
> > > | Omnimol | 1.2917 |
> > >
> > > Table 6: Lipophilicity
> > >
> > > | Method | MAE |
> > > | --- | --- |
> > > | DeepSeekV3 | 0.9842 |
> > > | Qwen3 | 3.4005 |
> > > | Llama4 | 1.9060 |
> > > | Omnimol | 1.1690 |
> > >
> > > All our evaluation code is open-sourced in our codebase for reference.
> > >
> > > > Additional Q2: Usage of SELFIES
> > > >
> > >
> > > A2: Thanks for the comment. SELFIES is widely used by InstructMol (COLING 2025), PRESTO (ACL 2025)  on description, captioning, and property tasks. Besides, we have additionally run a quick ablation study on MolCap, which shares the same architecture with OmniMol and is trained for 12 epochs. The results below confirm that when we only train on Molcap, it can achieve similar even better than baselines with both SELFIES and SMILES. That said, SELFIES is also suitable for description tasks.
> > >
> > > | Methods | BLEU-2 | BLEU-4 | METEOR | ROUGE-1 | ROUGE-2 | ROUGE-L |
> > > | --- | --- | --- | --- | --- | --- | --- |
> > > | InstructMol (SELFIES) | 0.475 | 0.371 | 0.509 | 0.566  | 0.394  | 0.502  |
> > > | HIGHT (SMILES) | 0.498  | 0.397  | 0.525 | 0.582 | 0.414 | 0.518 |
> > > | OmniMol only on Molcap | 0.480 | 0.388 | 0.532 | 0.571 | 0.411 | 0.510 |
> > > | OmniMol | 0.529 | 0.440 | 0.541 | 0.604 | 0.447 | 0.571 |
> > >
> > > > Additional Q3: Ablation on model backbone
> > > >
> > >
> > > A3: This ablation study has already been implemented in **Appendix F.1 Ablation on Language Backbone** and shown in Table 10. This ensures that our conclusions are not contingent on using newer backbones, and the advantage holds across different LLM families.
> > >
> > > We are eager to reiterate our experimental process and findings. The baselines you mentioned,  InstructMol and PRESTO, all utilize **Vicuna 7B**. Notably, Vicuna 7B was released in March 2023, predating the release of LLaMA 2 you mentioned in July 2023.
> > >
> > > To ensure that our observed performance gains are not merely attributed to newer language backbones such as LLaMA 3.1 and LLaMA 3.2, we replaced OmniMol’s language model with Vicuna 7B and conducted experiments across several tasks. As demonstrated in Table 10, even with a larger parameter size, **Vicuna 7B performs significantly better than the LLaMA 3 1B model**.

---

> > > > ### Author Response · Authors · 2025-08-08
> > > > **Response to Further Discussion (2)**
> > > >
> > > > > Additional Q4: Elaboration on experimental details
> > > > >
> > > >
> > > > A4: Thanks for the comment. The reported results of OmniMol are totally the same with our paper. The results of Mol-llama are from our re-implemented results. Due to the smaller task coverage of Mol-llama, we re-implement it based on the open-source code and re-train it on our datasets. To ensure the fair comparison, we unify the LLM backbone with Llama 3 1B. All comparisons are conducted with the same random seed, same training recipe and identical data splits for fair comparison. We also release the complete training pipeline, logs, and configuration files to ensure full reproducibility.
> > > >
> > > > > Additional Q5: Novelty of datasets
> > > > >
> > > >
> > > > A5: Our dataset is, to the best of our knowledge, the largest **molecular instruction-tuning dataset** in academia to date, comprising **2.8B tokens**. For context, in the NLP domain, pretraining typically involves around **780B tokens**, whereas instruction tuning is often conducted with only **1.4B tokens** [1].
> > > >
> > > > Specifically, datasets such as **MolCap** and **Description QA** exhibit a high degree of similarity, which raises the risk of data leakage, for example, a sample from the QA test set might also appear in the training split of Description. To address this, we conducted a **comprehensive, instance-level comparison** between the test sets and training sets of both datasets. We then **systematically removed every single training sample** that appeared in the test set of either dataset, ensuring a clean separation between training and evaluation data.
> > > >
> > > > For classification tasks, we leveraged **GPT-4o** to **rephrase instructions in diverse ways**, thereby enriching prompt variation and enabling the model to generalize to a broader spectrum of natural language queries rather than overfitting to fixed prompt templates.
> > > >
> > > > In addition, for tasks such as **Molecule Editing**, we are the **first** to convert them into a fully-fledged **instruction-format dataset**, carefully designing task descriptions, input–output pairs, and edge cases to make them compatible with instruction-tuned models.
> > > >
> > > > This process was **far from an “easy compilation” of previously curated datasets**. Instead, it involved **labor-intensive curation, cross-dataset de-duplication, instruction engineering, and format unification** across heterogeneous molecular tasks. Constructing such a dataset required integrating heterogeneous molecular sources, rigorous deduplication across train and evaluation splits, and substantial manual curation to ensure instruction quality and diversity, tasks that are prohibitively time-consuming without domain expertise and computational resources. The resulting dataset is the product of **extensive manual and automated validation**, ensuring both **high quality** and **strong generalization potential** for downstream applications.
> > > >
> > > > Overall, we have carefully constructed and **rephrased** high-quality instructions, and our dataset is **publicly available**. In terms of **data scale**, **instruction quality**, and **training effectiveness**, our instruction-tuning dataset represents a significant contribution to molecular AI research, and lays a solid foundation for advancing instruction-tuned models in the molecular domain.
> > > >
> > > > > Additional Q6: Comparisons with specialized models
> > > > >
> > > >
> > > > A6: Thank you for your highly professional and insightful comments. In fact, we have already provided a detailed comparison with specialized models in the appendix.
> > > >
> > > > Please refer to **Appendix E.3:** Comparison on Mol2Num Tasks with GNN, where we re-implemented GraphMVP as our baseline and re-trained it on tasks such as quantum mechanics property prediction, LogP prediction, molecular weight prediction, and TPSA prediction.
> > > >
> > > > As shown in **Table 8**, Omni-Mol significantly outperforms traditional GNN models like GraphMVP, achieving up to a **91%** improvement on tasks such as molecular weight prediction.
> > > >
> > > > [1] Chung H W, Hou L, Longpre S, et al. Scaling instruction-finetuned language models[J]. Journal of Machine Learning Research, 2024, 25(70): 1-53.
> > > >
> > > > ---
> > > >
> > > > Once again, thank you for your highly professional feedback. **We will incorporate the related work and supplementary experiments into the next version, cite them appropriately, and further refine the discussion and comparisons.**
> > > >
> > > > We look forward to hearing from you!

---

> ### Author Response · Authors · 2025-08-09
> **Response to Further Discussion (3)**
>
> The OOD performance of LLaMo is coming!
>
> >Concern on OOD generalization ability comparison with LLaMo
>
> Thanks for the comment. We re-implement LLaMo and utilize the officially released `lamo_checkpoint.ckpt` for evaluation. To begin with the conclusion: on classification tasks, it completely fails to generalize, producing a chaotic mix of outputs including numbers, SMILES strings, text, and more. Its OOD generalization ability on regression tasks is also extremely poor.
>
> You may question the fairness of our comparison. However, we have made the entire reproduction process fully transparent — including the reproduction code, datasets, logs, and results — all of which have been uploaded to our codebase. You can directly find the reproduction logs under `llamo-reimplement/all_checkpoints`, the complete MoleculeNet datasets under `llamo-reimplement/bbbp`, `llamo-reimplement/esol`, and `llamo-reimplement/lipophilicity`, and the environment setup as well as file configurations for the reproduction code in `llamo-reimplement/README.md`. The process is absolutely reliable.
>
> Specifically, it has no OOD generalization ability on BBBP, a classification task from MoleculeNet. Here is part of the results:
>
> ```markdown
> {"prediction": "0.024", "target": "True"}
> {"prediction": "2-Methyl-4-phenyl-1,3-benzodioxole", "target": "False"}
> {"prediction": "0.035", "target": "True"}
> {"prediction": "The molecule is a member of the class of ethylxanthenes that is ethylxanthene in which the hydrogen at position 4 is replaced by an ethyl group. It is a member of the class of ethylxanthenes, an aromatic ether and an organic cation.", "target": "False"}
> {"prediction": "The molecule is a metabolite found in or produced by Saccharomyces cerevisiae.", "target": "True"}
> {"prediction": "0.038", "target": "True"}
> {"prediction": "CC(=O)N1CCN(C(=O)CC2=CC(F)=CC(F)=C2)C(CN3CCC(O)C3)C1", "target": "False"}
> {"prediction": "The molecule is a metabolite found in or produced by Saccharomyces cerevisiae.", "target": "False"}
> {"prediction": "The molecule is a natural product found in Solanum tuberosum and Triticum aestivum with data available.", "target": "True"}
> ```
>  To ensure the fairness, we even modify the instruction with more detailed information, such as "only output the true or false as the response". This result is shown in `llamo-reimplement/all_checkpoints/desc_bbbp_output/lightning_logs/version_1`.
>
> We can clearly see that LLaMo just outputs a chaotic mix of outputs including numbers, SMILES strings, text, and more. More details are in `llamo-reimplement/all_checkpoints` .
>
> ---
>
> As for regression, its outputs still contain 3 mixed samples in ESOL and 24 in Lipophilicity, so we have to remove these noisy entries in order to calculate the MAE. The results are shown below. We can obviously see that the OOD performance is worse than OmniMol.
>
> Table 1: OOD results on ESOL
>
> | Method | MAE |
> | --- | --- |
> | DeepSeekV3 | 2.0342 |
> | Qwen3 | 3.6416 |
> | Llama4 | 1.5639 |
> | Llamo | 4.704664 |
> | Omnimol | 1.2917 |
>
> Table 2: OOD results on Lipophilicity
>
> | Method | MAE |
> | --- | --- |
> | DeepSeekV3 | 0.9842 |
> | Qwen3 | 3.4005 |
> | Llama4 | 1.9060 |
> | Llamo | 2.287543 |
> | Omnimol | 1.1690 |
>
> In conclusion, this further confirms that OmniMol, universally instruction-tuned on 16 molecular tasks, is a powerful generalist molecular model that makes a significant and lasting contribution to the community.

---

> > ### Author Response · Authors · 2025-08-09
> >
> > Dear Reviewer 8Wa4,
> >
> > Thank you very much for your professional feedback. We believe you have a deep understanding of the molecular LLM field, and we sincerely hope you can share further thoughts on our discussion. We are more than willing to address any additional concerns you may have. Thank you again for your time and expertise. With only 6 hours left before the discussion ends, we look forward to your reply!

---

> > > ### Author Response · Authors · 2025-08-09
> > >
> > > Dear Reviewer 8Wa4,
> > >
> > > Thank you for your expert feedback. We appreciate your deep understanding of molecular LLMs and would welcome any further discussion. We are ready to address any additional concerns. With just over 4 hours remaining in the discussion period, we look forward to your reply !

---

> > > > ### Author Response · Authors · 2025-08-09
> > > >
> > > > Dear Reviewer 8Wa4,
> > > >
> > > > Thank you for the highly professional feedback, which demonstrates your strong grasp of the molecular LLM field. We are keen to continue this conversation and address any remaining questions you may have. With just over **2 hours** remaining before the discussion concludes, we would be extremely grateful for a prompt reply. Thank you again for lending us your time and expertise.

---

> > > > ### Comment · Reviewer_8Wa4 · 2025-08-09
> > > >
> > > > Thank you for the further clarification and additional experimental results.
> > > >
> > > > I acknowledge that SELFIES is beneficial in certain cases, that the training of the multimodal projector is reasonable, that the proposed model architecture (GAL) is effective in addressing the targeted points, and that the proposed Omni-Mol model is robust to permutation.
> > > >
> > > > However, I still have concerns about the following points:
> > > > - Originality of the proposed dataset: While the authors have made efforts in curating the dataset, including adding instructions, unification, and de-duplication, it is primarily composed of existing curated datasets, not introducing a truly novel task.
> > > > - Generalizability: Although the proposed model is designed for generalizability, its performance is moderate on OOD tasks.
> > > >
> > > > Overall, I raise my score accordingly.

---

> ### Author Response · Authors · 2025-08-09
>
> **Dear Reviewer 8Wa4,**
>
> Thank you very much for your reply to our comments and for recognizing our contributions. We have learned a great deal from your suggestions and feedback, which have been instrumental to improving our work. We sincerely appreciate the time and effort you have devoted to our paper.
>
> Regarding your concerns:
>
> > Originality of the proposed dataset
> >
>
> The molecular editing (MolEdit) task is novel in the context of **instruction-following** datasets, comprising approximately **220K** samples. We independently designed and authored the instructions, formulating the dataset explicitly in an instruction-following format. Furthermore, we optimized the computational tool for the MolEdit task’s evaluation metric (see Appendix E.4 and Appendix B.3), ensuring a fairer comparison between our proposed model and the baselines. We also provide case studies in Appendix G.3 to illustrate the behavior of the LLM on this newly introduced task. In addition, we conducted a statistical analysis of the entire dataset, reporting metrics such as Atom Count, Ring Count, Molecular Weight, and Bertz Complexity. This analysis offers a more **comprehensive and transparent characterization** of the dataset.
>
> > Generalizability
> >
>
> We acknowledge OOD performance is an interesting topic. However, Omni-Mol, a **generalist** model capable of handling diverse tasks (Any-to-Any modality) using a **single set** of model weights while maintaining competitive performance (as detailed in Appendix I, line 1114), is not **designed** for OOD generalizability. The architectural design emphasizes **balancing multi-task learning** and **enhancing performance** across more than 10 tasks.
>
>
> Once again, we sincerely thank you for your time, expertise, and constructive engagement during the discussion period.

---

> ### Author Response · Authors · 2025-08-09
>
> Dear Reviewer 8Wa4,
>
> Thank you once again for your highly professional comments. Your profound insight into molecular LLMs has been incredibly valuable to us. Once again, we sincerely appreciate the time and effort you devoted to the review.
>
> As the discussion period is scheduled to conclude in 30 minutes, we wish to let you know that we will be standing by. Should you have any final thoughts or questions about our response, we would be more than happy to provide further clarification.
>
> We are truly grateful for your comment!

---

### Official Review · Reviewer_m1Tk · 2025-06-23

**Clarity:** 2
**Significance:** 3
**Originality:** 3
**Rating:** 5
**Confidence:** 4

**Summary:**

This work presents Omni-Mol, a model for molecular tasks enabling textual- and molecular inputs and outputs. Omni-Mol uses an LLM-based backbone and further includes a GNN-based graph tokenizer, as well as two methods for handling task-diversity common in molecular learning: GAL is a LoRA variant which can more flexibly learn the optimal rank for different tasks, and MoGE is a MoE approach which allows for different experts to handle different input modalities. Omni-Mol always performs among the best models in a variety of tasks and is ablated to study the effects of different architectural design choices.

**Questions:**

- Q1: Can the authors please provide information about the runtime needed to train Omni-Mol?
- Q2: Can the authors elaborate on the necessity for the GNN encoder?
- Q3: Could the authors explain whether simply treating SELFIES as text and fine-tuning an LLM on interleaved natural language and SELFIES could be a competitive approach to Omni-Mol, or whether such a comparison was considered?

**Ethical Concerns:**

["NO or VERY MINOR ethics concerns only"]

**Final Justification:**

The authors have provided a very solid rebuttal overall and largely addressed my concerns. Overall I find this work to be highly relevant,  and the experimental study conducted well, in particular after error bars and the LLM-only baseline were added during the rebuttal. As a result, I have decided to increase my score.

**Limitations:**

yes

**Paper Formatting Concerns:**

No concerns.

**Quality:**

2

**Strengths And Weaknesses:**

### Strengths

- S1: The work considers a very flexible setting which allows one to interleave natural language, with graph- as well as SELFIES-based molecular information.
- S2: It is an interesting finding that molecular models benefit from MoE to handle multiple input modalities.
- S3: The topic is very interesting and the move towards flexible multi-modal molecular models is important and timely.

### Weaknesses
- W1: It does not become clear whether the GNN-based encoder is truly essential for Omni-Mol. This part of the framework is not properly ablated, yet it seems that training Omni-Mol would become much simpler without the additional GNN representation. The paper justifies this design choices poorly. The authors should at least properly justify this design choice in plain text and ideally perform an ablation study to identify whether the GNN is essential for Omni-Mol. Perhaps I am also overlooking that the GNN is architecturally necessary. In that case, the authors should explain this more clearly in the main paper.
- W2: The experimental results lack error bars or standard deviation. This is despite the fact that the authors claim in the paper checklist that they consider statistical significance. Without standard deviations, the significance arising from the experimental results is questionable. I strongly suggest that the authors repeat the downstream tasks on multiple random seeds to estimate the standard deviation. While the authors do not specify the training time required for one training run of Omni-Mol, I would understand that training multiple models is not feasible within the time of rebuttal. In that case, I recommend to at least run the inference, which inhibits some stochasticity due to the decoder loop of the LLM, multiple times, as described here: https://arxiv.org/abs/2411.00640.
- W3: In my opinion, the paper lacks a very important baseline: Fine-tuning the same backbone LLM as used in Omni-Mol, using the same data as Omni-Mol, simply treating SELFIES as text. As far as I understand, this baseline is not considered, yet would allow to estimate the actual improvement arising from treating multiple molecular modalities differently.

---

> ### Author Rebuttal · Authors · 2025-07-31
>
> > Q1: Ablation on GNN-based encoder.
>
> A1: Thanks for the comment. We follow a long established line of works [1, 2, 3] that uses GNN features as multi-modal feature, which can enhance the model’s ability to understand molecules’ stuctures. To further validate the role of the GNN component within our framework, we conducted an additional ablation study. Due to time constraints, the experiments were carried out on a subset of the tasks. Specifically, we remove the graph feature derived by projector and re-train a pure text model that uses only features from LLM’s word embedding, the results are shown below.
>
> Molcap
> | Method    | BLEU-2 | BLEU-4 | METEOR | ROUGE-1 | ROUGE-2 | ROUGE-L |
> |-----------|--------|--------|--------|---------|---------|---------|
> | Omni-Mol   | 0.511  | 0.421  | 0.556  | 0.593   | 0.434   | 0.530   |
> | w/o GNN   | 0.495  | 0.408  | 0.542  | 0.582   | 0.423   | 0.521   |
>
> Reagent
> | Method    | Exact  | BLEU  | RDK   | MACCS | Morgan | Leven   | val  |
> |-----------|--------|-------|-------|-------|--------|---------|------|
> | Omni-Mol   | 0.266  | 0.749 | 0.586 | 0.651 | 0.542  | 14.026  | 1.000 |
> | w/o GNN   | 0.258  | 0.738 | 0.591 | 0.650 | 0.546  | 14.126  | 1.000 |
>
> Retrosynthesis
> | Method    | Exact  | BLEU  | RDK   | MACCS | Morgan | Leven  | val  |
> |-----------|--------|-------|-------|-------|--------|--------|------|
> | Omni-Mol   | 0.594  | 0.962 | 0.861 | 0.910 | 0.828  | 8.386  | 1.00 |
> | w/o GNN   | 0.577  | 0.963 | 0.855 | 0.909 | 0.822  | 8.779  | 1.000 |
>
> Yield
> | Method    | BH     | SM     |
> |-----------|--------|--------|
> | Omni-Mol   | 0.953  | 0.688  |
> | w/o GNN   | 0.9025 | 0.6299 |
>
> Catalyst
> | Model     | Exact  | BLEU  | RDK   | MACCS | Morgan | Leven | val  |
> |-----------|--------|-------|-------|-------|--------|-------|------|
> | Omni-Mol   | 0.742  | 0.794 | 0.911 | 0.899 | 0.747  | 1.843 | 1    |
> | w/o GNN   | 0.744  | 0.845 | 0.906 | 0.885 | 0.750  | 1.784 | 1.000 |
>
> Quantum Mechanics Property Prediction
> | Method    | HOMO  | LUMO  | GAP   | AVG   |
> |-----------|-------|-------|-------|-------|
> | Omni-Mol   | 0.0038| 0.0047| 0.0049| 0.0044|
> | w/o GNN   | 0.0038| 0.0037| 0.0044| 0.0040|
>
> We can see that Omni-Mol benefits significantly from the GNN encoder on Molcap, Reagent, Yield etc., which particularly rely on the structural information of molecules. For other tasks, we observe similar performance with or without the GNN, indicating that Omni-Mol relies more heavily on textual information to generate responses in those scenarios.
>
> > Q2: The experimental results lack error bars or standard deviation.
>
> A2: Thanks for the comment. Following the guidance provided in the cited paper, we varied the sampling temperature and random seed to perform multiple inference runs. The experimental results are shown below.
>
> |Task|Exact|BLEU|LEVEN|MACCS|RDK|MORGAN|Val|
> |-|-|-|-|-|-|-|-|
> |Forward|0.763±0.001|0.982±0.001|5.282±0.055|0.949±0.000|0.906±0.001|0.886±0.001|1±0.000|
> |Reagent|0.276±0.004|0.742±0.002|14.033±0.070|0.660±0.003|0.596±0.004|0.555±0.003|0.999±0.000|
> |Retrosynthesis|0.574±0.002|0.961±0.000|8.325±0.067|0.915±0.001|0.870±0.001|0.835±0.001|0.999±0.000|
> |Solvent|0.548±0.001|0.771±0.001|2.603±0.008|0.692±0.001|0.694±0.002|0.666±0.002|1±0.000|
> |Catalyst|0.746±0.003|0.781±0.003|2.153±0.068|0.893±0.002|0.913±0.001|0.751±0.003|1±0.000|
> |TextGuidedMolGen|0.133±0.001|0.834±0.001|22.324±0.123|0.730±0.001|0.576±0.002|0.452±0.002|0.961±0.001|
>
> |Task|LOGP|TPSA|WEIGHT|
> |-|-|-|-|
> |Mol2Num|0.460±0.002|6.225±0.058|11.257±0.031|
>
> |Task|HOMO|LUMO|GAP|AVG|
> |-|-|-|-|-|
> |Quantum Mechanics|0.0038±0.000|0.0038±0.000|0.0047±0.000|0.0041±0.000|
>
> |Task|BH|SM|
> |-|-|-|
> |Yield|0.898±0.031|0.632±0.013|
>
> |Task|BLEU-2|BLEU-4|METEOR|ROUGE-1|ROUGE-2|ROUGE-L|
> |-|-|-|-|-|-|-|
> |Molcap|0.536±0.001|0.447±0.001|0.574±0.001|0.605±0.001|0.449±0.001|0.542±0.001|
> |Experiment|0.574±0.001|0.450±0.001|-|0.533±0.001|0.275±0.001|0.466±0.001|
>
> > Q3: Lacks a baseline with pure text.
>
> A3: Thanks for the comment. Please refer to A1 above.
>
> > Q4: Runtime to train Omni-Mol
>
> A4: Thanks for the comment. The runtime for 15 tasks in total is 3 * 24 * 8 = 576 A100 80G GPU hours.
>
> [1] Cao, H., Shao, Y., Liu, Z., Liu, Z., Tang, X., Yao, Y. and Li, Y., 2024. PRESTO: Progressive pretraining enhances synthetic chemistry outcomes. arXiv preprint arXiv:2406.13193.
>
> [2] Li, S., Liu, Z., Luo, Y., Wang, X., He, X., Kawaguchi, K., Chua, T.S. and Tian, Q., 2024. Towards 3d molecule-text interpretation in language models. arXiv preprint arXiv:2401.13923.
>
> [3] Cao, H., Liu, Z., Lu, X., Yao, Y. and Li, Y., 2023. Instructmol: Multi-modal integration for building a versatile and reliable molecular assistant in drug discovery. arXiv preprint arXiv:2311.16208.

---

> > ### Author Response · Authors · 2025-08-02
> > **Supplementary Response to Reviewer m1Tk**
> >
> > Thank you very much for your positive and encouraging feedback on our work. We truly appreciate your recognition. From your review, it is evident that you have developed a deep and accurate understanding of our multimodal architecture.
> >
> > > First, regarding the GNN encoder, we have demonstrated through extensive experiments that it is indispensable for tasks that heavily rely on chemical structure information, while introducing minimal interference in other tasks. This provides strong evidence for the effectiveness of the GNN encoder as a modality encoder for graph-based information.
> > >
> >
> > > Second, concerning the lack of error bars in our original results, we followed the convention of mainstream papers, which typically do not perform multiple inferences. In response to your suggestion, and building on Reviewer fT5E’s recommendation to explore scaling, we increased the number of experts to 16 and conducted additional experiments with varied sampling temperatures. These repeated trials allowed us to report error bars. For detailed results, please refer to our response in A2.
> > >
> >
> > Thank you for taking the time to help us improve the quality of our manuscript. If you have any further suggestions, we would be more than happy to engage in further discussion with you.

---

> ### Comment · Reviewer_m1Tk · 2025-08-05
>
> I want to thank the authors for their rebuttal which has largely addressed my concerns. I am not fully convinced that the GNN encoder is really an essential component of Omni-Mol, given the results above. While the GNN definitely improves scores on some tasks, there are also quantum-chemistry tasks where I would expect the structure to be rather important but Omni-Mol w/o GNN seems to perform better. That said, I do not regard this as a fundamental weakness of Omni-Mol, rather an empirical observation which deserves attention, in particular regarding future work on developing approaches like Omni-Mol further. I encourage the authors to add these results to the paper, accompanied by a nuanced discussion of the importance of the GNN encoder.

---

> > ### Author Response · Authors · 2025-08-05
> > **Official Comment by Authors**
> >
> > Dear Reviewer m1Tk,
> >
> > Thank you for providing such a detailed clarification on your concern. We deeply appreciate the time and effort you have invested in reviewing our work and offering these detailed comments.
> >
> > We will certainly add these results along with comprehensive ablation studies to the paper, in order to provide clearer guidance for the advancement of the field. In future versions, we will also be committed to developing molecular models that better integrate graph and textual modalities.
> >
> > Thank you again for your time and expertise. Your feedback has been instrumental in enhancing the quality of our work.

---

> > ### Author Response · Authors · 2025-08-06
> >
> > Dear Reviewer m1Tk,
> >
> > We sincerely thank you for your thoughtful feedback. If you have any further questions or concerns regarding our work, **please do not hesitate to let us know**, we would be more than happy to engage in further discussion.
> >
> > Sincerely,
> > Submission 1411 Authors

---

### Official Review · Reviewer_fT5E · 2025-06-24

**Clarity:** 3
**Significance:** 3
**Originality:** 3
**Rating:** 4
**Confidence:** 3

**Summary:**

This paper introduces Omni-Mol, a unified and scalable multimodal large language model tailored for molecular science, capable of handling a wide spectrum of tasks across four modalities: Mol2Mol, Mol2Text, Mol2Num, and Text2Mol. The framework leverages a novel Gradient Adaptive LoRA (GAL) mechanism and a large-scale multitask instruction dataset (>1.3M samples across 16 tasks) to enable effective parameter-efficient fine-tuning. The authors report strong performance on a variety of benchmark tasks, and their model surpasses several larger-scale baselines in terms of both effectiveness and efficiency.

**Questions:**

1. How sensitive is GAL’s performance to hyperparameter tuning?

2. How well does the model generalize to truly novel molecular tasks not seen in training?

3. In tasks requiring multiple molecules as input (e.g., reactions),  how graph encoder handle multi-input task?

**Ethical Concerns:**

["NO or VERY MINOR ethics concerns only"]

**Limitations:**

1. Due to limited computational resources, the current study is unable to scale Omni-Mol to larger parameter sizes. This restricts the exploration of potential performance gains from model scaling or deeper expert routing.

2. Although instruction tuning is central to the framework, many prompts are manually templated and task-specific. There is no mechanism for dynamic prompt generation or adaptation, potentially limiting model usability by non-expert chemists or in unstructured settings.

**Paper Formatting Concerns:**

No concerns regarding formatting; the presentation is clean and conforms to the conference template.

**Quality:**

3

**Strengths And Weaknesses:**

## Strengths
1. The authors propose a novel and comprehensive multimodal LLM framework that supports four distinct categories of molecular tasks: Mol2Mol, Mol2Text, Mol2Num, and Text2Mol—this "any-to-any" modality design.

2. The authors propose Gradient Adaptive LoRA (GAL), which dynamically adjusts low-rank update magnitudes based on task-specific intrinsic dimensions, thereby enhancing multi-task adaptability.

3. The authors curated a dataset comprising over 1.3 million samples across 16 tasks, making it the largest molecular instruction tuning dataset to date.

## Weaknesses
1. Despite the claim of “unified instruction tuning,” the work lacks an empirical analysis of robustness to instruction variability. The paper does not report performance under paraphrased or noisy instructions, nor does it assess zero-shot generalization to unseen prompts.

2. While Omni-Mol demonstrates strong results on most tasks, its performance on the Catalyst Prediction task lags behind several baselines. This suggests potential difficulty in modeling subtle structure-function relationships specific to catalysis, which may involve sparse and high-variance distributions.

---

> ### Author Rebuttal · Authors · 2025-07-31
>
> > Q1: Lack of empirical analysis of robustness to instruction variability.
>
> A1: Thanks for the comment. In our dataset, each input instruction is paraphrased using at least six different prompt templates. During evaluation, we also test the model with various paraphrased prompts rather than relying on a single fixed instruction. You can refer to the repository link for the full set of prompt templates. In fact, each sample is paired with a uniquely crafted prompt, making the model robust to varying "zero-shot prompts". We would be happy to explore further directions and experiments if you have additional suggestions.
>
> > Q2: Difficulty in modeling subtle structure-function relationships specific to catalysis.
>
> A2: Thank you for the comment. Our Omni-Mol model has already demonstrated strong performance on 13 datasets. Although it slightly underperforms PRESTO on the catalyst task, it’s important to note that PRESTO benefits from two additional advantages: (1) it undergoes multiple stages of pretraining, including approximately 3 million samples of synthetic procedure descriptions and molecule name conversions, whereas the LLM backbone of Omni-Mol is used without any pretraining; and (2) PRESTO is based on a 7B-parameter model, while Omni-Mol uses a significantly smaller backbone of around 2B parameters, leaving substantial room for future improvement.
>
> > Q3: How sensitive is GAL’s performance to hyperparameter tuning?
>
> A3: Thanks for the comment. We tune the rank of GAL and the results are shown in the table below. Due to the time limit, we train the model on 8 tasks.
>
> Catalyst
> | Method    | Exact | BLEU  | RDK   | MACCS | Morgan | Leven | val   |
> |-----------|-------|-------|-------|-------|--------|-------|-------|
> | RANK=32   | 0.733 | 0.797 | 0.903 | 0.887 | 0.739  | 1.948 | 1.000 |
> | RANK=128  | 0.751 | 0.815 | 0.910 | 0.893 | 0.758  | 1.911 | 1.000 |
>
> Forward
> | Method    | Exact | BLEU  | RDK   | MACCS | Morgan | Leven | val   |
> |-----------|-------|-------|-------|-------|--------|-------|-------|
> | RANK=32   | 0.686 | 0.978 | 0.864 | 0.928 | 0.840  | 7.458 | 1.000 |
> | RANK=128  | 0.783 | 0.986 | 0.906 | 0.952 | 0.888  | 4.714 | 1.000 |
>
> Quantum Mechanics Property Prediction
> | Method   | HOMO  | LUMO  | GAP   | AVG   |
> |----------|-------|-------|-------|-------|
> | RANK=32  | 0.0039| 0.0040| 0.0047| 0.0042|
> | RANK=128 | 0.0037| 0.0040| 0.0046| 0.0041|
>
> Molcap
> | Method   | BLEU-2 | BLEU-4 | METEOR | ROUGE-1 | ROUGE-2 | ROUGE-L |
> |----------|--------|--------|--------|---------|---------|---------|
> | RANK=32  | 0.477  | 0.384  | 0.531  | 0.571   | 0.406   | 0.508   |
> | RANK=128 | 0.515  | 0.426  | 0.562  | 0.596   | 0.438   | 0.533   |
>
> Reagent
> | Method   | Exact | BLEU  | RDK   | MACCS | Morgan | Leven   | val   |
> |----------|-------|-------|-------|-------|--------|---------|-------|
> | RANK=32  | 0.200 | 0.679 | 0.509 | 0.588 | 0.472  | 16.711  | 1.000 |
> | RANK=128 | 0.283 | 0.749 | 0.591 | 0.655 | 0.557  | 13.624  | 1.000 |
>
> Retrosynthesis
> | Method   | Exact | BLEU  | RDK   | MACCS | Morgan | Leven  | val   |
> |----------|-------|-------|-------|-------|--------|--------|-------|
> | RANK=32  | 0.525 | 0.955 | 0.827 | 0.891 | 0.789  | 10.389 | 1.000 |
> | RANK=128 | 0.593 | 0.961 | 0.860 | 0.911 | 0.832  | 8.343  | 1.000 |
>
> Solvent
> | Method   | Exact | BLEU  | RDK   | MACCS | Morgan | Leven | val   |
> |----------|-------|-------|-------|-------|--------|-------|-------|
> | RANK=32  | 0.461 | 0.718 | 0.603 | 0.612 | 0.573  | 3.037 | 1.000 |
> | RANK=128 | 0.588 | 0.795 | 0.730 | 0.726 | 0.706  | 2.295 | 1.000 |
>
> Yield
> | Method   | BH     | SM     |
> |----------|--------|--------|
> | RANK=32  | 0.8877 | 0.6114 |
> | RANK=128 | 0.9430 | 0.6674 |
>
> In the experimental results, we observed that an increase in rank leads to performance improvements, validating the robustness of our method. The growth effect are not particularly evident in Quantum Mechanics Property Prediction task, but significant increases were observed in other tasks.
>
> > Q4: How well does the model generalize to truly novel molecular tasks not seen in training?
>
> A4: Thanks for the comment. Generalizing to out-of-domain data is indeed an interesting aspect. Coincidentally, one of the reviewers pointed out that we did not fully consider the molecular graph classification task. This gives us a great opportunity to use it as a testbed to evaluate the generalization capability of our model. We directly implement the zero-shot evaluation on three different classification datasets [1, 2]. Since the original pretrained weights of many baseline models are not publicly available, we use Qwen3, which represents the state-of-the-art on scientific tasks, as our primary comparison.
>
> BBBP
> | Model        | F1 Score            | Accuracy   |
> |--------------|---------------------|------------|
> | Omni-Mol      | 0.5725  | 0.4363     |
> | Qwen3      | 0.0617 | 0.23039    |
>
> HIV
> | Model       | F1 Score            | Accuracy   |
> |-------------|---------------------|------------|
> | Omni-Mol     | 0.0254 | 0.0650     |
> | Qwen3  | 0.000  | 0.03383    |
>
> SIDER
> | Model             | F1 Score            | Accuracy   |
> |------------|---------------------|------------|
> | Omni-Mol   | 0.4054 | 0.3846     |
> | Qwen3 | 0.1553  | 0.4891     |
>
> As the experimental results show, Omni-Mol demonstrates certain performance on completely unseen classification tasks, slightly outperforming Qwen3 in some tasks and metrics.
>
> > Q5: How graph encoder handle multi-input task?
>
> A5: Thanks for the comment. When multiple molecules involved, we stack them into batch dimensions and feed them to the graph encoder together, the resulting graph features are correspondingly concatenated together.
>
> > Q6: Limited explorations on model scaling.
>
> A6: Thanks for the comment. Despite currently using only a 2B model with 5 experts, we have already achieved state-of-the-art performance on the majority of tasks. We believe Omni-Mol has strong potential, and we will include scaling results for larger size models in the next version.
>
> > Q7: No mechanism for dynamic prompt generation.
>
> A7: Thanks for the comment. You make a good point regarding the deployment of general-purpose molecular models. In practice, we do apply prompt engineering techniques, for example, calling GPT-4 to refine or polish the instructions. However, these strategies fall outside the scope of the technical contributions of this paper, and thus are not discussed.
>
> [1] Liu, S., Nie, W., Wang, C., Lu, J., Qiao, Z., Liu, L., Tang, J., Xiao, C. and Anandkumar, A., 2023. Multi-modal molecule structure–text model for text-based retrieval and editing. Nature Machine Intelligence, 5(12), pp.1447-1457.
>
> [2] Michael Kuhn, Ivica Letunic, Lars Juhl Jensen, and Peer Bork. "The SIDER database of drugs and side effects". In: Nucleic acids research 44.D1 (2015), pp. D1075–D1079.

---

> > ### Author Response · Authors · 2025-08-02
> > **Supplementary Response to Reviewer fT5E**
> >
> > Thank you very much for your positive and encouraging feedback on our work. We truly appreciate your recognition. From your review, it is clear that you have gained a deep and accurate understanding of our method.
> >
> > > For the concern of limitation of scaling, Omni-Mol achieves strong performance with a relatively small number of parameters. Therefore, we believe that scaling is a meaningful direction to explore. We increase the number of experts from 5 to 16 to perform expert scaling, and observe a further comprehensive improvement in experimental results. The results are shown as below. All of the codes are totally open-sourced in our anonymous GitHub repository.
> >
> > ### Forward Pred
> >
> > | Experts | Exact | BLEU | LEVEN | MACCS | RDK | MORGAN | Val |
> > | --- | --- | --- | --- | --- | --- | --- | --- |
> > | 5 | 0.73   | 0.980  | 5.55 | 0.895  | 0.947  | 0.87 | 1 |
> > | 16 | **0.76** | **0.981** | **5.29** | **0.949** | **0.964** | **0.89** | **1** |
> >
> > ### Reagent Pred
> >
> > | Experts | Exact | BLEU | LEVEN | MACCS | RDK | MORGAN | Val |
> > | --- | --- | --- | --- | --- | --- | --- | --- |
> > | 5 | 0.23  | 0.726  | 14.59  | 0.627 | 0.557  | 0.52 | 1 |
> > | 16 | **0.27** | **0.743** | **13.93** | **0.654** | **0.589 **| **0.55** | 1 |
> >
> > ### Retrosynthesis
> >
> > | Experts | Exact | BLEU | LEVEN | MACCS | RDK | MORGAN | Val |
> > | --- | --- | --- | --- | --- | --- | --- | --- |
> > | 5 | 0.57   | 0.960  | 8.97  | 0.909  | 0.864  | 0.83 | 1.00 |
> > | 16 | **0.58** | **0.961** | **8.24** | **0.915** | **0.872** | **0.84** | 0.999 |
> >
> > ### Solvent Pred
> >
> > | Experts | Exact | BLEU | LEVEN | MACCS | RDK | MORGAN | Val |
> > | --- | --- | --- | --- | --- | --- | --- | --- |
> > | 5 | 0.52   | 0.759 | 2.71  | 0.671 | 0.673  | 0.64 | 1 |
> > | 16 | **0.55** | **0.769** | **2.61** | **0.689** | **0.691** | **0.66** | 1 |
> >
> > ### Catalyst Pred
> >
> > | Experts | Exact | BLEU | LEVEN | MACCS | RDK | MORGAN | Val |
> > | --- | --- | --- | --- | --- | --- | --- | --- |
> > | 5 | 0.72  | 0.792 | 1.96  | 0.904  | 0.886  | 0.72 | 1 |
> > | 16 | **0.74** | 0.792 | 2.10 | 0.900 | **0.912** | **0.75** | 1 |
> >
> > ### MOL2NUM
> >
> > | Experts | LOGP | TPSA | WEIGHT |
> > | --- | --- | --- | --- |
> > | 5 | 0.49 | 5.89 | 11.07 |
> > | 16 | **0.45** | 6.13 | **11.05** |
> >
> > ### HOMO LUMO
> >
> > | Experts | HOMO | LUMO | GAP | AVG |
> > | --- | --- | --- | --- | --- |
> > | 5 | 0.0038    | 0.0047 | 0.0049 | 0.0044 |
> > | 16 | 0.0038 | **0.0039** | **0.0047** | **0.0041** |
> >
> > ### Yield
> >
> > | Experts | BH | SM |
> > | --- | --- | --- |
> > | 5 | 0.94 | 0.68 |
> > | 16 | **0.95** | 0.67 |
> >
> > ### Experiment
> >
> > | Experts | BLEU-2 | BLEU-4 | ROUGE-1 | ROUGE-2 | ROUGE-L |
> > | --- | --- | --- | --- | --- | --- |
> > | 5 | 0.572  | 0.448  | 0.532  | 0.274  | 0.464 |
> > | 16 | **0.573** | **0.449** | 0.532 | 0.274 | **0.465** |
> >
> > ### Molcap
> >
> > | Experts | BLEU-2 | BLEU-4 | METEOR | ROUGE-1 | ROUGE-2 | ROUGE-L |
> > | --- | --- | --- | --- | --- | --- | --- |
> > | 5 | 0.529  | 0.440  | 0.541   | 0.604 | 0.447 | 0.571 |
> > | 16 | **0.537** | **0.447** | **0.575** | **0.606** | **0.450** | 0.543 |
> >
> > > To address the concern regarding the limitations of unstructured input, we devise a method to test the robustness of input representations. Specifically, we randomly shuffle the original instructions and retained only the input **SELFIES** and key terms such as *“predict”* and *“reaction prediction”* as the prompt. We then use GPT-4 to reorganize these elements into coherent instructions. We evaluate this approach (named Omni-Mol w. R) on three major reaction prediction tasks, and the experimental results are presented below. The results indicate that our model can achieve performance comparable to the original setup as long as a sufficiently powerful LLM is used to rephrase and restructure the input.
> >
> > ### Forward Pred
> >
> > | Methods | Exact | BLEU | LEVEN | MACCS | RDK | MORGAN | Val |
> > | --- | --- | --- | --- | --- | --- | --- | --- |
> > | Omni-Mol | 0.73   | 0.980  | 5.55 | 0.895  | 0.947  | 0.87 | 1 |
> > | Omni-Mol w. R | 0.73 | 0.982 | 5.57 | 0.903 | 0.949 | 0.97 | 1 |
> >
> > ### Reagent Pred
> >
> > | Methods | Exact | BLEU | LEVEN | MACCS | RDK | MORGAN | Val |
> > | --- | --- | --- | --- | --- | --- | --- | --- |
> > | Omni-Mol | 0.23  | 0.726  | 14.59  | 0.627 | 0.557  | 0.52 | 1 |
> > | Omni-Mol w. R | 0.22 | 0.727 | 14.49 | 0.611 | 0.559 | 0.52 | 1 |
> >
> > ### Retrosynthesis
> >
> > | Methods | Exact | BLEU | LEVEN | MACCS | RDK | MORGAN | Val |
> > | --- | --- | --- | --- | --- | --- | --- | --- |
> > | Omni-Mol | 0.57   | 0.960  | 8.97  | 0.909  | 0.864  | 0.83 | 1 |
> > | Omni-Mol w. R | 0.57 | 0.961 | 8.94 | 0.912 | 0.860 | 0.83 | 1 |

---

> > ### Comment · Reviewer_fT5E · 2025-08-07
> >
> > I appreciate the authors’ thorough rebuttal. I have no further concerns and hope the final version will include the newly proposed experiments.

---

> > > ### Author Response · Authors · 2025-08-07
> > >
> > > Dear Reviewer fT5E,
> > >
> > > Thank you for your insightful comments. We deeply appreciate the time and effort you have invested in reviewing our work. In the final version of paper, we will be sure to incorporate these newly results along with comprehensive ablation studies. We believe these additions will offer clearer guidance for the advancement of the field by providing a more in-depth analysis of Omni-Mol's capabilities.
> > >
> > > Thank you again for your time and expertise. Your feedback has been instrumental in enhancing the quality of our work.

---

### Note · Authors · 2025-08-12

We sincerely thank all of your effort. We aim to answer three key concerns raised by the reviewers here:

1. **OOD performance of Omni-Mol (Reviewers fT5E and 8Wa4)**

    We would like to clarify that OOD generalization is *not* the primary target of generalist model. We are the SOTA generalist to date that can solve the most comprehensive molecular tasks. That said, our focus is on balancing and effectively learning across 16 tasks within a single unified framework. We also conducted additional evaluations on datasets outside the training set and found that Omni-Mol delivers competitive performance compared to commercial models and substantially outperforms similar open-source generalist baselines such as Llamo.

2. **Novelty of Dataset (Reviewer FtzT and 8Wa4 )**

    **First**, we proposed the most comprehensive molecular instruction-tuning dataset (2.8B tokens) in academic and puts a lot of efforts. **Second**, some subsets like MolEdit (important for property-oriented drug discovery) are truly novel and are the first time to be constructed originally.  We independently designed and authored the instructions, formulating the dataset explicitly in an instruction-following format. Furthermore, we optimized the computational tool for the MolEdit task’s evaluation metric (see Appendix E.4 and Appendix B.3), ensuring a fairer comparison between our proposed model and the baselines. **Third**, we conducted a statistical analysis of the entire dataset, reporting metrics such as Atom Count, Ring Count, Molecular Weight, and Bertz Complexity. This analysis offers a more **comprehensive and transparent characterization** of the dataset.

3. **Technical delta is small (Reviewer FtzT)**

    No existing baseline has been able to fully handle the range of tasks. The main challenge lies in the instability of LLM training when faced with highly diverse task types—simply adding more task-specific data does not lead to effective scaling. OmniMol is the first to address this problem by introducing GAL-MoE, a carefully designed mechanism that enables stable and scalable multi-task learning.

4. **Use of SELFIES and GNN (Reviewer m1Tk,  FtzT and 8Wa4)**

    We supplemented ablation on GNN features, and found that multi-modal features indeed can provide structural information and benefit the performance of Omni-Mol . Regarding SELFIES, we include ablation experiments demonstrating that SELFIES consistently improves robustness in our setting.

---

### Decision · Program_Chairs · 2025-09-17

**Decision:**

Accept (poster)

**Comment:**

This paper introduces a multimodal LLM framework supporting four categories of molecular tasks (Mol2Mol, Mol2Text, Mol2Num, Text2Mol), enabled by an “any-to-any” modality design. The work also proposes Gradient Adaptive LoRA (GAL) to enhance adaptability in multi-task learning and contributes a large-scale dataset of over 1.3M samples across 16 tasks. The topic is timely and significant, as generalist molecular LLMs have the potential to benefit both computer science and chemistry communities.

Strengths include the flexible multimodal setting, the use of MoE architectures, and a well-motivated GAL approach with clear ablation studies. The paper is generally well-written and presents results convincingly. Weaknesses include modest originality, as prior work (e.g., Text+ChemT5) can already perform most of the proposed tasks, some concerns about potential cherry-picked results, and minor clarity issues with figure organization.

Overall, this is an interesting and valuable contribution that advances the development of multimodal molecular LLMs. Most reviewers recommend acceptance.